

# Biogeochemical characteristics of suspended particulates at deep chlorophyll maximum layers in the East China Sea

Qianqian Liu[1], Selvaraj Kandasamy[1,2], Baozhi Lin[1], Huawei Wang[1], and Chen-Tung Arthur Chen[3]

[1]State Key Laboratory of Marine Environmental Science, Xiamen University, Xiamen 361102, China
[2]Department of Geological Oceanography, College of Ocean and Earth Sciences, Xiamen University, Xiamen 361102, China
[3]Department of Oceanography, National Sun Yat-sen University, Kaohsiung 80424, Taiwan, R.O.C.
*Correspondence to*: Selvaraj Kandasamy (selvaraj@xmu.edu.cn)

**Abstract.** Continental shelves and marginal seas are key sites of particulate organic matter (POM) production, remineralization and sequestration, playing an important role in the global carbon cycle. Elemental and stable isotopic compositions of organic carbon and nitrogen are frequently used for characterizing organic matter and distinguishing their sources in suspended particulates and surface sediments in the marginal seas. Here we investigate suspended particulate matters (SPM) collected from the deep chlorophyll maximum (DCM) layer in the continental shelf of the East China Sea for particulate organic carbon and nitrogen (POC and PN) contents and their isotopic compositions ($\delta^{13}C_{POC}$ and $\delta^{15}N_{PN}$) to understand biogeochemical characteristics of POM straddling at biotic-dominated DCM depths. When combined with hydrographic parameters, such as temperature, salinity and turbidity, and chlorophyll *a* (Chl *a*), these elemental and isotopic results revealed that POM in the DCM layers was largely from the newly-produced, *in situ* phytoplankton-dominated OM and have wider $\delta^{13}C_{POC}$ and $\delta^{15}N_{PN}$ compositions than previously thought. As supported by the POC to Chl *a* ratio, a large variation of $\delta^{13}C_{POC}$ was resulted from the changes in primary productivity and phytoplankton species, whereas the nutrient status and $\delta^{15}N$ of dissolved nitrate were the main controlling factors of $\delta^{15}N_{PN}$ variability in the DCM layers. Consistently, the spatial distribution of $\delta^{15}N_{PN}$ showed a similarity with the current pattern in the East China Sea, with $^{15}N$-enriched freshwater in the coastal region and Kuroshio Water in the northeast of Taiwan Island, but nutrient-depleted Taiwan Warm Current Water in the mid-shelf; as the latter seems to have promoted the $N_2$-fixation, resulting in the depleted $\delta^{15}N_{PN}$ in the mid-shelf. Furthermore, SPM investigated here seems not to be influenced by the terrestrial organic matter supplied by the Yangtze River (Changjiang) in summer 2013, a finding that is contrary to a number of previous studies' conclusion. Nonetheless, given the complications associated with stable isotopes of organic matter, additional parameters such as radiocarbon and biomarkers are crucial to



revalidate whether or not SPM in the DCM depths is influenced by terrestrial organic compounds in the river-dominated East China Sea.

## 1 Introduction

Stable isotopes of organic carbon and nitrogen ($\delta^{13}$C, $\delta^{15}$N) and molar carbon to nitrogen (C/N) ratio are the most frequently used natural tracers for identifying the source and fate of terrestrial organic matter (OM) in the estuarine and marine environments (Meyers, 1994; Hedges et al., 1997; Goñi et al., 2014; Selvaraj et al., 2015). This approach is based on the significant difference in $\delta^{13}$C, $\delta^{15}$N and C/N ratio between different endmembers, especially terrestrial and marine, and the assumption that only physical mixing of OM from compositionally distinct endmembers occurs in these marginal settings (Thornton and McManus, 1994; Hedges et al., 1986). Quantifying fractions of endmembers by using mass balance models thus requires known and constant values of elemental and isotopic endmembers of major sources of OM to the depositional system (e.g., Goñi et al., 2003). Any study applying mixing models for the OM source discrimination should therefore clearly identify representative values for the local sources of OM inputs into the area under investigation. However, in most cases, endmember values of $\delta^{13}$C, $\delta^{15}$N and molar C/N ratios were simply replaced by 'typical' numbers, such as ca. –20 ‰ and –27 ‰ for $\delta^{13}$C of marine phytoplankton and terrestrial plants, respectively, but without measuring endmember values in real, local or regional OM source materials. For example, isotopic values of marine phytoplankton have not been measured in a number of earlier studies that employed endmember mixing models to distinguish marine versus terrestrial organic matter in surface sediments (e.g., Kao et al., 2003; Wu et al., 2013), or these numbers simply represented by values of particulate organic matter (POM) in surface waters in the studied system (e.g., Zhang et al., 2007) or elsewhere from other ocean basins (e.g., Hale et al., 2012). Phytoplankton as the primary producer of marine OM should therefore be considered when studying the dynamics of POM in the marine water column.

It is known that stable isotopes ($\delta^{13}$C, $\delta^{15}$N) and molar C/N ratios of POM in estuarine and marine areas are representative of these values in primary production-derived OM and in that they are largely synthesized by phytoplankton (Gearing et al., 1984). The chlorophyll *a* (Chl *a*) in the sea water is often used as an index of phytoplankton biomass, and the deep chlorophyll maximum (DCM) layer, which have a significant contribution to the total biomass and primary production in the whole water column (Weston et al., 2005; Hanson et al., 2007; Sullivan et al., 2010), is approximately equal to the subsurface biomass maximum layer (e.g., Sharples et al., 2001; Ryabov et al., 2010). The formation of maximum chlorophyll concentration at the DCM has been explained by several mechanisms: the differential zooplankton grazing with depths (Riley et al., 1949; Lorenzen, 1967), adaption of the phytoplankton to light intensities or to increased concentration of nutrients (Nielsen and Hansen,





1959; Gieskes et al., 1978), chlorophyll accumulation by sinking of phytoplankton detritus (Gieskes et al., 1978; Karlson et al., 1996), and the decomposition of chlorophyll by light (Nielsen and Hansen, 1959). DCM is common in both coastal and open oceans, occurring at a relatively shallow depths (1–50 m) in coastal seas, but in deeper depths (80–130 m) in open ocean (Gong et al., 2015), and often variable in time and space (Karlson et al., 1996).

For example, the DCM layers were reported at depths of 30–50 m across the shelf in the southern East China Sea during summer from 1991 to 1995 (Gong et al., 2010). Hence, $\delta^{13}C$, $\delta^{15}N$ and molar C/N ratios of POM in DCM layers in the continental shelf waters should reflect the $\delta^{13}C$, $\delta^{15}N$ and molar C/N ratios of phytoplankton, which in turn, the elemental and isotopic compositions of marine productivity (Savoye et al., 2003; 2012; Gao et al., 2014).

East China Sea, one of the largest marginal seas in the world, receives huge quantities of freshwater (905.1 km$^3$ yr$^{-1}$; Dai et al., 2010) and organic carbon (2.93 Tg C yr$^{-1}$, Tg = 10$^{12}$ g; Qi et al., 2014) from the Yangtze River (Changjiang). Nutrient-enriched freshwater input in turn stimulates the water column productivity significantly in coastal waters compared to the open ocean. The annual primary production for the entire shelf of East China Sea

is high among the marginal seas and has been estimated to be 85 Tg C yr$^{-1}$ in 2008 (Tan et al., 2011). Several studies have been carried out on the physical, chemical and biological aspects of East China Sea, including distributions of seasonal currents (e.g., Gong et al., 2010), chemical hydrography and nutrients distribution (Chen, 1996, 2008) and phytoplankton species in the water column (e.g., Zheng et al., 2015; Jiang et al., 2015). Likewise, $\delta^{13}C$, $\delta^{15}N$ and molar C/N ratios of POM have been constrained in a limited number of transects across

the East China Sea (e.g., Wu et al., 2003; 2007a) as well as in a wide area of the western North Pacific marginal seas (Chen et al., 1996) . Nonetheless, studies on elemental ratios and stable isotopic compositions of POM in DCM layers in the continental shelf of East China Sea, especially along the indirect transport pathway of the Yangtze-derived terrestrial material to the Okinawa Trough (Chen et al., 2017), are almost unavailable. In a recent study, Gao et al. (2014) investigated these parameters in surface, DCM and bottom waters in different

seasons and years, but they focused on the northern part of the East China Sea with a scanty attention has been paid on the biogeochemical processes involved in the DCM layers. Here, we investigate $\delta^{13}C$, $\delta^{15}N$ and molar C/N ratios of suspended POM in DCM layers in the continental margin of the East China Sea, in particular the area south of the Yangtze estuary, aiming (1) to comprehend the sources of POM in DCM layers of East China Sea; (2) to understand the factors controlling $\delta^{13}C$ and $\delta^{15}N$ in DCM layers; and (3) to estimate the POC inventory

in DCM layers of East China Sea.

**2 Study area**





The East China Sea (ECS; Fig. 1) is the largest river-dominated marginal sea in the northwestern Pacific region. The continental shelf of ECS is relatively shallow (<130 m) with an average water depth of 60 m, but wide (>500 km). The Yangtze River (Fig. 1), with a catchment area of more than $1.94 \times 10^6$ km$^2$ (Liu et al., 2007), is the main source of freshwater and sediment to the continental shelf. It is the fifth largest river in terms of water discharge

(900 km$^3$ yr$^{-1}$) and the fourth largest river in terms of sediment discharge (470 Mt yr$^{-1}$) in the world (Milliman and Farnsworth, 2011).

In addition to the huge inputs of nutrients (DIN: $61.0 \pm 13.5 \times 10^9$ mol yr$^{-1}$ for the interval of 1981–2006; Chai et al., 2009, and references therein) and sediments from the Yangtze River, the ECS is characterized by the complex

circulation pattern largely driven by the seasonally reversing East Asian monsoon winds (He et al., 2014; Chen et al., 2017). The surface circulation in the shelf of ECS is characterized by the south-north China Coastal Current (CCC) in the west, northward-moving Taiwan Warm Current (TWC) in the middle part and the north-northeastward-flowing Kuroshio Current (KC) in the east (Fig. 1) (Liu et al., 2006). Changjiang Diluted Water (CDW) is a mixture of freshwater of Yangtze River and shelf water of East China Sea, characterized by a low

salinity (<30, Umezawa et al., 2014). Owing to huge freshwater input of Yangtze into the ECS, it has been believed that the CDW is the main source of CCC (Fig. 1). Because of the East Asian monsoon, where there is a strong northeast monsoon in winter and a weaker southwest monsoon in summer, the CDW flows southward along the coastline of mainland China, as a narrow jet in winter (Chen, 2008; Han et al. 2013), whereas the same spreads mainly to the northeast in summer (Isobe et al., 2004). Taiwan Warm Current (TWC) is a mixture of the

warm water from the Taiwan Strait and the intruding saline Kuroshio water; the latter is thought to be the most dominant source of heat and salt to the ECS (Su and Pan, 1987; Zhou et al., 2015). In addition, there is an upwelling of Kuroshio Subsurface Water (KSSW) in the northeast off Taiwan Island due to an abrupt change of seafloor topography in the outer shelf of ECS (dashed ellipse in Fig. 1) (Su et al., 1989; Sheu et al., 1999). The upwelled, oxygen-unsaturated KSSW is characterized by low temperature, but high salinity and high nutrient (Liu

et al., 1988; Wong et al., 1991). The water exchange rate between the East China Sea water and Kuroshio water was estimated to be about $22{,}000 \pm 9000$ km$^{-3}$ yr$^{-1}$, which is approximately 25 times the amount of Yangtze runoff into the ECS (Li et al., 1994; Sheu et al., 1999). Furthermore, Kuroshio water made up 90% of the shelf water in the ECS (Chen, 1996; Sheu et al., 1999).

The primary productivity in the ECS is limited by nitrogen deficiency in summer, but light in winter (Chen et al., 2001; Chen and Chen 2003). It showed distinct spatial and temporal variations, with the highest primary production in summer and an annual primary production of 155 g C m$^{-2}$ yr$^{-1}$, 144 g C m$^{-2}$ yr$^{-1}$ and 145 g C m$^{-2}$ yr$^{-1}$ in the northwestern ECS, southeastern ECS and the entire ECS, respectively, in 1998 (Gong et al., 2003). The




primary productivity has however decreased 86% between 1998 and 2003 due to a large number of impoundments in the drainage basin of Yangtze River (Gong et al., 2006).

## 3 Material and methods

### 3.1 Sample collection

To investigate the biogeochemical characteristics of POM in DCM layers of East China Sea, suspended particles were collected at around water depths of DCM from thirty-six stations in seven transects across the continental
shelf by the *Science 3* cruise (organized by the Institute of Oceanography, Chinese Academy of Sciences) during June 22–July 21, 2013 (Fig. 1). At each site, the physical properties of water column were recorded by a Conductivity-Temperature-Depth (CTD) rosette (Seabird, SBE911+) fitted with a Seapoint chlorophyll fluorometer to detect the fluorescence maximum. Sea water was collected using the rosette of Niskin water bottles attached with the CTD frame and then stored in PVC bottles. The volume of each water sample was measured by
graduated cylinder before filtration. Suspended particles were obtained by filtering 0.5–2 L seawater collected at the fluorescence maximum layer through 0.7 µm/47 mm Whatman Glass Fiber Filters (GF/F), which were wrapped in aluminum foil. All GF/F filters had been pre-combusted at 450 ºC for 4 h in a muffle furnace to remove the background carbon and pre-weighed for determining the concentration of suspended particulate matter (SPM). After filtration, filters were folded and wrapped again in aluminum foil and stored at –20 ºC immediately in
a freezer onboard before they were brought back to the laboratory for further analysis. In the laboratory, filters with suspended particles were freeze-dried and filters for SPM concentration were further dried in an oven at 50 ºC for 48 h. The weight difference between the dried filter and its counterpart before filtration was used to calculate the weight of SPM.

### 3.2 Analysis of POC, PN, $\delta^{13}C$ and $\delta^{15}N$

Prior to the measurement of POC and PN contents and their stable isotope values ($\delta^{13}C_{POC}$ and $\delta^{15}N_{PN}$) in SPM samples, a half of each filter was treated with 1N HCl for 16 h to remove inorganic carbon (mainly carbonate). De-carbonated sample was dried at 50 °C for 48 h in an oven for HCl evaporation. Then half of de-carbonated
filter (i.e. a quarter of the original GF/F filter) was then punched with tin capsules for further analysis. The POC and PN contents and their $\delta^{13}C_{POC}$ and $\delta^{15}N_{PN}$ compositions were measured at the Stable Isotope Facility of University of California Davis in USA, by using an elemental analyzer (EA) (Elementar Analysensysteme GmbH, Hanau, Germany) interfaced to a continuous flow isotope ratio mass spectrometer (IRMS; PDZ Europa 20–20, Sercon Ltd., Cheshire, UK). Organic carbon and nitrogen in samples were combusted at 1080 °C in a reactor and




transformed into $CO_2$ and $N_2$, respectively, which were then separated through a molecular sieve adsorption trap before entering the IRMS. During the isotopes ($\delta^{13}C_{POC}$ and $\delta^{15}N_{PN}$) analyses, different working standards (Bovine Liver, Glutamic Acid, Enriched Alanine and Nylon 6) of compositionally similar to the samples being analyzed were used and were calibrated against NIST Standard Reference Materials (IAEA–N1, IAEA–N2,

IAEA–N3, USGS–40, and USGS–41). The standard deviation is 0.2 ‰ for $\delta^{13}C$ and 0.3 ‰ for $\delta^{15}N$. Isotopic values were presented in standard δ-notation as per mil deviations relative to the conventional standards, i.e. VPDB (Vienna Pee Dee Belemnite) for carbon and atmospheric $N_2$ for nitrogen, that is $\delta X$ (‰) = [($R_{sample}$ − $R_{standard}$)/$R_{standard}$] × $10^3$, where X = $^{13}C$ or $^{15}N$, R = $^{13}C/^{12}C$ or $^{15}N/^{14}N$, $R_{sample}$ and $R_{standard}$ are the heavy ($^{13}C$ or $^{15}N$) to light ($^{12}C$ or $^{14}N$) isotope ratios of sample and standard, respectively (e.g., Selvaraj et al., 2015). Conventional

standards have an isotope value of 0 ‰ for $\delta^{13}C$ of VPDB (a marine limestone) and $\delta^{15}N$ of $N_2$ in air, respectively. In general, organic carbon synthesized by biota is usually depleted in $^{13}C$ relative to VPDB and has a negative $\delta^{13}C$ value (Fry and Sherr, 1989). On the other hand, $\delta^{15}N$ values of POM do not show a routine mainly due to their enriched oxidation states (Sigman et al., 2009).

**4 Results and interpretations**

**4.1 Hydrographic characteristics**
4.1.1. Chlorophyll *a* (Chl *a*)

The Chl *a* concentration varied up to18.0 µg $L^{-1}$ in the study area (Fig. 2). The highest Chl *a* concentration was observed in surface water of station DH3–1, and all other remaining values are less than 8.0 µg $L^{-1}$. The vertical profiles of Chl *a* show a clear maximum in the subsurface layer at around 20 m in near coastal stations and 50 m in outer shelf stations (Fig. 2). Around 70 % of SPM sampled in this study fall in the DCM layers (Chl *a* = >1.0 µg $L^{-1}$; open squares in Fig. 2), ideally representing the biogeochemical behaviors of POM straddling in DCM layers.

Based on the photosynthetically active radiation (PAR), we defined the euphotic depth, as a depth at which the PAR is 1 % of its value at the sea surface and photosynthesis can take place (Kirk, 1994; Ravichandran et al., 2012; Guo et al., 2014a). The euphotic depth increased from the inner shelf (20 m) to outer shelf (100 m) region. This is consistent with average euphotic depth of 33 m calculated based on the empirical relation: $Z_{eu}$ = 4.605/$K_d$(PAR) (Kirk, 1994), where $K_d$(PAR) = 1.22$K_d$(490) (Tang et al., 2007; Ravichandran et al., 2012) and a

mean value of 0.115 for $K_d$(490) for the East China Sea in summer was taken from Chen and Liu (2015). The presence of DCM layers near the euphotic depths suggests a close relationship between the light availability and deep chlorophyll maximum, and the OM in the SPM samples was likely to be dominated by the phytoplankton productivity.





### 4.1.2 Turbidity

The turbidity in the water column of the ECS varied from 0.0 to 20.9 Formazin Turbidity Unit (FTU) (Fig. 2). In the inner shelf region, the vertical distribution of turbidity shows an obvious downward increasing trend and these

high turbidity stations were limited along the coast (Fig. 2). This indicates sediment resuspension from the sea floor, which was probably induced by hydrodynamic forces such as tides, waves and currents in the shallow coastal region. In the outer shelf stations, the turbidity was uniformly low from the surface to the bottom. Overall, most water depths where from SPM sampled have a low turbidity (<2.0 FTU), except for stations CON02 (4.75), DH5–1 (3.44), and DH7–1 (5.52) (Fig. 2).

### 4.1.3 Temperature and salinity

Figure 3 illustrates the vertical distributions of temperature and salinity along seven transects in the ECS. In the entire study area, temperature in the water column varied from 5 ºC to 30 ºC and distinct water column

stratification was evident from the temperature profiles (Fig. 3). The temperature decreases when depth increasing, with the high temperature (>30 ºC) spreads widely in the surface water and the lowest temperature (5 ºC) was observed in stations DH7–8 and DH7–9 at water depths of 850 m and 800 m, respectively (Fig. 3). The variation of temperature at SPM sampling depths ranges from 19.1 °C to 28.2 °C, showing a general decreasing trend from the inner to outer shelf in each transect (Fig. 3).

Salinity shows a general increasing trend with water depths (Fig. 3), varying from 26.9 to 34.8 with an average value of 34.6 in the entire water column. An increasing trend of salinity from the west to east in all seven transects can be found (Fig. 3). The low salinity (<30) was constrained in the upper 10 m in four coastal stations (DH1–1, DH2–1, DH3–1, CON02; Fig. 3), wherein temperature is <24 ºC, indicating the limited influence of CDW

plume. The middle salinity (30<S<34.1) was observed at a depth interval between 10 m and 30 m in stations (DH1–1, DH1–2, DH2–1, DH2–2, DH3–1; Fig. 3), but it spreads to a depth interval between surface and 30 m in the remaining stations. High salinity was mostly prevalent at bottom depths in all stations investigated. The salinity distribution at depths of SPM sampling shows an increasing trend from the inner to outer shelf (Fig. 3) and varies from 32.7 to 34.7 with an average salinity of 34.0, indicating insignificant influence of CDW at DCM

depths in the ECS.

### 4.2 POC and PN





The concentration of SPM ranged from 1.7 to 14.7 mg L$^{-1}$ with a mean value of 4.4 mg L$^{-1}$ (Table 1). The spatial distribution of SPM shows higher values in the inner shelf region and lower values in the outer shelf region (Fig. 4), consistent with the water column turbidity (Fig. 3). Concentration of POC in the DCM layer varied between 20.4 and 263.0 μg L$^{-1}$, with a mean value of 85.5 μg L$^{-1}$ (n = 36) (Fig. 4). The PN ranged from 4.4 to 52.8 μg L$^{-1}$,

with a mean value of 17.7 μg L$^{-1}$ (n = 36). The spatial distribution of POC and PN resembles each other (Fig. 4). Higher concentrations of POC (>90 μg L$^{-1}$) and PN (>21 μg L$^{-1}$) are mostly observed in the inner shelf along the coastal line, decreasing gradually towards the offshore direction (Fig. 4). The lowest concentrations of POC and PN are observed in the easternmost stations, nearby off northeast Taiwan Island (Fig. 4). Although the concentrations of both POC and PN varied more than an order of magnitude (Fig. 4), the molar C/N ratios are

fairly uniform at DCM layers of the entire ECS, ranging from 4.1 to 6.3 with a mean ratio of 5.6±0.5 (n = 36) (Table 1).

### 4.3 δ$^{13}$C and δ$^{15}$N

Spatial distributions of δ$^{13}$C$_{POC}$ and δ$^{15}$N$_{PN}$ in SPM at around DCM layers are presented in Fig. 5. δ$^{13}$C$_{POC}$ decreased from the inner shelf to offshore region, varying widely from −25.8 ‰ to −18.2 ‰ (Table 1). The range of δ$^{15}$N$_{PN}$ is 4.2 ‰, varying between 3.8 ‰ and 8.0 ‰ (Table 1). The lowest δ$^{13}$C$_{POC}$ values (−25.8 ‰ and −25.2 ‰) are found in the Okinawa Trough, off northeast Taiwan Island, while the δ$^{15}$N$_{PN}$ values in the same locations are also relatively higher (6.73 ‰ and 7.78 ‰) than the nearby locations (Fig. 5). The spatial distribution

of δ$^{13}$C$_{POC}$ is quite similar to the spatial distribution of POC (Fig. 4), and the correlation coefficient (R$^2$) between δ$^{13}$C$_{POC}$ and POC was 0.55 (p<0.0001; Fig. 10).

### 5 Discussion

#### 5.1 Influences from water masses

In order to identify the different water sources in the study area, temperature–salinity (*T–S*) diagrams were drawn for the entire water column (Fig. 6a) as well as for the SPM sampling depth at around DCM layers (Fig. 6b). The *T–S* diagram for all the water depths (Fig. 6a) shows a convergence region at around 17 ºC, 34.6, representing

the upwelling of KSSW (Umezawa et al., 2014). There are two trends in the *T–S* diagram, indicating a mixing of three water masses: one is less saline and much colder water, mainly CDW, another is more saline and warmer, mainly Taiwan Warm Current Water (TWCW), and the third one is KSSW (Fig. 6a). The shelf water in the entire ECS in summer is mixed primarily by four water masses, CDW, SMW, KSSW, and TWCW (Fig. 6a). However, the water at depths where from SPM sampled seems to be weakly influenced by the CDW (Fig. 6b). Furthermore,

based on the *T–S* range of different water masses (Fig. 6), we further delineated the area influenced along with





water depths of three important water masses: CDW, TWCW and KSSW (Fig. 7). Interestingly, the influence of CDW was constrained only in the upper 0–10 m in four coastal stations during the sampling time, whereas TWCW influences around 0–30 m, covering three-fourths of the study area, and KSSW seems to be largely influenced the bottom water of the entire study area (Fig. 7).

In summary, although the river runoff was huge, the influence of CDW plume in the southern part of the ECS was insignificant during summer, except in summer 2003 when the plume front moved southward (Bai et al., 2014), mainly because most of the CDW plume was transported to northeastward of the Yangtze estuary to the Korean coast (Isobe et al., 2004; Bai et al., 2014; Gao et al., 2014). Meanwhile, the intrusion of TWCW and KSSW was strong in the continental shelf of the East China Sea during summer.

## 5.2 Characterization of POM in DCM layers

### 5.2.1 Molar C/N Ratio

A necessary first step in the source analysis of POM using bulk carbon and nitrogen isotopes as well as the molar carbon and nitrogen ratio is to identify the form of total nitrogen in the measured SPM, so that inorganic nitrogen is not miss-assigned into nitrogenous organic endmember (Hedges et al., 1986). The linear relationship ($R^2$ = 0.98, p<0.0001; Fig. 8a) between POC and PN suggests that nitrogen is strongly associated with organic carbon. The slope of linear regression of POC against PN corresponds to a molar C/N ratio of 5.76 (Fig. 8a). The
positive intercept on the PN axis when POC is zero representing the amount of inorganic nitrogen (~0.03 μM), indicating that essentially all nitrogen are in the organic form. The molar C/N ratios of all SPM samples (4.1–6.3) from the DCM layers of ECS are lower than the canonical Redfield ratio (6.63) (Fig. 8a), but are similar to the average molar C/N ratios of 5.6 for marine POM (Copin-Montegut and Copin-Montegut, 1983) and 6 for POM in cold, nutrient-rich waters at high latitudes (Martiny et al., 2013). The range also falls within the range of 3.8 to 17
reported for marine POM (Geider and La Roche, 2002), but is higher than an unprecedented low C/N ratio (2.65±0.19) of POM in Canada Basin, which was attributed to a dominant contribution of small sizes (<8 μm) phytoplankton to POC (Crawford et al., 2015). Wu et al. (2003) investigated the C/N ratio of POM at all depths in the *PN* transect in ECS (4.3–29.2), a standard cross-shelf section extending from the Yangtze estuary southeast to the Ryukyu Islands, crosscutting the Okinawa Trough and perpendicular to the principle axis of Kuroshio
Current (Fig. 1). Liu et al. (1998) measured the C/N ratio of POM in the surface water of ECS and found that the C/N ratio ranged from 4.0 to 26.9 with a mean ratio of 7.6 in spring and from 4.7 to 34.3 with a mean ratio of 15.2 in autumn in 1994. The authors attributed the lower C/N in spring to an intense biological activity than in autumn, and the spatial distribution of C/N was thought to be related to that of phytoplankton abundance.





Comparing with C/N ratios of POM in Wu et al. (2003) and Liu et al. (1998), the C/N ratio at DCM layers in this study is relatively uniform and low. Such distinct characteristic of low C/N ratio in DCM layers indicate a protein-enriched chemical composition in POM. As the Redfield ratio is achieved when OM contains about 45 % protein (C/N = 3.8) and 10 % nucleic acids (C/N = 2.6) (Geider and La Roche, 2002), the relatively low C/N ratio in DCM

layers is consistent with the C/N ratio of phytoplankton (<50 μm) (4.5–5.9) in surface water and zooplankton (200–363 μm) (5.3–6.4) from 10 m above the bottom to surface water of the northern ECS in July 2010 (Chang et al., 2014). In addition, Kuroshio Water and TWCW induced *Trichodesmium* (Chen et al., 1996b; Jiang et al., 2017) may be partly responsible for the low C/N ratio in POM at DCM layers. According to Mague et al. (1977) and Letelier and Karl (1996, 1998), the C/N ratio of *Trichodesmium* varied narrowly from 4.1 to 7.3, and the relatively

low C/N ratio in *Trichodesmium* was thought to be due to its nitrogen-fixing capability (Geider and La Roche, 2002; Chen et al., 2004). Moreover, the inshore-offshore decreasing trend of abundance of water mass related *Trichodesmium* in ECS during summer (Jiang et al., 2017) is consistent with the general inshore-offshore increasing distribution of $\delta^{15}$N in POM at DCM layers in this study, except for two stations in the northeast of Taiwan Island (Fig. 5), which will be discussed later.

Characteristically, a narrow range of low C/N ratios in our SPM samples confirms the lack of terrestrial signals transported mainly by the Yangtze River. We therefore suggest that the POM in DCM layers of East China Sea is dominated by marine-sourced OM with an unrecognized contribution of terrestrial OM. The low C/N ratios further restrict the assumption of degradation of nitrogen-enriched OM, which normally increases the C/N ratio than the

Redfield ratio. Therefore, the molar C/N ratio can be better explained as a source signal of OM rather than OM degradation in SPM samples investigated in this study.

### 5.2.2 POC/Chl *a* Ratio

The moderate linear correlation ($R^2$ = 0.49, p<0.0001; Fig. 8b) between POC and Chl *a* further indicates that the phytoplankton productivity is largely responsible for the POC production in the SPM samples. Moreover, the POC/Chl *a* ratio of 35.3 g g$^{-1}$ derived from the slope of a regression line (y = 35.3 (±8.56) x + 44.0 (±6.27) (Fig. 8b) is consistent with the reported POC/Chl *a* ratio of 36.1 g g$^{-1}$ for SPM samples at 40 m water depths in the ECS (Chang et al., 2003) and nearly similar to the reported POC/Chl *a* ratio of 48 g g$^{-1}$ for the DCM layer of

Northwestern Pacific (Furuya, 1990) (Table 2). However, the POC/Chl *a* ratio obtained in this study is lower than that estimated (64 g g$^{-1}$) for the sinking particles at around 20 m in the inner shelf of the ECS and 100 m at the Kuroshio region, off northeast Taiwan Island (Hung et al., 2013) (Table 2). The range is well within the range (13–93 g g$^{-1}$) reported based on the similar regression analysis between POC and Chl *a* concentration by Chang et al. (2003) for POM at 2 m water depth in the ECS, which is consistent with the 18–94 g g$^{-1}$ for phytoplankton carbon





to Chl *a* ratio, estimated from phytoplankton cell volumes by the same authors (Table 2). Although the Chl *a* concentration in our study was measured in sea water *in situ* by fluorometer attached with the CTD, it is more or less similar to Chl *a* concentrations obtained in the above-mentioned studies, which were mostly extracted from filtered particles by acetone with (Chang et al., 2003) or without acidification (Hung et al., 2013).

POC/Chl *a* ratio has been used for the discrimination of POM sources in coastal ocean waters (Cifuentes et al., 1988). POC/Chl *a* ratio in living phytoplankton varies with temperature, growth rate, day length, phytoplankton species, and irradiance (Savoye et al., 2003, and references therein). The POC/Chl *a* ratio of living phytoplankton was reported to be close to 40 g g$^{-1}$ (Montagnes et al. 1994), lower than 70 g g$^{-1}$ (Geider, 1987), lower than 100

g g$^{-1}$ (Head et al. 1996) or lower than 140 g g$^{-1}$ (Thompson et al. 1992). Furthermore, a POC/Chl *a* ratio of less than 200 g g$^{-1}$ is an indication of a predominance of newly produced phytoplankton (or autotrophic-dominated) in POM, and that a value higher than 200 is an indication of detrital or degraded organic matter (or heterotrophic/mixture-dominated) (Cifuentes et al., 1988; Savoye et al., 2003; Liénart et al., 2016, 2017). The POC/Chl *a* ratio in the DCM layer of East China Sea is almost <200 g g$^{-1}$ (26–200 g g$^{-1}$), with two exceptions

(DH5–2: 369 g g$^{-1}$; CON02: 617 g g$^{-1}$; Fig. 9), indicating that POM in DCM layers of ECS was dominated by phytoplankton, as also indicated by the low C/N ratios (4.1–6.3). The relatively high POC/Chl *a* ratios only in two stations, DH5–2 and CON02 (Fig. 9), suggest that POM in these two samples was likely from degraded phytoplankton OM, terrestrial OM, or heterotrophic-dominated OM. However, the molar C/N ratio of DH5–2 (5.6) and CON02 (5.3) are lower than the canonical Redfield ratio (6.63), eliminating the probability of degraded and

terrestrial OM sources. In addition, the insignificant linear correlation between C/N ratio and POC/Chl *a* ratio (Fig. 9) supports the non-degraded POM, a process resulting in a simultaneous increase of C/N and POC/Chl *a* ratios, mainly because of the preferential decomposition of N-enriched OM, as well as a fast degradation of Chl *a* than the bulk POC pool (e.g., Savoye et al., 2003). Thus, the POM in these two stations seems to be dominated by heterotrophic biota, though the exact reason for the dominance of heterotrophic biota only at two locations in our

study area is unknown and needs further investigation.

Briefly, several clues indicate the predominance of newly produced, phytoplankton-synthesized OM in the DCM layers of East China Sea: 1) less influence of fresh water and unrecognized content of terrestrial POM, 2) low molar C/N ratio, 3) a linear correlation between POC and chlorophyll *a*, and 4) low POC/Chl *a* ratio, mostly <200

g g$^{-1}$.

### 5.3 Dynamics of $\delta^{13}C_{POC}$ in POM in DCM




Although a narrow range of molar C/N ratio in SPM suggested an aquatic origin for POM at DCM layers, the wide variability of $\delta^{13}C_{POC}$ (−25.8 to −18.2 ‰) suggests that POM at DCM layers to be a mixture of terrestrial C3 plants with a typical $\delta^{13}C$ value of ca. −27 ‰ (e.g., Peters et al., 1978; Wada et al., 1987) and marine phytoplankton with a typical $\delta^{13}C$ range of −18 to −20 ‰ (e.g., Goericke and Fry, 1994). However, Fig. 5 illustrates a distinct

decreasing trend of $\delta^{13}C_{POC}$ towards the outer shelf; a pattern opposite to an increasing trend of $\delta^{13}C$ evident in suspended particles and surface sediments, i.e. seaward decrease of terrestrial OC in surface sediments of many river-dominated margins (Emerson and Hedges, 1988; Meyers, 1994; Hedges et al., 1997; Kao et al., 2003; Wu et al., 2003). Such a spatial distribution with more positive $\delta^{13}C_{POC}$ values in the coastal region, but more negative $\delta^{13}C_{POC}$ values in the middle-outer shelf is inconsistent with the idea of terrestrial OC influence. The

elevated $\delta^{13}C_{POC}$ values (average of −20.7 ‰) in the coastal region, concomitant with high POC concentrations (Fig. 4), is consistent with the higher marine primary productivity (11 g C m$^{-2}$ yr$^{-1}$) reported in the western than that in the eastern parts of East China Sea (Gong et al., 2003). The lower $\delta^{13}C_{POC}$ occurred in the middle-outer shelf region where oligotrophic Taiwan Warm Current Water and Kuroshio Water spread (Fig. 5). The lowest $\delta^{13}C_{POC}$ (−25.8 ‰) was observed at a water depth of 85 m, off  northeastern Taiwan, likely influenced by the

intrusion of Kuroshio Subsurface Water with low $\delta^{13}C$ from −31 ‰ to −27 ‰ (Wu et al., 2003), as also in agreement with the hydrographic parameters of this location (Figs. 3 and 7).

A positive linear correlation between POC and $\delta^{13}C_{POC}$ ($R^2 = 0.55$, p<0.0001; Fig. 10a), a characteristic feature of productive oceanic regions (Savoye et al., 2003), suggesting the effect of growing primary productivity (and or

increasing cell growth rate) on a decrease of carbon fractionation during photosynthesis. This is likely because of a limitation of dissolved $CO_2$, which cannot be compensated in time by the surrounding water in a relatively closed system because of stratification (Kopczyńska et al., 1995). Further, high productivity makes $^{13}C$-enriched OM in phytoplankton (Fry and Wainwright, 1991; Nakatsuka et al., 1992). Although primary productivity has a significant correlation with $\delta^{13}C_{POC}$, only 55 % of $\delta^{13}C_{POC}$ variation can be explained by primary productivity (Fig.

10a), implying that other factors, such as species and sizes of phytoplankton, must have influenced $\delta^{13}C$ values in phytoplankton living in the DCM layers.

In East China Sea, the distribution of phytoplankton community is affected by physicochemical properties (temperature, salinity and nutrients) of different water masses and surface currents (Umezawa et al., 2014; Jiang

et al., 2015). Diatoms and dinoflagellates are the main phytoplankton communities in summer with 136 taxa of diatoms from 55 genera and 67 taxa of dinoflagellates from 11 genera have been reported along with minor communities of chrysophyta, chlorophyta and cyanophyta (Guo et al., 2014b). There is a clear decreasing trend of phytoplankton abundances in the East China Sea from the surface to bottom, as well as from the coastal to offshore region, which is widely believed to be due to nutrient availability (Zheng et al., 2015). As for species, the




phytoplankton species have distinct spatial characteristics, but no significant species differences between surface waters and DCM (Zheng et al., 2015). Diatoms with large cell sizes were the dominant species in the coastal region, while phytoplankton with small sizes was dominant in the oligotrophic offshore shelf and Kuroshio waters (Furuya et al., 2003; Zhou et al., 2012). According to Jiang et al (2015), the contribution of micro- (>20 µm),

nano- (3–20 µm) and pico-phytoplankton (<3 µm) to Chl $a$, respectively, was 40 %, 46 % and 14 % in inshore nutrient-rich waters, and 14 %, 34 %, and 52 % in offshore regions in summer 2009. The outer shelf region was composed of small size phytoplankton, mainly cyanobacteria and cryptophytes that transported by Taiwan Warm Current and Kuroshio Current. It has been reported that diatoms have larger $\delta^{13}$C values (–19 to –15 ‰) than dinoflagellates (–22 to –20 ‰; Fry and Wainwright, 1991). Likewise, large size species have larger $\delta^{13}$C values

than small size species and heterotrophic dinoflagellates have large $\delta^{13}$C values than autotrophic dinoflagellates (Kopczyńska et al., 1995). Similarly wide variations of $\delta^{13}C_{POC}$ (–22.05 to –27.62 ‰) at DCM layers in the northern East China Sea were documented by Gao et al. (2014). Significant variations of $\delta^{13}$C in suspended OM in Delaware estuary (–25 to –20 ‰; Cifuentes et al., 1988) was reported, which was dominated by phytoplankton. These variations were influenced largely by isotopic fractionation during phytoplankton photosynthesis and

degradation than by changes in the relative contributions of terrestrial and aquatic OM (Fogel and Cifuentes, 1993).

The range and distribution of $\delta^{13}C_{POC}$ in newly produced POM has been comprehensively deciphered by variations in primary productivity (or biomass) and phytoplankton species compositions in East China Sea, as

discussed above. However, it remains unclear that is this newly produced POM in DCM made by phytoplankton in surface water that subsequently sank to DCM depths or made by $in$ $situ$ phytoplankton, which inhabit in the DCM layers?

The nutrient N/P ratio and a selective zooplankton grazing are mostly controlling the distribution of phytoplankton

community in the East China Sea (Guo et al., 2014). The zooplankton grazing is important for phytoplankton species abundance when a selective grazing on one specific species. Pilati and Wurtsbaugh (2003) demonstrated that zooplankton grazing coupled with nutrient transport is of importance for the persistence of DCM in a nutrient-depleted mountain lake. The DCM depths in the East China Sea were near the nitracline and an upward vertical transport of nutrients from the sea bottom can contribute significantly to the DCM layer and

maintain the chlorophyll maximum in the shelf region of ECS (Lee et al., 2016). Based on the nutrient flux into the ECS and Redfield ratio (C/N/P = 106:16:1; Redfield, 1958), one can calculate the nutrient-supported marine primary productivity. Since the limiting nutrient for phytoplankton growth was P in the Yangtze input-influenced inner shelf and N in the Kuroshio water-dominated outer shelf (Chen, 1996; Chen et al., 1996), we use C/P for calculating the Yangtze-sourced, nutrients-supported marine primary productivity, while C/N was used for





calculating the Kuroshio-driven nutrients-supported marine primary productivity. The results show that primary productivity derived from the Yangtze-sourced nutrients in the ECS is $1.7 \times 10^{12}$ g C yr$^{-1}$ (DIP: $1.36 \times 10^9$ mol yr$^{-1}$; Chai et al., 2009), while Kuroshio-driven nutrients are responsible for much higher primary productivity, i.e. $16.3 \times 10^{12}$ g C yr$^{-1}$ (DIN: $205 \times 10^9$ mol yr$^{-1}$; Chen, 1996). By using the *in situ* Chl *a* concentration, Gong et al. (2003)

estimated the average annual primary production for the ECS continental shelf in 1998 as 145 g C m$^{-2}$ yr$^{-1}$, equal to $72.5 \times 10^{12}$ g C yr$^{-1}$. This estimation is approximately four times higher than the Yangtze- and Kuroshio water-derived primary productivity achieved in this study, implying the potential role of recycled nutrients from OM remineralization for the primary productivity in the ECS.

If it is the case, then the *in situ* primary productivity in DCM layers should be mainly derived from remineralized nutrients of OM that associated with the surface sediment, as evident from the vertical distribution of turbidity (Fig. 2). This mechanism may result in a higher $\delta^{13}$C and C/N in surface sediments than in phytoplankton in DCM layers and thus POM due to the preferential remineralization of $^{12}$C and N-enriched OM. Consistently, a relatively constant, but higher $\delta^{13}$C values and C/N ratios ($\delta^{13}$C: –22.4 to –20.1 ‰ and C/N ratio: 6.4–10, unpublished data;

Selvaraj Kandasamy) observed in surface sediments collected during the same cruise and at same stations as SPM (Fig. 1), indicating that this mechanism is most likely.

A step forward, we use the POC concentration (μg L$^{-1}$), the surface area of the ECS continental shelf with water depths of <200 m ($0.5 \times 10^6$ km$^2$), and the difference of DCM depths between the inner shelf region (10 m) and

the outer shelf region (70 m) (Fig. 2), to estimate the inventory of suspended POC within the DCM layer. Two stations with bottom depths deeper than 200 m (DH7–8: 853 m, DH7–9: 800 m) in the Okinawa Trough were excluded from this estimation. The results show that the maximum (263 μg L$^{-1}$), minimum (37.3 μg L$^{-1}$) and mean concentration (92.4 μg L$^{-1}$) of POC within the DCM interval yields POC inventory of $7.9 \times 10^{12}$ g, $1.1 \times 10^{12}$ g and $2.8 \times 10^{12}$ g, respectively.

### 5.4 Dynamics of $\delta^{15}N_{PN}$ in POM in DCM layers

In contrast to POC and $\delta^{13}C_{POC}$ relationship (Fig. 10a), there is no significant relationship between PN and its isotopic composition ($\delta^{15}N_{PN}$) in SPM investigated at DCM layers (Fig. 10b), implying that primary productivity has

no significant control on $\delta^{15}N_{PN}$. As the OM in SPM at DCM layers was dominantly from newly produced, phytoplankton-synthesized source, $\delta^{15}N_{PN}$ is inferred to be similar as $\delta^{15}$N in phytoplankton. Considering the prevalence of low N/P ratio in the DCM layer in the East China Sea (Lee et al., 2016), the degree of nitrate utilization for phytoplankton should be high and would result in the composition of $\delta^{15}N_{PN}$ similar to $\delta^{15}$N of nitrate ($\delta^{15}N_{NO3^-}$) (Altabet and Francois, 1994; Minagawa et al., 2001). Therefore, the spatial distribution of $\delta^{15}N_{NO3^-}$ is



probably crucial to decipher the distribution of $\delta^{15}N_{PN}$ in DCM layers. The spatial distribution of $\delta^{15}N_{PN}$ (Fig. 5) resembles the surface current pattern (Fig. 1), as well as the distribution of different water masses (Fig. 7), suggesting that nitrate and the $\delta^{15}N_{NO3^-}$ of CDW, TWCW and Kuroshio Water are largely governing the distribution of $\delta^{15}N_{PN}$ in the study area.

According to Li et al. (2010), the range of $\delta^{15}N_{NO3^-}$ in the Yangtze River was 7.3–12.9 ‰, with a mean value of 8.3 ‰. In the northeast of Taiwan Island, $\delta^{15}N_{NO3^-}$ was 5.5–6.1 ‰ at depths of 500 m to 780 m (Liu et al., 1996). However, TWCW is nutrient-depleted, enabling incorporation of N-fixer derived nitrogen in suspended POM. This general spatial pattern of $\delta^{15}N_{NO3^-}$, i.e. higher $\delta^{15}N_{NO3^-}$ (>6 ‰) in the northeast coastal region and off northeast

Taiwan, but lower $\delta^{15}N_{PN}$ in between these two regions, exactly resembles the distribution of $\delta^{15}N_{PN}$ in the DCM layers of this study (Fig. 5). Therefore, $\delta^{15}N_{PN}$ in DCM layer in the East China Sea was primarily governed by the nutrient status and $\delta^{15}N_{NO3^-}$.

There is another possibility that high $\delta^{15}N_{PN}$ (6.7 ‰, 7.8 ‰) in DCM layers, off northeast Taiwan (Fig. 5), may not

be resulted from the high degree of nitrate utilization, but the incorporation of inorganic nitrogen in POM. According to Chen et al. (1996) and Liu et al. (1996), $NO_3^-$ and $NH_4^+$ concentrations in KSSW were high due to the decomposition of OM in sinking particles. However, the concentrations of Chl $a$ as well as POC and PN are low (Figs. 2 and 4). The low Chl $a$ might be limited by the low temperature in this high nutrient low chlorophyll region (Umezawa et al., 2014). Because of the low temperature, the prevailing high $CO_2$ pressure expected to

decrease $\delta^{13}C$ in DIC and also drive a great carbon isotopic fractionation during carbon assimilation by phytoplankton (Rau et al., 1992), a reason why $\delta^{13}C_{POC}$ values in these two stations are low (–25.8 ‰ and –25.2 ‰) compared to values of other locations in the ECS. Consistently, the low concentration of POC restricts that the high $\delta^{15}N_{PN}$ could not be from the denitrification effect. The high $\delta^{15}N_{PN}$ (6.7 ‰, 7.8 ‰) are probably due to the incorporation of inorganic nitrogen (mainly $NH_4^+$), the process can make the $\delta^{15}N_{PN}$ in POM as high as that

of inorganic nitrogen $\delta^{15}N$ (Coffin and Cifuentes, 1999). Although $\delta^{15}N$ of $NH_4^+$ in Kuroshio Water is not available, according to York et al. (2010), $\delta^{15}N$ of remineralized $NH_4^+$ was greater than $\delta^{15}N$ of $NO_3^-$. This possibility is also supported by the high concentrations of $NO_3^-$ and $NH_4^+$ in Kuroshio Subsurface Water (Liu et al., 1996), the low contents of POC (<1 %; 0.96 %, 0.98 %) and low molar C/N ratio (4.1, 5.4) in these two SPM samples.

**5.5 Impact of Yangtze River on POM in DCM of ECS**

Our elemental and isotopic results of POM indicated that influence of terrestrial OM in the DCM water depths of ECS is significantly less. The missing of terrestrial OM signals, especially transported by Yangtze River, may be because of reservoir and dam building in recent years that believed to shift the location of the Yangtze-derived



POC deposition from the ECS inner shelf to terrestrial reservoirs (Li et al., 2015). The sediment delivered from the river to the estuary has reduced by 40 % since 2003 when Three Gorges Dam (TGD) completed (Yang et al., 2011 and references therein). Recently, Dai et al. (2014) reported that the Yangtze suspended particles discharge declined to 150 Mt yr$^{-1}$, less than ~70% of its sediment delivery to the ECS during 1950s. Although 87 %

of the mean annual sediment of Yangtze River was discharged during the flood season from June to September (Wang et al., 2007; Zhu et al., 2011), among them, approximately 60 % of the fine-grained sediments are temporarily deposited near the estuary and later resuspended and transported southward along the inner shelf, off the China mainland coastline (Chen et al., 2017 and references therein). Yangtze-transported POM moves up toward the northeast across the shelf along the so called a Changjiang transport pathway in summer season (e.g.,

Gao et al., 2014), which is affected by the combining effects of high river discharge, southwest summer monsoon and the intensified TWC (Beardsley et al., 1985; Ichikawa and Beardsley, 2002; Lee and Chao, 2003). The *T–S* diagrams (Fig. 7) in this study also proved this view.

Accompanying with decreasing sediment input, dam building in the Yangtze River basin has buried around
4.9±1.9 Mt yr$^{-1}$ biospheric POC since 2003, approximately 10% of the world riverine POC burial flux to the oceans (Li et al., 2015). The POC input flux from the Yangtze into the ECS (range: 1.27–8.5 × 10$^{12}$ g C yr$^{-1}$; Wang et al., 1989; Qi et al., 2014) was significantly less than the estimated primary productivity (72.5 × 10$^{12}$ g C yr$^{-1}$; Gong et al., 2003), implying the predominance of marine-sourced organic matter in the East China Sea. Moreover, substantial quantities of organic substances that transported by Yangtze River may be completely
modified before being ultimately deposited onto the East China Sea inner shelf and being transported further offshore (Katoh et al., 2000; Lie et al., 2003; Chen et al., 2008; Isobe and Matsuno, 2008). Wu et al. (2007b) observed an advanced stage of POM degradation in the entire Yangtze River with little variations of an average degradation index of –1.1. The degradation index for protein amino acids was considered as an indicator for the degree of degradation, ranging from +1 for fresh OM to –1.5 for highly degraded OM (Dauwe and Middelburg,
1998). Wang et al. (2016), based on the investigation of lipids biomarkers, in a sediment core collected from the ECS (27.8° N, 122.2° E), suggested that marine autochthonous organic matter dominated the total OM (~90 %).

**Summary and conclusions**

In this study, we comprehensively characterized particulate organic matter (POM) straddling in the depth chlorophyll maximum layers of East China Sea using hydrographic parameters (temperature, salinity and turbidity), fluorescence (chlorophyll) as well as elemental (POC, PN) and stable carbon and nitrogen isotopic (δ$^{13}$C$_{POC}$ and δ$^{15}$N$_{PN}$) values. All these parameters indicated that POM is dominantly composed of *in situ* (phytoplankton) newly produced OM with less significant contribution from terrestrial input, although the study




area is one of the best examples of river-dominated, shallow continental margins in the world. The $\delta^{13}C$ in phytoplankton contained relatively low values than that of typical marine phytoplankton (−18 to −20 ‰). Both $\delta^{13}C_{POC}$ and $\delta^{15}N_{PN}$ show large variations, which are due to the variations in primary productivity and phytoplankton species compositions for $\delta^{13}C_{POC}$, and uptake of nitrate diffused through the thermocline or locally

regenerated ammonia for $\delta^{15}N_{PN}$, respectively.

This study emphasizes the sufficient investigation of prerequisite endmember variability and provides a relatively accurate $\delta^{13}C_{POC}$ and $\delta^{15}N_{PN}$ values of marine endmember in the East China Sea, which is crucial for the estimation of relative contributions of terrestrial and marine OM by endmember mixing model. Therefore, our

results with highly variable $\delta^{13}C_{POC}$ and $\delta^{15}N_{PN}$ values in autotrophic-dominated deep chlorophyll maximum layers can provide an unique range of these two isotopes in the East China Sea, especially the region of South of 29 °N, and form a basis for the long-term evaluation of organic carbon burial along the inner shelf mud-belt which is largely accumulated during the Holocene in the East China Sea.

**Acknowledgments.** We thank the captain, chief scientist and crew of the *Science 3* Cruise for their support and assistance during the sampling in summer 2013. We also appreciate Wenbin Zou and Xinlei Jiang for their help with onboard sampling. SK is grateful to the National Natural Foundation of China (41273083), Shanhai Fund of Xiamen University (2013SH012) and Open Funds of First Institute of Oceanography (0050-K2015003, 0050-K2016008) for the financial support.

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



**Table 1.** Summary statistics of elemental and isotopic compositions of suspended particulate matter (SPM) at around DCM layers in the continental shelf region of East China Sea (n=36).

|  | Sampling depth | SPM | $\delta^{13}C_{POC}$ | $\delta^{15}N_{PN}$ | POC | PN | POC | PN | C/N |
|---|---|---|---|---|---|---|---|---|---|
|  | m | mg L$^{-1}$ | ‰ | ‰ | µg L$^{-1}$ | µg L$^{-1}$ | % | % | molar |
| Min | 10 | 1.7 | −25.8 | 3.8 | 20.4 | 4.5 | 0.90 | 0.19 | 4.1 |
| Max | 130 | 14.7 | −18.2 | 7.8 | 263.0 | 52.8 | 4.74 | 0.95 | 6.3 |
| Mean | 45 | 4.4 | −23.1 | 6.1 | 85.5 | 17.8 | 2.17 | 0.45 | 5.6 |
| SD | 21 | 2.7 | 1.45 | 1.0 | 49.5 | 9.94 | 0.94 | 0.18 | 0.5 |



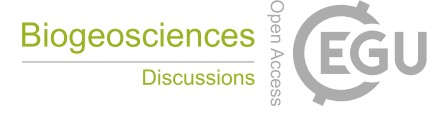

**Table 2.** POC/Chl *a* in the Western Pacific compiled from the literature.

| Region | Depth (m) | POC/Chl *a* (g g⁻¹) | Method | Note | References |
|---|---|---|---|---|---|
| ECS | DCM | 35.3 | POC vs. Chl *a* regression | $R^2 = 0.49$, $p<0.0001$ | This study |
| ECS | 2 – 150 | 64 | POC vs. Chl *a* regression | $R^2 = 0.61$, $p<0.001$ | Hung et al., 2013 |
| Coastal of ECS | 2 / 5 | 13 | POC vs. Chl *a* regression | $R^2 = 0.43$, $p<0.05$ | Chang et al., 2003 |
| Middle shelf and Kuroshio zone of ECS | 2 | 93 | POC vs. Chl *a* regression | $R^2 = 0.65$, $p<0.001$ | Chang et al., 2003 |
| Coastal zone of ECS | 5 | 18 | Cell volume | | Chang et al., 2003 |
| Middle shelf of ECS | 5 | 67 | Cell volume | | Chang et al., 2003 |
| Kuroshio zone of ECS | 5 | 94 | Cell volume | | Chang et al., 2003 |
| Western Pacific | DCM | 52 | Cell volume | | Furuya, 1990 |




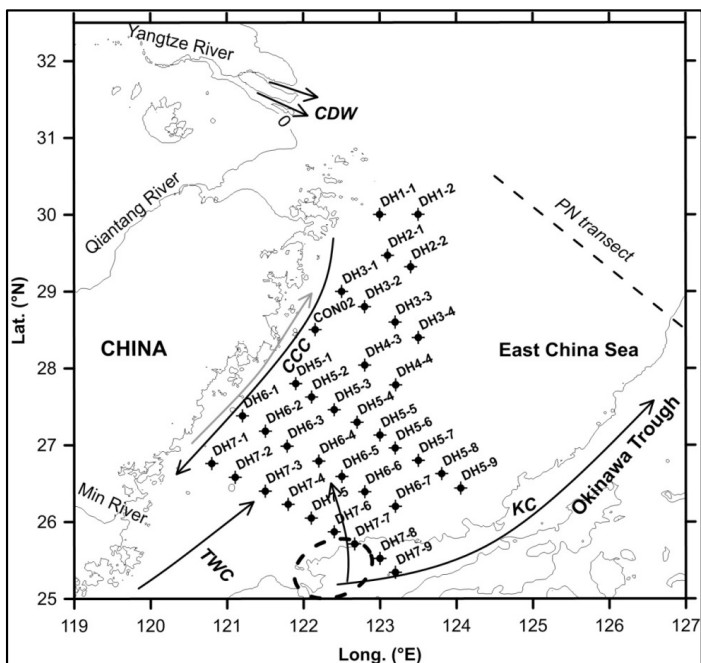

**Figure 1.** Map showing the locations of suspended particulate matters (SPM) collected in summer (June 22–July 21) 2013 for the present investigation along with modern current patterns in the East China Sea. Also shown are station names along with locations. CDW – Changjiang Diluted Water, CCC – China Coast Current, TWC – Taiwan Warm Current and KC – Kuroshio Current. The dashed ellipse represents the center of Kuroshio upwelling due to an abrupt change in the bottom topography in the northeast of Taiwan Island (Wong et al., 2000). Also shown is the PN transect, a cross shelf transect that is relatively well studied for particulate organic matter dynamics in the East China Sea.





**Figure 2.** Vertical distributions of chlorophyll *a* and turbidity along seven transects in the East China Sea in summer 2013.



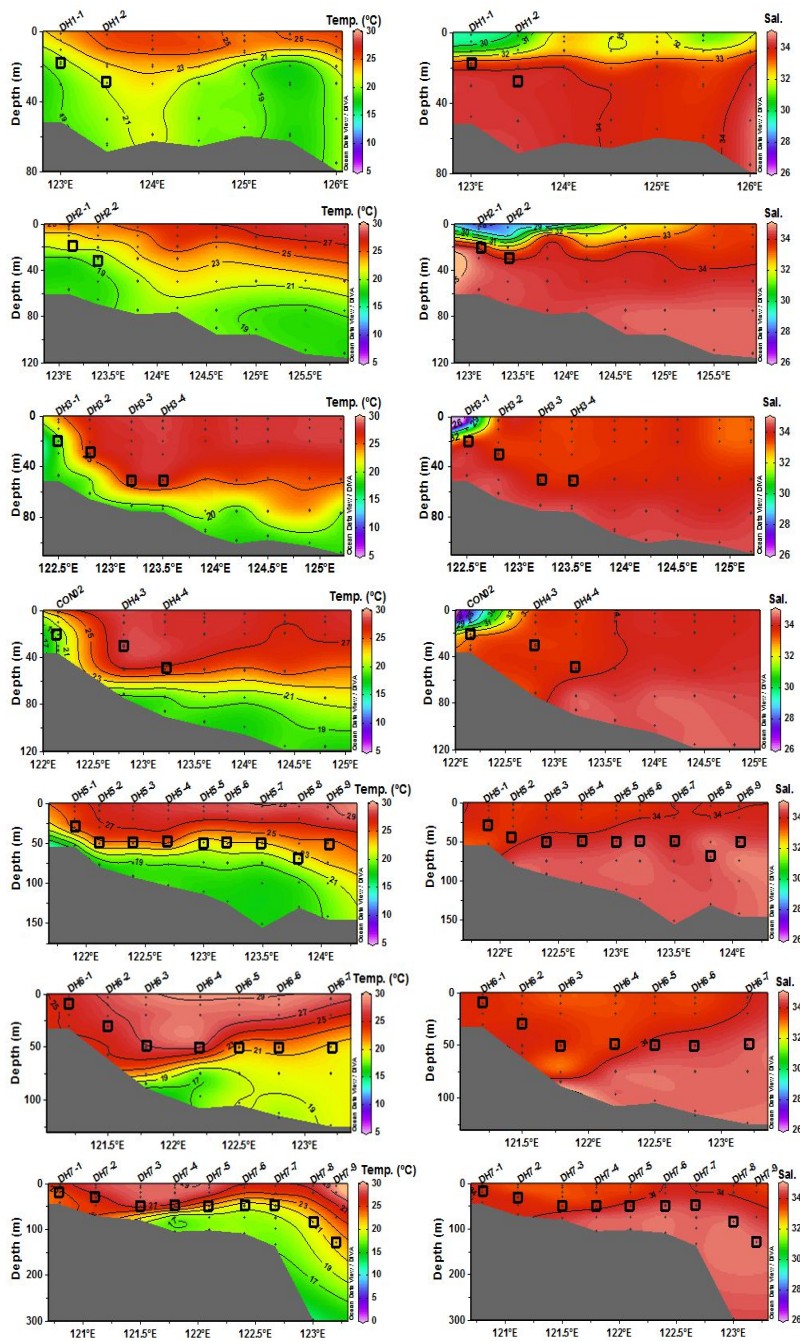

**Figure 3.** Vertical distributions of temperature and salinity along seven transects in the East China Sea in summer 2013. Note that there is an obvious thermally-stratified water column during the collection of suspended particulate matters in the East China Sea.





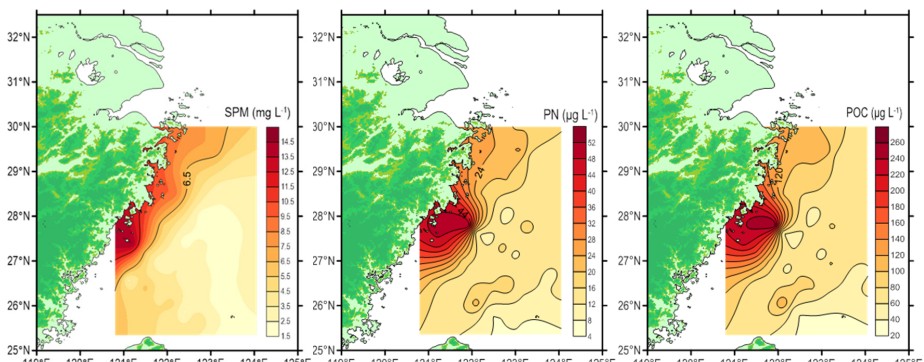

**Figure 4.** Spatial distributions of suspended particulate matter (SPM, mg L⁻¹), particulate organic carbon (POC, µg L⁻¹) and nitrogen (PN, µg L⁻¹) at around the deep chlorophyll maximum layer in the East China Sea in summer 2013.





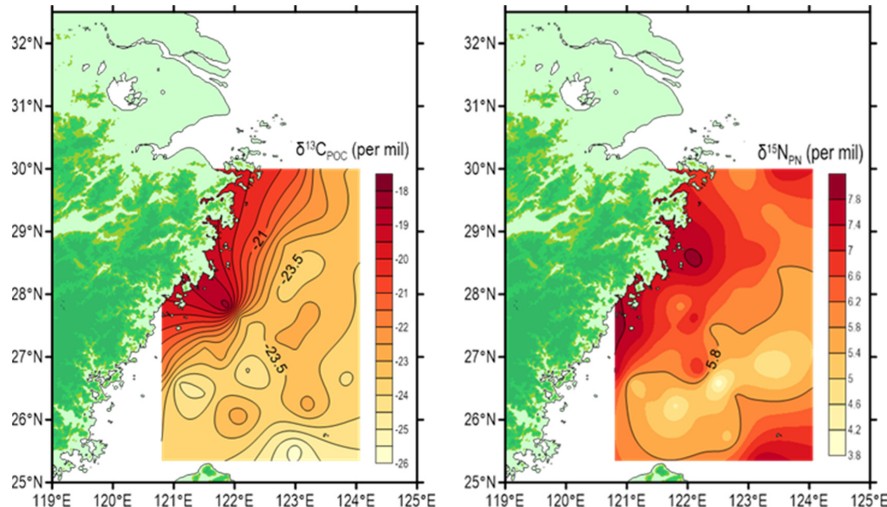

**Figure 5.** Spatial distributions of stable isotopic values of particulate organic carbon and nitrogen ($\delta^{13}C_{POC}$ and $\delta^{15}N_{PN}$) in suspended particulate matters at around the deep chlorophyll maximum layers in the East China Sea in summer 2013.





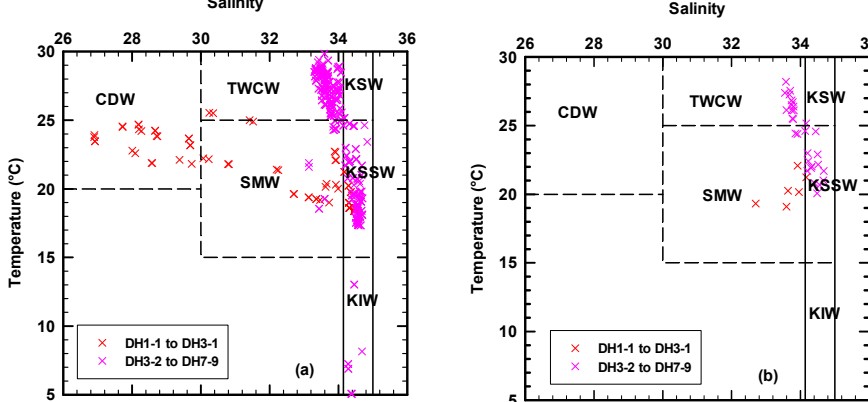

**Figure 6.** Temperature–Salinity (*T–S*) diagrams for (a) the entire water column in the East China Sea and (b) the deep chlorophyll maximum layers where from the suspended particulate matters collected for the present investigation. *T–S* ranges of seven water masses are taken from Umezawa *et al.* (2014). TWCW – Taiwan Warm Current Water; SMW – Shelf Mixed Water; KSW – Kuroshio Surface Water; KSSW – Kuroshio Subsurface Water; KIW – Kuroshio Intermediate Water.





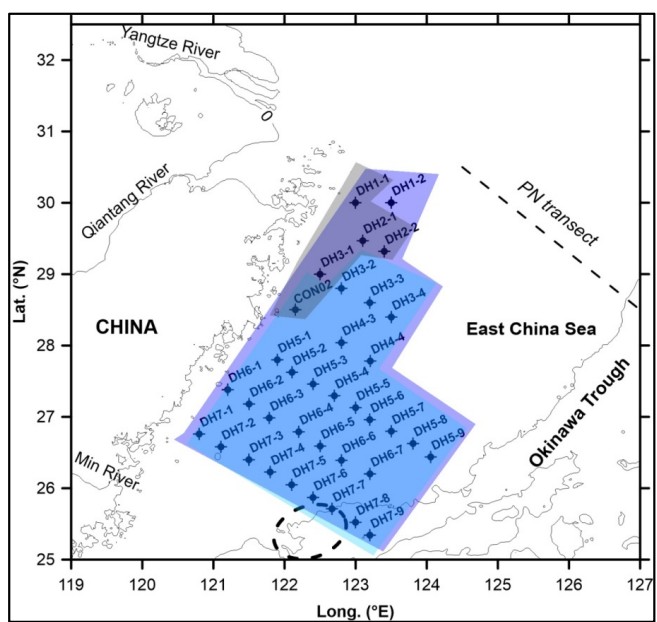

**Figure 7.** Diagram delineating regions influenced by three main water masses based on the *T–S* (Figs. 3 and 6) in the study area. Area with black polygon represents the CDW influence, which is limited only in the upper 10 m, area with sky blue represents the TWCW dominance, which influenced ~30 m below the surface, and the polygon colored by deep blue represents the area influenced by KSSW, which influenced the bottom water of the entire study area. Also shown is the PN transect, a cross shelf transect that is relatively well studied for POM dynamics in the East China Sea.



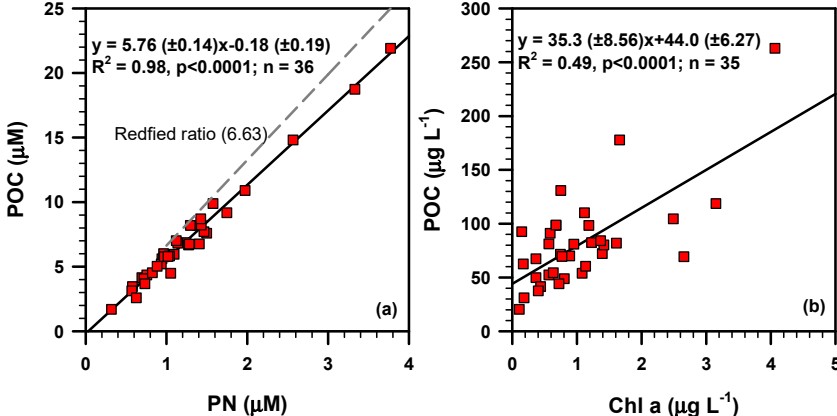

**Figure 8.** Bi-plots showing the relationships of (a) PN vs. POC and (b) Chl *a* vs. POC in suspended particulate matters investigated in this study. Redfield ratio (dashed line in panel a) is taken from Redfiled (1958).



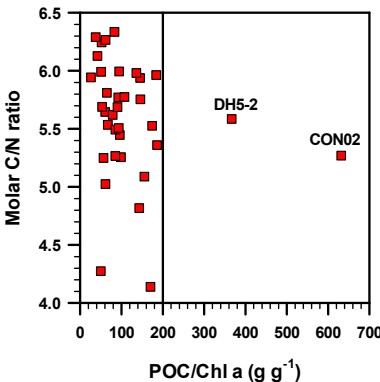

**Figure 9.** POC/Chl *a* vs. molar C/N ratio in suspended particulates investigated in this study. The vertical line represents POC/Chl a ratio of 200 g g$^{-1}$, which is the upper limit of this ratio for phytoplankton-dominated particulate organic matter (Savoye et al., 2003). See text for more details. CON02 is the station where red tide was observed during the sampling time and the color of the surface water was brown and dissolved oxygen in the bottom water was 1.6 mg L$^{-1}$.



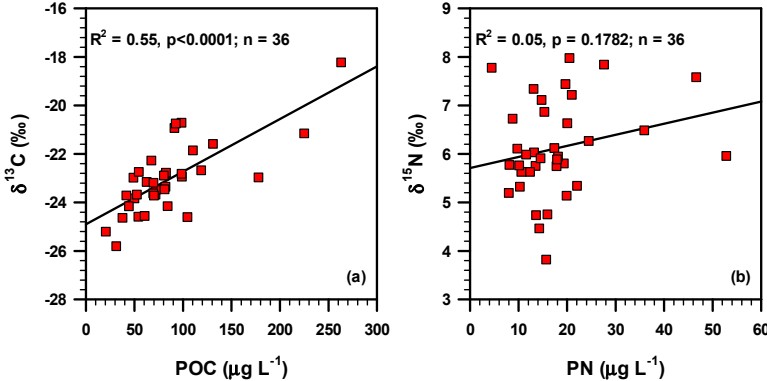

**Figure 10.** Bi-plots showing the relationships of (a) POC vs. $\delta^{13}C_{POC}$ and (b) PN vs. $\delta^{15}N_{PN}$ in suspended particulate matters from the deep chlorophyll maximum layers in the East China Sea.