# Peer review of "Biogeochemical characteristics of suspended particulates at deep chlorophyll maximum layers in the southern East China Sea"

_Biogeosciences, 2017_

## Referee Comment (RC1) · P.A. Meyers (Referee) · 3 Aug 2017

Liu and colleagues present the results of their study of the organic carbon and nitrogen contents in suspended particles collected around deep chlorophyll maximum layers in the East China Sea. They measured carbon and nitrogen concentrations and isotopic compositions of 36 samples collected from 7 cross-shelf transects and augmented these data with a suite of standard hydrographic measurements. These data allowed them to conclude that little land-derived organic matter contributes to the suspended particulate matter despite the proximity of the sampling locations to the mouth of the Yangtze River. Instead, the organic matter freshly produced by phytoplankton. The

authors attribute variations that they found in the carbon and nitrogen isotopic values of the organic matter to local differences in productivity rates and differences in the nitrate isotopic signatures of the major water masses in the area. The study seems to have been designed well, and the authors seem to have interpreted their results properly, but problems with the presentation make this contribution hard to read and appreciate. The English badly needs refining, and some additional details should be addressed.

For a start, the second paragraph of the Introduction seems to be missing something (lines 27-32. After that, Section 3.1 on sample collection should include a tabulation of the 36 samples that shows their water depths and some of their hydrographic properties. This tabulation could be an appendix, but the information that it would contain should be available to interested readers. Then, the explanation for higher d13C and C/N values in surface sediments that phytoplankton (page 14, lines 10-16) seems out of place. This contribution is about POM, not sediments.

To continue with details that need correction, neither Table 1 nor Table 2 contribute much to the paper as they exist. I suggest either expanding Table 1 as suggested above or deleting it and providing a detailed appendix. The figures are effective, but Figure 3 could be improved by inverting the salinity color code so that salinity (and hence density) increases downward and Figure 8 needs to have the spelling of Redfield corrected the left panel and in the legend.

Phil Meyers August 3, 2017

---

## Referee Comment (RC2) · Anonymous Referee #2 · 8 Aug 2017

General overview

The manuscript of Liu et al. is focussed on characteristics of suspended particulate organic matter (SPOM) in the deep chlorophyll maximum (DCM) of the Eastern China Sea (ECS) during summer 2013. It is based on bulk descriptors of SPOM (C/N and POC/Chl a ratios as well as $\delta$13C and $\delta$15N). The mains findings are: 1) DCM SPOM mainly originates from in situ primary production, 2) terrestrial POM slightly or insignificantly contributes to DCM SPOM composition, and 3) the latter is contradictory to previous studies but illustrates the drastic decrease in the contribution of the terrestrial POM originating from the Yangtze River to the SPOM composition in the ECS. These

findings are sounded and clearly illustrated by the present data set.

The manuscript is well organized and usually well illustrated.

It is of broad audience for scientist who are interested in organic matter cycling and land-to-sea export. It is within the scope of BG.

However there are some issues in the present version of the manuscript that preclude the acceptance of the manuscript in its present version. These issues are:

- a lack of information in the methods

- many unneeded details and miscellaneous information that are not needed in the discussion, that rend the discussion too wordy and that dilute the main messages of the study. Authors should focus on what the data indicate, which is usually very clear.

- interpretations of $\delta$15N data set that are not correct or at least very partial. This data set cannot be published within this manuscript without deeply reconsidering its interpretation and without additional data set regarding N-nutrient (at least nitrate and ammonium) concentrations.

There are also some inconsistencies and language errors that have to be corrected.

Thus, I recommend major revision.

Detailed comments

Section 3: lack of information and details needed

Conversion of fluorescence into chlorophyll a concentration: Since Chl a is a key parameter of the study and is used to calculate POC/Chl a ratio (which values are compared to reference values), it should be explained how in situ fluorescence was converted into Chl a concentration.

Section 3.1, line 9: indicate the range of depths of the samplings for SPM.

Section 3.1: indicate how the filters were rinsed right after the filtration

Section 3.2, line 28: detail how the filters were treated with HCl 1N

Section 3.2, line 30: indicate the diameter of the punches

Section 3.2, lines 30-31: it looks like $\delta$15N and PN were analyzed on the decarbonated part of the filter. Why not on the un-decarbonated part of the filter? There is always chance to bias $\delta$15N and PN using decarbonated material for these measurements (e.g. Lorrain et al (2003) and other references). Also, it looks like there was a very small part of the filters that were analysed for C and N elemental and isotopic composition. What quantities of C and N were analysed?

Influence of CDW at DCM depth in the ECS

It cannot be stated that the influence of CDW in the study site is nil or insignificant. The low salinity measured at some of the sites (Fig. 3) clearly indicates the influence of CDW. It is mainly the case in surface water but also the case at some of the DCM depths where water was sampled for SPM (stations DH1-1, DH2-1, DH2-2, DH3-1, CON02). This is also clear from Fig. 6b where five stations falls within the SMW square, SMW being clearly a water body composed of a mixing between KSSW and CDW.

It should better be written that the influence of CDW in the study site is low (see some of the 'minor points' below) or weak (as written in section 5.1, P8, line 34).

End of section 5.1 (P9, lines 1-9) and Fig. 7

Only the DCM depths (= the depths of interest for the present study) should be considered for delineate the polygons of Fig. 7. Was it the case? For similar reason, I think that the sentence "Interestingly...study area (Fig. 7)" should be deleted or reworded without citing Fig. 7 (but rather Fig. 3?) since it is quite confusing. Another option may be to not cite depth limitation of the water masses influences but only describe Fig. 7.

The last paragraph of section 5.1 is also quite confusing. Reword it as: "In summary, although the river runoff was huge, the influence of CDW plume in the southern part

of the ECS was insignificant during summer 2013, mainly because most of the CDW plume was transported to northeastward of the Yangtze estuary to the Korean coast (Isobe et al., 2004; Bai et al., 2014; Gao et al., 2014). This contrasts with summer 2003 when the plume front moved southward (Bai et al., 2014). Meanwhile, the intrusion of TWCW and KSSW was strong in the continental shelf of the East China Sea during summer 2013."

Section 5.2.1

I fully agree the main conclusions of this section and most of data interpretations (especially the first and last paragraphs).

However the second paragraph adds detailed discussion with literature comparison that is not needed (especially when dealing with zooplankton and Trichodesmium) for the present study. Authors should better goes directly to the conclusion (i.e. the last paragraph) without diluting the main conclusions with unneeded wording. Thus, the second paragraph should be deleted.

Section 5.2.2

As for the previous section, I fully agree the main conclusions and most of the data interpretation, but this section is too wordy and gives too many details (especially too many values from the literature). Authors should better focus on the main information and the main conclusions.

Thus, I suggest the following:

- P10, lines 26-34: one-two sentence(s) should be enough

- P11, lines 2-4: keep this sentence but rephrase the last line as "filtered particles (Chang et al., 2003; Hung et al., 2013)"

- P11, lines 8-10: do not report all these values

Regarding the high POC/Chl a ratio, did authors check if these high values were rather

due to very low Chla concentration or high POC concentration? If the former, these high values may be associated to high uncertainty on the Chl a estimation when values are low. If the latter (high POC concentration associated to Chl a concentration similar to surrounding stations), this may be effectively due to heterotrophic biomass.

Section 5.3: first three paragraphs

I fully agree the main conclusions and the data interpretations of the first three paragraph of this section.

I suggest authors to have a look at Lowe et al. (2014) and Miller et al (2013): these articles are of interest for the present section.

Page 12, Lines 18-26: two other processes may influence phytoplankton $\delta$13C: temperature and degradation. This is discussed in Savoye et al. (2003) that authors cite in many occurences. Authors may have a look at biplots like $\delta$13C vs temperature and versus POC/Chl a and C/N (considering these ratios may also indicate phytoplankton decay). They also may check the normalization of $\delta$13C by temperature (as in Savoye et al., 2003) before plotting normalized $\delta$13C versus POC, since temperature usually have (indirect) influence on phytoplankton $\delta$13C.

Section 5.3: last three paragraphs

I do not think that the last three paragraph of the section are needed. The objective of the two paragraphs before the last (from "The nutrient N/P ratio" to "this mechanism is most likely") is to decipher whether POM sampled in the DCM came from in situ production or from surface production (cf. the fourth paragraph of the section). In fact, these two paragraphs lay on very putative argumentation and do not allow (and are not convincing in) deciphering between the two hypotheses. These hypotheses have already been discussed in sections 5.2.1 and 5.2.2 with sufficient argumentation for considering that POM mainly came from in situ production. To my point of view, these two paragraphs of section 5.3 are not needed in this section neither in the manuscript.

The last paragraph of the section is a tentative of inventory of POC in the DCM layer. The estimation is very rough, is associated to large uncertainty, and the calculation is not convincing. It is also completely disconnected from the rest of the section, which is focused on $\delta$13C dynamics (see the title of the section). Again, this paragraph is not needed in this section neither in the manuscript. Thus, the last three paragraphs of the section should simply be deleted and the fourth paragraph of the section ("The range... DCM layers?") should be replaced with a brief conclusion of the first three paragraphs of the section.

Section 5.3 would better stand with the first three paragraphs and a conclusion without the unclear and unconvincing last three paragraphs.

Section 5.4

This section is the less clear and the less convincing of the manuscript. The main conclusion (POM $\delta$15N distribution is primarily governed by the nutrient status and $\delta$15N of nitrate) is mainly guess. One of the main issues of the section is the lack of nutrient data. This rend the data interpretation mainly guess-work. Another issue is that authors mainly take into account nitrate as a nutrient for phytoplankton. Ammonium appears only in the last paragraph. The other species of N-nutrient (N2, dissolved organic nitrogen as urea) are not mentioned. However, it is reported that "Kuroshio Water and TWCW induced Trichodesmium" (P10, line 7). Thus, PN $\delta$15N values should also be discussed considering N2-fixers (diazotrophs). At last, many sentences are not clear. This gives the impression that authors do not fully have in mind what processes drives PN and phytoplankton $\delta$15N.

Thus, this section should be deeply reworked including deep data re-interpretation. To me, such section dedicated to PN/phytoplankton $\delta$15N cannot stand without data of nutrient concentration originating from the same cruise. If these data are not available, PN/phytoplankton $\delta$15N cannot be discussed. If it would be the case, $\delta$15N data should be removed from the manuscript.

Section 5.5

This section gives an important conclusion: the influence of terrestrial POM (mainly originating from the Yangtze River) has drastically decreased in the ECS. This section is mainly based on literature data and conclusions. These inputs from the literature are of interest, but the section should also compare data from the present study with previous data. Thus, this section should start with the comparison of POC/Chl a and C/N ratios, and $\delta$13C values between the present study and previous studies. Then, the decrease of terrestrial POC fluxes and deposition can be cited (literature data). Last lines of the section: there are again unneeded details in these lines. Avoid describing the degradation index but directly give the conclusions of Wu et al (2007b).

English language

The language is usually quite understandable, but there are many errors or mistakes. Part of them is listed in the 'minor points' below. Nevertheless, the whole manuscript should be deeply checked for English language.

Inconsistencies

There are some inconsistencies between values that are cited in the text and values reported in the tables (see 'minor points' below). Please, check the consistency between all the values reported in the text and tables.

Abstract, Introduction and Summary and conclusions

Sections 'Abstract' and 'Summary and conclusions' should partly be re-written taking into account the above detailed comments.

The third objective that appear in section Introduction should be removed (see one of my comments dedicated to section 5.3 — three last paragraphs — above).

Minor points

- P1, Line 20: what do you mean with 'straddling'?

- P2, line 9 and in the whole manuscript: replace 'endmember' with 'end-member'

- P3, line 8: remove 'which in turn, the elemental and isotopic compositions of marine productivity' since it is not correct

- P5, line 1: depending what you want to say, add 'by' or 'to' between 'decreased' and '86%'

- P5, line 17: replace 'had' with 'have'

- Last sentence of section 3.1: place this sentence in section 3.2 since it is not sample collection. Rename section 3.2 as 'Determination of SPM concentration and analysis of POC, PN, $\delta$13C and $\delta$15N

- P5, line 30: replace 'with' with 'and placed in'

- Section 3.2, P5-6, sentence "organic carbon and nitrogen ... entering the IRMS": remove the sentence since such level of detail is not needed

- Section 3.2: remove the three last sentences ("Conventional...Sigman et al., 2009" since the first one is unneeded detail and the two last ones do not stand in a section dedicated to methods.

- P6, line 22: add 'usually' between 'profiles of Chl a' and 'show'.

- Section 4.1.3: since Fig. 3 illustrates at maximum the first 300m of the water column and since the sampling depth was within this depth interval, please do not describe deeper water, either the reading is quite disturbing. Thus, the temperature ranged between 30 and ca. 15°C.

- P7, line 16: replace 'increasing' with 'increases'; reword "with the high temperature (>30°C) spreads widely".

- P7, line 29: replace 'insignificant' with 'low'.

- P8, line 4: '4.4' or '4.5' as reported in table 1?

- P8, line 5: '17.7' or '17.8'?

- P8, lines 7-8: please also indicate where the highest POC and PN concentration were located.

- P8, line 17: '8.0' or '7.8' as reported in Table 1?

- P8, lines 17-18: please also indicate where the highest $\delta 13C$ values were located.

- P8, line 21: Fig. 10 is cited before Fig. 5. Check the numbering of the figures.

- P8, line 33: SMW is a water body that is composed of a mixture between two other water masses (CDW and KSSW; Fig. 6). So, do not consider SMW as a water mass and remove it from this list.

- P10, line 25: remove the word 'moderate' since this information is not useful here.

- P10, line 29: '48' or '52' as reported in Table 2?

- P11, line 28: replace 'less' with 'low'; delete "and unrecognized content of terrestrial POM".

- P12, 2: replace 'to be' with 'would be'.

- P12, line 8: replace 'more positive' with 'less negative'.

- P12, line 34: delete 'As for species,'.

- P13, line 7: delete 'that'.

- P13, line 8, 9 and 10: replace 'larger' with 'higher'

- P13, lines 9 and 10: replace 'size species' with 'phytoplankton'

- P15, line 33: replace 'significantly less' with 'low'.

- P16, line 12: replace 'proved' with 'illustrate'.

- Table 1: add POC/Chl a values in this table; indicate in the caption what means 'SD'.

- Figure 1: indicate KSSW on the figure; be consistent with Fig. 6. Indicate on this figure the location of the stations that appear on Fig. 2 and 3 but were not sampled for SPM in the DCM.

- Figure 6: the two colours are not distinguishable. Choose other colours. Remove 'from' in the second line of the caption. Add 'were' after 'matters' in the third line of the caption.

- Figure 7: replace 'black' with 'grey' in the second line of the caption.

- Figure 8, first line of the caption: it is POC vs. PN and POC vs. Chl a.

Additional references

Lorrain A., N. Savoye, L. Chauvaud, Y-M. Paulet and N. Naulet, 2003. Decarbonation and preservation method for the analysis of organic C and N contents and stable isotope ratios of low-carbonated suspended particulate materiel. Analytica Chimica Acta, 491, 125-133.

Lowe, A. T., A. W. E. Galloway, J. S. Yeung, M. N. Dethier, and D. O. Duggins. 2014. Broad sampling and diverse biomarkers allow characterization of nearshore particulate organic matter. Oikos. 123: 1341–1354, doi:10.1111/oik.01392

Miller, R. J., H. M. Page, and M. A. Brzezinski. 2013. $\delta$13C and $\delta$15N of particulate organic matter in the Santa Barbara Channel: drivers and implications for trophic inference. Mar. Ecol. Prog. Ser. 474:53-66, doi:10.3354/meps10098

——————————————

---

## Referee Comment (RC3) · Anonymous Referee #3 · 28 Aug 2017

Review note

Manuscript number: Biogeosciences Discuss., https://doi.org/10.5194/bg-2017-290

General Comment: This manuscript characterized the bulk and isotopic composition of organic matter collected in DCM layer of the south East China Sea in summer time. The study is well designed and neatly presented. It observed the marine derived material is the dominant organic matter in DCM layer. The influence of the Yangtze River is very limited. Additionally, the nitrogen isotopes elucidated the potential role of N2 fixing in middle shelf where TWCW is dominated and remineralized nutrients plays an important role. However, in this point, the depth profile will be very helpful to strengthen

the discussion but it is lack in the manuscript. Furthermore, the influence of lateral transport (cross shelf) better to be considered when the authors estimate the nutrients contributions from different sources and POC inventory.

Specific comment: 1) The title better be more specific in study region, such as southern East China Sea 2) Abstract: OK 3) Introduction: Better to emphasize the status of DCM in ECS and the potential role in POC inventory estimation and hypothesis 4) Methodsïïjʐ̌P5, the filtration volume was in the range of 0.5-2L, the author used half filter for POC or PN analysis, Did they have enough material for reliable analysis, especially for nitrogen? 5) Result and interpretations: The order of hydrographic characteristics can be adjusted as salinity, turbidity and chla 6) Discussion: In general, the authors gave proper credit to related work and clearly indicate their own new contribution to the biogeochemical cycles in the study area. Some minor suggestions: a. P13-14, How to use C/P ratio to estimate the Yangtze-sourced nutrients for marine primary productivity, how does the lateral cross shelf transport contribute to the POC inventory? b. P15, L20, this paragraph is a bit speculative and need more straightforward data to support itself, the depth profile data could be bit helpful to elucidate. c. Comments on 5.5, why the Yangtze River will play an important role in DCM OM in south ECS, the paragraph of TGD can be moved which seems less related to this topic, also I am confused about how the author summarized in the abstract" SPM investigated here seems not to be influenced by the terrestrial organic matter supplied by the Yangtze River (Changjiang) in summer 2013, a finding that is contrary to a number of previous studies' conclusion.", which is not convinced in this part. d. The quality of Figure 4 and 5 should be improved. Figure 8b, there is 35 samples summarized but all the other plotted based on 36 samples, why?

---

## Author Comment (AC1) · 2 Oct 2017

Referee 1 (Prof. P.A. Meyers)

Referee 1: Liu and colleagues present the results of their study of the organic carbon and nitrogen contents in suspended particles collected around deep chlorophyll maximum layers in the East China Sea. They measured carbon and nitrogen concentrations and isotopic compositions of 36 samples collected from 7 cross-shelf transects and augmented these data with a suite of standard hydrographic measurements. These data allowed them to conclude that little land-derived organic matter contributes to the suspended particulate matter despite the proximity of the sampling locations to

the mouth of the Yangtze River. Instead, the organic matter freshly produced by phytoplankton. The authors attribute variations that they found in the carbon and nitrogen isotopic values of the organic matter to local differences in productivity rates and differences in the nitrate isotopic signatures of the major water masses in the area. The study seems to have been designed well, and the authors seem to have interpreted their results properly, but problems with the presentation make this contribution hard to read and appreciate. The English badly needs refining, and some additional details should be addressed.

Reply: Thank you very much for your appreciation on the overall performance of the research work that presented in our manuscript. We took the utmost care to refine our English in the revised version.

Referee 1: For a start, the second paragraph of the Introduction seems to be missing something (lines 27-32).

Reply: It seems that there is no connection between the last two sentences of the first paragraph and the first sentence of the second paragraph and therefore this part has been revised as follows:

It is known that stable isotopes ($\delta$13C, $\delta$15N) and molar C/N ratios of POM in estuarine and marine areas are representative of these values in primary production-derived OM and in that they are largely synthesized by phytoplankton (Gearing et al., 1984). Since phytoplankton is the main primary producer of marine OM, it should therefore be considered when studying the dynamics of POM in the marine water column.

The chlorophyll a (Chl a) in the sea water is often used as an index of phytoplankton biomass. The deep chlorophyll maximum (DCM) layer, which contributes significantly to the total biomass and primary production in the whole water column (Weston et al., 2005; Hanson et al., 2007; Sullivan et al., 2010), is approximately equal to the subsurface biomass maximum layer (e.g., Sharples et al., 2001; Ryabov et al., 2010).
Referee 1: After that, Section 3.1 on sample collection should include a tabulation of the 36 samples that shows their water depths and some of their hydrographic properties. This tabulation could be an appendix, but the information that it would contain should be available to interested readers.

Reply: Table S1 is included in the appendix in the revised version. Table S1 includes depth, chlorophyll fluorescence concentration, temperature, salinity and turbidity at the SPM sampling layer, as well as, depth and chlorophyll fluorescence concentration in the deep chlorophyll maximum layer.

Referee 1: Then, the explanation for higher $\delta$13C and C/N values in surface sediments that phytoplankton (page 14, lines 10-16) seems out of place. This contribution is about POM, not sediments.

Reply: We agree with the comment from Referees 1 and 2, and thus this part has been deleted in the revision.

Referee 1: To continue with details that need correction, neither Table 1 nor Table 2 contribute much to the paper as they exist. I suggest either expanding Table 1 as suggested above or deleting it and providing a detailed appendix. The figures are effective, but Figure 3 could be improved by inverting the salinity color code so that salinity (and hence density) increases downward and Figure 8 needs to have the spelling of Redfield corrected the left panel and in the legend.

Reply: Table 1 is expanded as suggested above and Table S1 with the hydrographic data is also included in the appendix.

As suggested, Table 2 has been deleted in the revised revision. Figure 3 has been improved, as suggested; however, we don't know how to invert the salinity colour code using Ocean Data View. The spelling of Redfield has been corrected in Figure 8.

Thank you very much.

Please also note the supplement to this comment:
https://www.biogeosciences-discuss.net/bg-2017-290/bg-2017-290-AC1-
supplement.pdf

―――――――――――――――――

[Figure]

**Figure 3.** Vertical distributions of temperature and salinity along seven transects in
the East China Sea in summer 2013. Note that there is an obvious thermally-stratified
water column during the collection of suspended particulate matters in the East China
Sea.

**Fig. 1.** Revised Figure 3

[Figure]

**Figure 8.** Bi-plots showing the relationships of (a) POC vs. PN and (b) POC vs. Chl *a* in
suspended particulate matters investigated in this study. Redfield ratio (dashed line in panel a)
is taken from Redfield (1958).

**Fig. 2.** Revised Figure 8

**Supplement:**

**Appendix**

**Table S1.** Hydrographic properties in both the deep chlorophyll maximum layer and SPM sampling layer investigated in this study.

| Station | Long. (°E) | Lat. (°N) | Chlorophyll Maximum Layer | | SPM Sampling Layer | | | | |
| | | | Depth (m) | Chlorophyll Fluorescence (µg $L^{-1}$) | Depth (m) | Chlorophyll Fluorescence (µg $L^{-1}$) | Temp. (°C) | Sal. | Tur. (FTU) |
|---|---|---|---|---|---|---|---|---|---|
| DH1-1 | 123.00 | 30.00 | 5 | 1.90 | 20 | 0.68 | 20.26 | 33.64 | 0.46 |
| DH1-2 | 123.50 | 30.00 | 10 | 1.47 | 30 | 0.57 | 21.24 | 34.17 | 1.97 |
| DH2-1 | 123.10 | 29.47 | 10 | 3.92 | 20 | 1.12 | 22.09 | 33.92 | 0.18 |
| DH2-2 | 123.40 | 29.32 | 3 | 3.03 | 30 | 0.37 | 20.16 | 33.96 | 0.19 |
| Con2 | 122.15 | 28.50 | 2 | 7.40 | 20 | 0.15 | 19.33 | 32.70 | 4.75 |
| DH3-1 | 122.50 | 29.00 | 3 | 17.46 | 20 | 0.59 | 19.09 | 33.60 | 0.60 |
| DH3-2 | 122.80 | 28.80 | 20 | 1.64 | 30 | 0.68 | 26.14 | 33.60 | 1.07 |
| DH3-3 | 123.20 | 28.60 | 50 | 0.75 | 50 | 0.75 | 26.72 | 33.78 | 0.12 |
| DH3-4 | 123.50 | 28.40 | 50 | 1.39 | 50 | 1.39 | 27.26 | 33.68 | 0.14 |
| DH4-3 | 122.80 | 28.04 | 50 | 0.88 | 30 | 0.37 | 28.20 | 33.58 | 0.08 |
| DH4-4 | 123.21 | 27.78 | 50 | 1.23 | 50 | 1.23 | 26.50 | 33.75 | 0.24 |
| DH5-1 | 121.90 | 27.80 | 30 | 4.07 | 30 | 4.07 | 25.47 | 33.79 | 3.45 |
| DH5-2 | 122.11 | 27.63 | 30 | 1.47 | 50 | 0.17 | 24.43 | 33.86 | 1.26 |
| DH5-3 | 122.40 | 27.46 | 50 | 0.57 | 50 | 0.57 | 24.61 | 34.14 | 0.07 |
| DH5-4 | 122.70 | 27.30 | 50 | 1.43 | 50 | 1.43 | 25.16 | 34.17 | 0.27 |
| DH5-5 | 123.00 | 27.13 | 50 | 0.81 | 50 | 0.81 | 21.91 | 34.31 | 0.14 |
| DH5-6 | 123.20 | 26.96 | 50 | 1.62 | 50 | 1.62 | 22.90 | 34.50 | 0.25 |
| DH5-7 | 123.50 | 26.80 | 50 | 2.66 | 50 | 2.66 | 23.00 | 34.21 | 0.21 |

| | | | | | | | | |
|---|---|---|---|---|---|---|---|---|
| DH5-8 | 123.81 | 26.63 | 70 | 0.72 | 70 | 0.72 | 22.16 | 34.52 | 0.12 |
| DH5-9 | 124.06 | 26.44 | 50 | 0.89 | 50 | 0.89 | 24.58 | 34.44 | 0.17 |
| DH6-1 | 121.20 | 27.38 | No data | No data | 10 | No data | 25.52 | 33.77 | No data |
| DH6-2 | 121.50 | 27.18 | 30 | 1.66 | 30 | 1.66 | 26.11 | 33.81 | 1.54 |
| DH6-3 | 121.79 | 26.99 | 50 | 1.19 | 50 | 1.19 | 26.84 | 33.74 | 0.45 |
| DH6-4 | 122.20 | 26.79 | 50 | 1.08 | 50 | 1.08 | 27.38 | 33.55 | 0.13 |
| DH6-5 | 122.50 | 26.59 | 50 | 1.37 | 50 | 1.37 | 21.71 | 34.20 | 0.30 |
| DH6-6 | 122.80 | 26.39 | 50 | 0.95 | 50 | 0.95 | 22.02 | 34.33 | 0.19 |
| DH6-7 | 123.20 | 26.20 | 20 | 0.99 | 50 | 0.64 | 20.67 | 34.56 | 0.09 |
| DH7-1 | 120.80 | 26.76 | 3 | 1.58 | 20 | 0.75 | 24.39 | 33.92 | 5.53 |
| DH7-2 | 121.11 | 26.58 | 30 | 2.49 | 30 | 2.49 | 26.39 | 33.80 | 0.49 |
| DH7-3 | 121.50 | 26.40 | 50 | 1.14 | 50 | 1.14 | 27.55 | 33.70 | 0.31 |
| DH7-4 | 121.80 | 26.23 | 50 | 0.77 | 50 | 0.77 | 26.12 | 33.75 | 0.18 |
| DH7-5 | 122.10 | 26.05 | 50 | 3.15 | 50 | 3.15 | 22.41 | 34.23 | 0.45 |
| DH7-6 | 122.40 | 25.87 | 22 | 1.54 | 50 | 0.44 | 20.07 | 34.49 | 0.14 |
| DH7-7 | 122.67 | 25.71 | 21 | 1.08 | 50 | 0.41 | 20.79 | 34.42 | 0.12 |
| DH7-8 | 123.00 | 25.52 | 20 | 0.63 | 85 | 0.18 | 21.01 | 34.67 | 0.05 |
| DH7-9 | 123.20 | 25.34 | 75 | 1.12 | 130 | 0.11 | 21.70 | 34.68 | 0.04 |

---

## Author Comment (AC2) · 2 Oct 2017

General overview

The manuscript of Liu et al. is focused on characteristics of suspended particulate organic matter (SPOM) in the deep chlorophyll maximum (DCM) of the Eastern China Sea (ECS) during summer 2013. It is based on bulk descriptors of SPOM (C/N and POC/Chl a ratios as well as $\delta$13C and $\delta$15N). The mains findings are: 1) DCM SPOM mainly originates from in situ primary production, 2) terrestrial POM slightly or insignificantly contributes to DCM SPOM composition, and 3) the latter is contradictory to previous studies but illustrates the drastic decrease in the contribution of the terrestrial POM originating from the Yangtze River to the SPOM composition in the ECS. These findings are sounded and clearly illustrated by the present data set.

The manuscript is well organized and usually well illustrated.

It is of broad audience for scientist who are interested in organic matter cycling and land-to-sea export. It is within the scope of BG. However there are some issues in the present version of the manuscript that preclude the acceptance of the manuscript in its present version. These issues are:

- a lack of information in the methods - many unneeded details and miscellaneous information that are not needed in the discussion, that rend the discussion too wordy and that dilute the main messages of the study. Authors should focus on what the data indicate, which is usually very clear. - interpretations of $\delta$15N data set that are not correct or at least very partial. This data set cannot be published within this manuscript without deeply reconsidering its interpretation and without additional data set regarding N-nutrient (at least nitrate and ammonium) concentrations. There are also some inconsistencies and language errors that have to be corrected. Thus, I recommend major revision

Reply: Thank you very much for your appreciation on the overall performance of the manuscript and critics on the discussion part and interpretations of $\delta$15N data.

Detailed comments

Referee 2: Section 3: lack of information and details needed Conversion of fluorescence into chlorophyll a concentration: Since Chl a is a key parameter of the study and is used to calculate POC/Chl a ratio (which values are compared to reference values), it should be explained how in situ fluorescence was converted into Chl a concentration.

Reply: Chlorophyll a was determined spectrophotometrically according to Lorenzen

(1967) and Aminot and Rey (2000). However, we randomly measured five SPM samples (DH1-2, DH2-1, DH3-1, DH7-1 and DH7-7; Fig. R1) from water depths between 20 m and 70 m. Linear correlation between the fluorescence values obtained directly from the calibrated sensor attached with the CTD rosette and our measured values is high with R2 = 0.93 (Fig. R1). Therefore, as mentioned in Page 11, Lines 18-21 in the original manuscript, we applied Chl a values obtained in situ by fluorometer without any conversion in this study.

Referee 2: Section 3.1, line 9: indicate the range of depths of the samplings for SPM.

Reply: The sentence has been revised as follows:

To investigate the biogeochemical characteristics of POM in DCM layers of the southern East China Sea, suspended particles around DCM water depths (10-130 m) were collected from thirty-six stations along seven transects across the continental shelf by the Science Cruise during summer (June 22-July 21) 2013 (Fig. 1).

Range of sampled depths (10 m -130 m) is also mentioned in Table 1.

Referee 2: Section 3.1: indicate how the filters were rinsed right after the filtration

Reply: In this study, filters were not rinsed/washed after the filtration. Therefore, the sentence has been revised as follows: After filtration, un-rinsed filters were folded and wrapped again in aluminium foil and stored at -20 °C immediately in a freezer onboard before they were brought back to the laboratory for further analysis.

Referee 2: Section 3.2, line 28: detail how the filters were treated with HCl 1N

Reply: A half of each filter with SPM was placed in a polyethylene culture dish and 3 ml of 1N HCl was then added into the dish by a dropper and allowed them to react 16 h to remove inorganic carbon (mainly carbonate).

Referee 2: Section 3.2, line 30: indicate the diameter of the punches

Reply: The diameter of filter has been included in the revised version as follows: Then

a half of de-carbonated filter (i.e. a quarter of the original GF/F filter - 11 mm) was then punched in tin capsules for further analysis.

Referee 2: Section 3.2, lines 30-31: it looks like $\delta$15N and PN were analyzed on the decarbonated part of the filter. Why not on the un-decarbonated part of the filter? There is always chance to bias $\delta$15N and PN using decarbonated material for these measurements (e.g. Lorrain et al (2003) and other references). Also, it looks like there was a very small part of the filters that were analysed for C and N elemental and isotopic composition. What quantities of C and N were analysed?

Reply: We thank the referee to bring the reference Lorrain et al. (2003) to our kind attention. Lorrain et al. (2003) cautioned that the measurement of PN and $\delta$15N after freezing increases the uncertainty of $\delta$15N and in combination with concentrated HCl treatment, leads to a loss of PN and alteration of the $\delta$15N signature. However, previous studies in the East China Sea always followed freeze-drying and carbonate removal using 1N HCl to analyse four parameters (POC, PN, $\delta$13C and $\delta$15N) from a single filter.

In general, SPM collected close to the major river-dominated margins contain particles, including particulate inorganic carbon in the form of calcite, aragonite and dolomite, either supplied from the land and/or from the surface productivity. When we deal with the particulate organic matter that produced mainly by primary producers (i.e. phytoplankton), PN and $\delta$15N values obtained using de-carbonated filters are more appropriate than such values obtained from the un-acidified filters. Similar methodological approach has been adopted by Wu et al. (2003) while investigating suspended particles along the PN transect in the East China Sea and by Hung et al. (1996) while studying the suspended particles in the entire East China Sea. For instance, the range of PN and $\delta$15N values obtained in the present study is comparable to the range of PN and $\delta$15N values obtained by Wu et al. (2003) ($\delta$15N: ca. 0.7-9.4‰. Since we made a comparative discussion of our $\delta$15N data with the data of Wu et al. (2003), similar pre-treatment of samples is a prerequisite for such comparison. On the other hand, Gao

et al. (2014) collected the SPM along the Changjiang transport pathway in different seasons and measured PN and $\delta$15N with the un-acidified filters. In their paper, $\delta$15N values are mostly shown as a distribution diagram and the range of scanned $\delta$15N values (ca. -3.0 to +9.4‰ indicate that the presence of inorganic N might be responsible for negative $\delta$15N values or $\delta$15N measured in their study may not fully represent the true range of $\delta$15N composition of organic matter in their study area.

In our study, the amount of measured C and N ranged from 68.24-322.18 $\mu$g and 14.46-64.69 $\mu$g, respectively. Precision for $\delta$13C and $\delta$15N decreases for samples containing less than 100 $\mu$gC and 20 $\mu$gN, respectively. Among thirty-six filters analyzed for the present study, only five (three) filters contain less than 100 $\mu$gC (20 $\mu$gN).

Influence of CDW at DCM depth in the ECS

Referee 2: It cannot be stated that the influence of CDW in the study site is nil or insignificant. The low salinity measured at some of the sites (Fig. 3) clearly indicates the influence of CDW. It is mainly the case in surface water but also the case at some of the DCM depths where water was sampled for SPM (stations DH1-1, DH2-1, DH2-2, DH3-1 and CON02). This is also clear from Fig. 6b where five stations falls within the SMW square, SMW being clearly a water body composed of a mixing between KSSW and CDW.

It should better be written that the influence of CDW in the study site is low (see some of the 'minor points' below) or weak (as written in section 5.1, P8, line 34).

Reply: As suggested by the Referee, we have softened the tone of the influence of CDW shown in Fig. 3 and Fig. 6b in the revised text.

End of section 5.1 (P9, lines 1-9) and Fig. 7 Referee 2: Only the DCM depths (= the depths of interest for the present study) should be considered for delineate the polygons of Fig. 7. Was it the case? For similar reason, I think that the sentence "Interestingly...study area (Fig. 7)" should be deleted or reworded without citing Fig. 7

(but rather Fig. 3?) since it is quite confusing. Another option may be to not cite depth limitation of the water masses influences but only describe Fig. 7.

Reply: We disagree with this suggestion mainly because the conditions of how different water masses (CDW, TWCW and KSSW) are influencing the DCM depths are shown in Fig. 6b. Further, in Fig. 7 we delineated areas influenced by three water masses both horizontally and vertically for the entire water column. Furthermore, the water masses were delineated based on the T-S combination and therefore, citing Fig. 3 along with Fig. 7 is fine, but deleting the mention of Fig.7 without water depths may mislead the meaning. Therefore, based on Referee's suggestions, we revised the first paragraph of section 5.1 as follows:

5.1 Influence of different water masses in the southern ECS

In order to identify the different water sources in the study area, temperature–salinity (T-S) diagrams were drawn for the entire water column (Fig. 6a) as well as for the SPM sampling depth around DCM layers (Fig. 6b). The T-S diagram for all the water depths (Fig. 6a) shows a convergence at around 17 °C, 34.6, representing the upwelling of KSSW (Umezawa et al., 2014). There are two trends in the T-S diagram, indicating a mixing of three water masses: one is less saline and much colder water, mainly CDW, another is more saline and warmer, mainly Taiwan Warm Current Water (TWCW), and the third one is KSSW (Fig. 6a). The shelf water in the entire ECS in summer 2013 was mixed primarily by three water masses, CDW, KSSW, and TWCW (Fig. 6a). The low salinity observed at five coastal sites (DH1-1, DH2-1, DH2-2, DH3-1 and CON02) indicates the influence of CDW mostly in surface water, but also some of the DCM depths where water was sampled for SPM (Fig. 3). This is also evident from Fig. 6b where these five stations fall within the area of SMW, which is a water body composed of a mixing between CDW and KSSW. However, except these five coastal stations, most DCM depths where water was sampled for SPM seem to be weakly influenced by the CDW (Fig. 6b). Based on the T-S range of different water masses (Fig. 6), we further delineated the area influenced, along with water depths, by three important

water masses: CDW, TWCW and KSSW (Fig. 7). Interestingly, the influence of CDW was constrained only in the upper 0-10 m in five coastal stations during the sampling time, whereas TWCW influences around 0-30 m, covering three-fourths of the study area, and KSSW seems to be largely influenced the bottom water of the entire study area (Figs. 3, 6a and 7).

Referee 2: The last paragraph of section 5.1 is also quite confusing. Reword it as: "In summary, although the river runoff was huge, the influence of CDW plume in the southern part of the ECS was insignificant during summer 2013, mainly because most of the CDW plume was transported to northeastward of the Yangtze estuary to the Korean coast (Isobe et al., 2004; Bai et al., 2014; Gao et al., 2014). This contrasts with summer 2003 when the plume front moved southward (Bai et al., 2014). Meanwhile, the intrusion of TWCW and KSSW was strong in the continental shelf of the East China Sea during summer 2013."

Reply: The last paragraph of section 5.1 has been reworded exactly, as suggested by the Referee.

Section 5.2.1 Referee 2: I fully agree the main conclusions of this section and most of data interpretations (especially the first and the last paragraphs).

However the second paragraph adds detailed discussion with literature comparison that is not needed (especially when dealing with zooplankton and Trichodesmium) for the present study. Authors should better goes directly to the conclusion (i.e. the last paragraph) without diluting the main conclusions with unneeded wording. Thus, the second paragraph should be deleted.

Reply: As suggested by the Referee, the second paragraph of section 5.2.1 has been deleted in the revised version.

Section 5.2.2 Referee 2: As for the previous section, I fully agree the main conclusions and most of the data interpretation, but this section is too wordy and gives too many

details (especially too many values from the literature). Authors should better focus on the main information and the main conclusions.

Thus, I suggest the following:

Referee 2: - P10, lines 26-34: one-two sentence(s) should be enough

Reply: Lines 26-34 have been shortened/revised as follows:

Moreover, the POC/Chl a ratio of 35.3 g g–1 derived from the slope of a regression line (y = 35.3 (±8.56) x + 44.0 (±6.27) (Fig. 8b) is consistent with the reported POC/Chl a ratios in the ECS, (36.1 g g-1; Chang et al., 2003) and the North-western Pacific (48 g g-1; Furuya, 1990). However, our POC/Chl a ratio is lower than that estimated (64 g g-1) for the sinking particles in the ECS and the Kuroshio region, off northeast Taiwan Island (Hung et al., 2013). The range is well within the range (13–93 g g-1) reported by Chang et al. (2003) in the ECS and estimated (18–94 g g-1) from phytoplankton cell volumes by the same authors.

Referee 2: - P11, lines 2-4: keep this sentence but rephrase the last line as "filtered particles (Chang et al., 2003; Hung et al., 2013)"

Reply: As suggested, the sentence has been rephrased as follows: Although the Chl a concentration in our study was measured in situ by fluorometer attached with the CTD, it is more or less similar to Chl a concentrations obtained in the above-mentioned studies, which were mostly extracted from filtered particles (Chang et al., 2003; Hung et al., 2013).

Referee 2: - P11, lines 8-10: do not report all these values

Reply: These lines have been modified in the revised version as follows: The POC/Chl a ratio of living phytoplankton was reported to be between 40 and 140 g g-1 (Geider, 1987; Thompson et al. 1992; Montagnes et al. 1994; Head et al. 1996).

Referee 2: Regarding the high POC/Chl a ratio, did authors check if these high values

were rather due to very low Chl a concentration or high POC concentration? If the former, these high values may be associated to high uncertainty on the Chl a estimation when values are low. If the latter (high POC concentration associated to Chl a concentration similar to surrounding stations), this may be effectively due to heterotrophic biomass.

Reply: In this study, only two SPM samples contain high POC/Chl a ratios of >200 g g-1 (DH5-2: 369 g g-1 and CON02: 617 g g-1). Although both show neither high POC concentration (DH5-2: 62.6 $\mu$g L-1, CON02: 92.6 $\mu$g L-1) nor high Chl a content (DH5-2: 0.17 $\mu$g L-1, CON02: 0.15 $\mu$g L-1), the Chl a values in these two stations are relatively low, as shown in Fig. 8b. Therefore, higher POC/Chl a ratio in these two SPM samples is perhaps because of the uncertainty associated with the estimation of Chl a using fluorometer on board.

Section 5.3: first three paragraphs Referee 2: I fully agree the main conclusions and the data interpretations of the first three paragraph of this section.

I suggest authors to have a look at Lowe et al. (2014) and Miller et al (2013): these articles are of interest for the present section.

Page 12, Lines 18-26: two other processes may influence phytoplankton $\delta$13C: temperature and degradation. This is discussed in Savoye et al. (2003) that authors cite in many occurences. Authors may have a look at biplots like $\delta$13C vs temperature and versus POC/Chl a and C/N (considering these ratios may also indicate phytoplankton decay). They also may check the normalization of $\delta$13C by temperature (as in Savoye et al., 2003) before plotting normalized $\delta$13C versus POC, since temperature usually have (indirect) influence on phytoplankton $\delta$13C.

Reply: As directed, a section 5.4 on "Temperature effect on $\delta$13CPOC" data has been included in the revised version as follows:

5.4 Temperature effect on $\delta$13CPOC data

Apart from primary production and the growth rate and species composition, temperature and biomass degradation may influence the carbon isotopic composition of phytoplankton (Savoye et al., 2003; Miller et al., 2013; Lowe et al., 2014). Temperature has an indirect effect on isotopic fractionation between phytoplankton carbon and dissolved CO2, and therefore on phytoplankton $\delta$13C (e.g., Rau et al., 1992; Savoye et al., 2003). The C/N ratio, POC/Chl a ratio and $\delta$13CPOC indicated that the POM around DCM layers is dominated by newly-produced phytoplankton OM (see Sections 5.1-5.3). Therefore, to understand the temperature effect on $\delta$13C of phytoplankton, we plotted our $\delta$13CPOC data against temperature into two groups by separating approximately at $\sim$24°C (Fig. 2Ra). Data points of both groups show a decreasing $\delta$13C of phytoplankton biomass while temperature increases at DCM layers in the southern East China Sea (Fig. 2Ra). Such a relationship is in contrast to the positive relationship between these two variables observed for the surface ocean around the world (Sackett et al., 1965; Fontugne, 1983; Fontugne and Duplessy, 1981).

The negative relationship between $\delta$13CPOC and temperature is likely related to biological activity and carbonate dissolution equilibrium, both may control the dissolved inorganic carbon concentration in the DCM layers, which are closer to euphotic depths (see Section 4.1). The weak correlation between $\delta$13CPOC and temperature supports a weak influence of temperature on $\delta$13CPOC around DCM layers in the southern East China Sea (Fig. 2Ra). A decrease in fractionation of approximately -0.56‰ °C-1 is estimated for POM collected at <24°C, whereas a decrease in fractionation of roughly -0.51 °C-1 is estimated for POM collected at >24°C (Fig. 2Ra). In order to distinguish the influence of biological parameters from temperature on $\delta$13CPOC, the $\delta$13CPOC data were corrected for the 'temperature effect' by normalizing the data using an equation: $\delta$13CPOC = f (T).

Since most of our $\delta$13CPOC values come from the DCM layers and the $\delta$13CPOC is negatively correlated with temperature, we applied our own temperature coefficients (-0.56‰ °C-1 and -0.51‰ °C-1) and $\delta$13CPOC was normalized at 24°C (i.e. the mean

temperature at sampled water depths) using the formula (Savoye et al., 2003):

$\delta 13C24°C = \delta 13C$ - s (T - 24), where $\delta 13C24°C$ is the temperature-normalized $\delta 13CPOC$, T is the seawater temperature in °C from water depths where SPM sampled, and s is the slope of the linear regression $\delta 13CPOC = f$ (T) in ‰ °C-1 obtained from Fig. 2Ra. There are significant correlations between $\delta 13C24°C$ of biomass and POC concentration (circles: R2 = 0.71; p<0.0001; n = 18 and triangles: R2 = 0.66; p<0.0001; n = 18; Fig. 2Rb), indicating that primary production drives ∼70% of the variation of phytoplankton $\delta 13C$ around DCM layers in the southern East China Sea. On the other hand, $\delta 13C24°C$ correlated insignificantly with POC/Chl a ratio and C/N ratio (Figs. 2Rc and 2Rd), implying that degradation has a minor effect on the isotopic composition of POM in this study.

Section 5.3: last three paragraphs

Referee 2: I do not think that the last three paragraph of the section are needed. The objective of the two paragraphs before the last (from "The nutrient N/P ratio" to "this mechanism is most likely") is to decipher whether POM sampled in the DCM came from in situ production or from surface production (cf. the fourth paragraph of the section). In fact, these two paragraphs lay on very putative argumentation and do not allow (and are not convincing in) deciphering between the two hypotheses. These hypotheses have already been discussed in sections 5.2.1 and 5.2.2 with sufficient argumentation for considering that POM mainly came from in situ production. To my point of view, these two paragraphs of section 5.3 are not needed in this section neither in the manuscript.

Reply: As suggested, these two paragraphs have been deleted in the revised version.

Referee 2: The last paragraph of the section is a tentative of inventory of POC in the DCM layer. The estimation is very rough, is associated to large uncertainty, and the calculation is not convincing. It is also completely disconnected from the rest of the section, which is focused on $\delta 13C$ dynamics (see the title of the section). Again, this

paragraph is not needed in this section neither in the manuscript. Thus, the last three paragraphs of the section should simply be deleted and the fourth paragraph of the section ("The range... DCM layers?") should be replaced with a brief conclusion of the first three paragraphs of the section.

Reply: We agree that the POC inventory in the DCM layer has been done approximately and therefore can be deleted in the revised version.

Referee 2: Section 5.3 would better stand with the first three paragraphs and a conclusion without the unclear and unconvincing last three paragraphs.

Reply: As suggested, all three paragraphs have been deleted in the revised version.

Section 5.4 Referee 2: This section is the less clear and the less convincing of the manuscript. The main conclusion (POM $\delta$15N distribution is primarily governed by the nutrient status and $\delta$15N of nitrate) is mainly guess. One of the main issues of the section is the lack of nutrient data. This rend the data interpretation mainly guesswork. Another issue is that authors mainly take into account nitrate as a nutrient for phytoplankton. Ammonium appears only in the last paragraph. The other species of N-nutrient (N2, dissolved organic nitrogen as urea) are not mentioned. However, it is reported that "Kuroshio Water and TWCW induced Trichodesmium" (P10, line 7). Thus, PN $\delta$15N values should also be discussed considering N2-fixers (diazotrophs). At last, many sentences are not clear. This gives the impression that authors do not fully have in mind what processes drives PN and phytoplankton $\delta$15N.

Thus, this section should be deeply reworked including deep data re-interpretation. To me, such section dedicated to PN/phytoplankton $\delta$15N cannot stand without data of nutrient concentration originating from the same cruise. If these data are not available, PN/phytoplankton $\delta$15N cannot be discussed. If it would be the case, $\delta$15N data should be removed from the manuscript.

Reply: Thank you for the suggestion! At present, we don't have depth profiles of nitrate or ammonia or nitrogen isotopic composition of nitrate to strengthen the PN/$\delta$15N data and related interpretations. To our knowledge, all our interpretations of PN/$\delta$15N are based on our understanding of nitrogen dynamics in the ECS and are consistent with the literature cited. This part is also appreciated by Referee 3, but also suggested to strengthen the interpretations using depth profiles of nutrients. However, there is no much information related to $\delta$15N data, especially from the biota-dominated DCM layers, in the East China Sea. Since we proved that the POM is dominated by phytoplankton, publishing $\delta$15N data of this study may create some awareness/interests among readers to conduct such investigations in the marginal seas of the western Pacific in detail. Given this, we request anonymous reviewers and Associate Editor to allow this section for publication because we interpreted PN/$\delta$15N data based on the published information from the East China Sea with some speculations, a practice that is normally encouraged in the scientific field when the availability of data is relatively less such as $\delta$15N.

Section 5.5 Referee 2: This section gives an important conclusion: the influence of terrestrial POM (mainly originating from the Yangtze River) has drastically decreased in the ECS. This section is mainly based on literature data and conclusions. These inputs from the literature are of interest, but the section should also compare data from the present study with previous data. Thus, this section should start with the comparison of POC/Chl a and C/N ratios, and $\delta$13C values between the present study and previous studies. Then, the decrease of terrestrial POC fluxes and deposition can be cited (literature data). Last lines of the section: there are again unneeded details in these lines. Avoid describing the degradation index but directly give the conclusions of Wu et al (2007b).

Reply: As directed, section 5.5 has been revised as follows:

We examined elemental and isotopic compositions of carbon and nitrogen (POC, PN, $\delta$13C and $\delta$15N) in suspended particulate matters and water column hydrographic and environmental parameters around the deep chlorophyll maximum (DCM) layers in the

continental shelf of the southern ECS during summer 2013. The range of POC/Chl a obtained in this study (26-200 g g-1) is within the range (<200 g g-1) reported for the phytoplankton-dominated POM in the coastal and shelf waters (e.g., Chang et al., 2003; Savoye et al., 2003; Hung et al., 2013; Liénart et al., 2016). We obtained a narrow range of C/N ratio (4.1-6.3), but a wide range of $\delta$13C (-25.8 to -18.2 ‰ compared to previous studies in the ECS (Liu et al., 1998; Wu et al., 2003). Our results indicated that POM around the DCM water depths was largely derived from the synthesis of in situ phytoplankton and the influence of terrestrial OM supplied by the Yangtze River in the ECS is low. The missing of terrestrial OM signals seems to be related to reservoir and dam buildings along the river in recent years that has shifted the location of the Yangtze-derived POC deposition from the inner shelf of the ECS to terrestrial reservoirs (Li et al., 2015). The sediment delivered from the river to the estuary has reduced by 40 % since 2003 when Three Gorges Dam (TGD) has completed (Yang et al., 2011 and references therein). Recently, Dai et al. (2014) reported that the particles discharged by the Yangtze has declined to 150 Mt yr-1, less than ∼70% of its sediment delivery to the ECS during 1950s. Although 87 % of the mean annual sediment of Yangtze River was discharged during the flood season from June to September (Wang et al., 2007; Zhu et al., 2011), approximately 60 % of the fine-grained sediments are temporarily deposited near the estuary and then later resuspended and transported southward along the inner shelf, off the mainland China coastline (Chen et al., 2017 and references therein). The Yangtze-transported POM moves up toward the northeast across the shelf along the so called the Changjiang transport pathway in summer season (e.g., Gao et al., 2014), which is largely driven by the combined effects of high river discharge, southwest summer monsoon and the intensified TWC (Beardsley et al., 1985; Ichikawa and Beardsley, 2002; Lee and Chao, 2003). The T-S diagrams (Figs. 6 and 7) of this study also illustrate this view.

Accompanying with the decreasing sediment input, dam building in the Yangtze River basin has buried around 4.9±1.9 Mt yr-1 biospheric POC since 2003, approximately 10% of the world riverine POC burial flux to the oceans (Li et al., 2015). The POC input

flux from the Yangtze to the ECS (range: 1.27-8.5 x 1012 g C yr-1; Wang et al., 1989; Qi et al., 2014) was significantly less than the estimated primary productivity (72.5 x 1012 g C yr-1; Gong et al., 2003), implying the predominance of marine-sourced organic matter in the ECS. Moreover, the substantial quantity of organic substances that transported by the Yangtze River may be completely modified before being ultimately deposited onto the ECS inner shelf and being transported further offshore (Katoh et al., 2000; Lie et al., 2003; Chen et al., 2008; Isobe and Matsuno, 2008). Wu et al. (2007b), for instance, observed an advanced stage of POM degradation in the entire Yangtze River with an average degradation index of -1.1. Based on the investigation of lipids biomarkers in a sediment core collected from the ECS, Wang et al. (2016) suggested the dominant preservation of marine autochthonous organic matter (∼90 %) in the ECS.

English language Referee 2: The language is usually quite understandable, but there are many errors or mistakes. Part of them is listed in the 'minor points' below. Nevertheless, the whole manuscript should be deeply checked for English language.

Reply: The whole manuscript has been carefully checked for grammatical errors.

Inconsistencies Referee 2: There are some inconsistencies between values that are cited in the text and values reported in the tables (see 'minor points' below). Please, check the consistency between all the values reported in the text and tables.

Reply: Cross-checked and corrected.

Abstract, Introduction and Summary and conclusions Referee 2: Sections 'Abstract' and 'Summary and conclusions' should partly be re-written taking into account the above detailed comments.

Reply: As suggested, these parts have been modified.

Referee 2: The third objective that appear in section Introduction should be removed (see one of my comments dedicated to section 5.3 âËŸAËĞT three last paragraphs

[Figure]

âËŸAËĞT above).

Reply: The third objective in the Introduction part will be deleted in the revision.

Minor points

- P1, Line 20: what do you mean with 'straddling'?

Reply: It means locating or moving around DCM depth intervals.

- P2, line 9 and in the whole manuscript: replace 'endmember' with 'end-member'

Reply: Replaced.

- P3, line 8: remove 'which in turn, the elemental and isotopic compositions of marine productivity' since it is not correct

Reply: Deleted.

- P5, line 1: depending what you want to say, add 'by' or 'to' between 'decreased' and '86%'

Reply: As suggested, 'by' is added.

- P5, line 17: replace 'had' with 'have'

Reply: Replaced.

- Last sentence of section 3.1: place this sentence in section 3.2 since it is not sample collection. Rename section 3.2 as 'Determination of SPM concentration and analysis of POC, PN, $\delta$13C and $\delta$15N

Reply: The last sentence of section 3.1 has been shifted to section 3.2. As suggested by the Referee, section 3.2 has also been renamed in the revised version.

- P5, line 30: replace 'with' with 'and placed in'

Reply: Replaced.

- Section 3.2, P5-6, sentence "organic carbon and nitrogen ... entering the IRMS": remove the sentence since such level of detail is not needed

Reply: The sentence has been deleted in the revised version.

- Section 3.2: remove the three last sentences ("Conventional...Sigman et al., 2009" since the first one is unneeded detail and the two last ones do not stand in a section dedicated to methods.

Reply: As suggested, the last three sentences in section 3.2 have been deleted in the revised version.

- P6, line 22: add 'usually' between 'profiles of Chl a' and 'show'.

Reply: Added.

- Section 4.1.3: since Fig. 3 illustrates at maximum the first 300m of the water column and since the sampling depth was within this depth interval, please do not describe deeper water, either the reading is quite disturbing. Thus, the temperature ranged between 30 and ca. 15 °C.

Reply: We agree with the Referee's view and therefore the paragraph describing the range of temperature has been revised as follows:

Figure 3 illustrates the vertical distributions of temperature and salinity along seven transects in the ECS. In the entire study area, temperature in the 300-m water column varied from 15 °C to 30 °C and distinct water column stratification was evident from the temperature profiles (Fig. 3). The temperature decreases when depth increases and the highest temperature (>30 °C) seen mostly in the surface water and the lowest temperature (5 °C) was observed in stations DH7–8 and DH7-9 at water depths of 850 m and 800 m, respectively (not shown). Temperature at sampling depths of SPM ranged from 19.1 °C to 28.2 °C, showing a general decreasing trend from the inner to outer shelf in each transect (Fig. 3).

- P7, line 16: replace 'increasing' with 'increases'; reword "with the high temperature (>30 °C) spreads widely".

Reply: Replaced. Please see our reply to the previous comment.

- P7, line 29: replace 'insignificant' with 'low'.

Reply: Replaced.

- P8, line 4: '4.4' or '4.5' as reported in table 1?

Reply: The correct value is 4.4 and it is corrected in Table 1.

- P8, line 5: '17.7' or '17.8'?

Reply: The correct value is 17.7 and Table 1 is corrected accordingly.

- P8, lines 7-8: please also indicate where the highest POC and PN concentration were located.

Reply: The following sentence has been included in the revised version: The highest concentrations of POC (263 $\mu$g L-1) and PN (52.8 $\mu$g L-1) are associated with station DH5-1 (Fig. 4).

- P8, line 17: '8.0' or '7.8' as reported in Table 1?

Reply: The correct value is 8.0 and Table 1 is revised accordingly.

- P8, lines 17-18: please also indicate where the highest $\delta$13C values were located.

Reply: The following sentence has been included in the revised version: Consistent to the POC concentration, the highest $\delta$13CPOC value (-18.2 ‰ is also associated with station DH5-1.

- P8, line 21: Fig. 10 is cited before Fig. 5. Check the numbering of the figures.

Reply: Fig. 10 is cited after Fig.5, which was cited just above in the text. We have cross-checked all figure numbers in the revised version.

[Figure]

- P8, line 33: SMW is a water body that is composed of a mixture between two other water masses (CDW and KSSW; Fig. 6). So, do not consider SMW as a water mass and remove it from this list.

Reply: Removed.

- P10, line 25: remove the word 'moderate' since this information is not useful here.

Reply: Deleted.

- P10, line 29: '48' or '52' as reported in Table 2?

Reply: According to Table 2, the value 48 is for the northwestern Pacific and the number 52 is for the western Pacific. As suggested by the referees, Table 2 has been deleted in the revised version.

- P11, line 28: replace 'less' with 'low'; delete "and unrecognized content of terrestrial POM".

Reply: Replaced and deleted.

- P12, 2: replace 'to be' with 'would be'.

Reply: Replaced.

- P12, line 8: replace 'more positive' with 'less negative'.

Reply: Replaced.

- P12, line 34: delete 'As for species,'.

Reply: Deleted.

- P13, line 7: delete 'that'.

Reply: Deleted.

- P13, line 8, 9 and 10: replace 'larger' with 'higher'

Reply: Replaced.

- P13, lines 9 and 10: replace 'size species' with 'phytoplankton'

Reply: Replaced.

- P15, line 33: replace 'significantly less' with 'low'.

Reply: Replaced.

- P16, line 12: replace 'proved' with 'illustrate'.

Reply: Replaced.

- Table 1: add POC/Chl a values in this table; indicate in the caption what means 'SD'.

Reply: POC/Chl a ratios have been included in Table 1 with SD abbreviation has been indicated.

- Figure 1: indicate KSSW on the figure; be consistent with Fig. 6. Indicate on this figure the location of the stations that appear on Fig. 2 and 3 but were not sampled for SPM in the DCM.

Reply: Fig. 1 shows the simplified current pattern in the ECS, and the center of the upwelling region. As one of the water masses, it is appropriate to show KSSW in Figures 6 and 7 along with other water masses.

Stations where SPM were not sampled at the deep chlorophyll maximum layer are listed in Table S1 and those stations are also marked in Fig. 1 in the revised version (red circles in Fig. 1).

Figure 6: the two colours are not distinguishable. Choose other colours. Remove 'from' in the second line of the caption. Add 'were' after 'matters' in the third line of the caption.

Reply: The two colours in Figure 6 are changed. Other corrections are included, as suggested.

Figure 7: replace 'black' with 'grey' in the second line of the caption.

Reply: Replaced.

Figure 8: first line of the caption: it is POC vs. PN and POC vs. Chl a.

Reply: Corrected.

Additional references

Lorrain A., N. Savoye, L. Chauvaud, Y-M. Paulet and N. Naulet, 2003. Decarbonation and preservation method for the analysis of organic C and N contents and stable isotope ratios of low-carbonated suspended particulate materiel. Analytica Chimica Acta, 491, 125-133. Lowe, A. T., A. W. E. Galloway, J. S. Yeung, M. N. Dethier, and D. O. Duggins. 2014. Broad sampling and diverse biomarkers allow characterization of nearshore particulate organic matter. Oikos. 123: 1341–1354, doi:10.1111/oik.01392 Miller, R. J., H. M. Page, and M. A. Brzezinski. 2013. _13C and _15N of particulate organic matter in the Santa Barbara Channel: drivers and implications for trophic inference. Mar. Ecol. Prog. Ser. 474:53-66, doi:10.3354/meps10098

Reply: These additional references are included in the revised version.

Thank you very much.

Additional References

Aminot, A., and Rey, F.: Standard procedure for the determination of chlorophyll a by spectroscopic methods, ICES Techniques in Marine Environmental Sciences, Copenhagen, Denmark, 8-11, 2000.

Fontugne, M. R.: Les isotopes stables du carbone organique dans l'océan: application à la paléoclimatologie, PhD thesis, Université de Paris XI, 1983.

Fontugne, M. R, and Duplessy, J. -C.: Organic carbon isotopic fractionation by marine plankton in the temperature range -1 to 31°C, Oceanol Acta, 4, 85-90, 1981.

[Figure]

Lorrain A., Savoye, N., Chauvaud, L., Paulet Y. -M., and Naulet, N.: Decarbonation and preservation method for the analysis of organic C and N contents and stable isotope ratios of low-carbonated suspended particulate material. Anal. Chim. Acta, 491, 125-133, 2003.

Lorenzen, C. J.: Determination of chlorophyll and pheo-pigments: Spectrophotometric equations, Limnol. Oceangr., 12, 343-346, 1967.

Lowe, A. T., Galloway, A. W. E., Yeung, J. S., Dethier, M. N., and Duggins, D. O.: Broad sampling and diverse biomarkers allow characterization of nearshore particulate organic matter, Oikos, 123, 1341-1354, 2014.

Miller, R. J., Page, H. M., and Brzezinski, M. A.: $\delta$13C and $\delta$15N of particulate organic matter in the Santa Barbara Channel: drivers and implications for trophic inference, Mar. Ecol. Prog. Ser., 474, 53-66, 2013.

Rau, G. H., Takahashi, T., Des Marais, D. J., Repeta, D. J., and Martin, H.: The relationship between $\delta$13C of organic matter and [CO2(aq)] in ocean surface water: data from a JGOFS site in the northeast Atlantic Ocean and a model, Geochim. Cosmochim. Acta, 56, 1413-1419, 1992.

Sackett, W. M., Eckelmann, W. R., Bender, M. L., and Bé, A. W. H.: Temperature dependence of carbon isotope composition in marine plankton and sediments, Science, 148, 235-237, 1965.

[Figure]

Figure R1

**Figure R1**. The relationship between the concentration of Chl *a* measured in the laboratory and the fluorescence values obtained using *in situ* fluorometer onboard.

**Fig. 1.** Figure R1

[Figure]

**Figure R2.** Bi-plots showing the relationships of (a) $\delta^{13}C_{POC}$ vs. temperature for samples separated into two categories based on temperature: <24 ℃ and >24 ℃, (b) temperature-normalized $\delta^{13}C$ ($\delta^{13}C_{24℃}$) vs. POC concentration, (c) $\delta^{13}C_{24℃}$ vs. POC/Chl $a$ ratio and (d) $\delta^{13}C_{24℃}$ vs. molar C/N ratio in suspended particulate matters from the deep chlorophyll maximum layers in the southern East China Sea.

**Fig. 2.** Figure R2

[Figure]

**Figure 1.** Map showing the locations of suspended particulate matters (SPM) collected around the deep chlorophyll maximum (DCM) layers from the East China Sea (ECS) during summer (June 22–July 21) 2013 for the present investigation. Also shown are the modern current patterns in the ECS. Red circles mark the SPM samples that were collected either below or above the DCM layer. CDW – Changjiang Diluted Water, CCC – China Coast Current, TWC – Taiwan Warm Current and KC– Kuroshio Current. The dashed ellipse represents the center of Kuroshio upwelling due to an abrupt change in the bottom topography in the northeast of Taiwan Island (Wong et al., 2000). Also shown is the PN transect, a cross shelf transect that is relatively well studied for particulate organic matter dynamics in the East China Sea.

**Fig. 3.** Revised Figure 1

[Figure]

**Figure 6.** Temperature–Salinity (*T*–*S*) diagrams for (a) the entire water column in the East China Sea and (b) the deep chlorophyll maximum layers where the suspended particulate matters were collected for the present investigation. *T*–*S* ranges of six water masses are taken from Umezawa *et al.* (2014). TWCW – Taiwan Warm Current Water; SMW – Shelf Mixed Water; KSW – Kuroshio Surface Water; KSSW – Kuroshio Subsurface Water; KIW – Kuroshio Intermediate Water.

**Fig. 4.** Revised Figure 6

---

## Author Comment (AC3) · 2 Oct 2017

General Comment: This manuscript characterized the bulk and isotopic composition of organic matter collected in DCM layer of the south East China Sea in summer time. The study is well designed and neatly presented. It observed the marine derived material is the dominant organic matter in DCM layer. The influence of the Yangtze River is very limited. Additionally, the nitrogen isotopes elucidated the potential role of N2 fixing in middle shelf where TWCW is dominated and remineralized nutrients plays an important role. However, in this point, the depth profile will be very helpful to strengthen

the discussion but it is lack in the manuscript. Furthermore, the influence of lateral transport (cross shelf) better to be considered when the authors estimate the nutrients contributions from different sources and POC inventory.

Reply: Thank you for your positive opinions on the overall work presented here. At present, we don't have depth profiles of nitrate or ammonia or nitrogen isotopic composition of nitrate to strengthen the PN/$\delta$15N data and related interpretations. However, we reiterate that there is no much information related to $\delta$15N data, especially from the biota-dominated DCM layers, in the southern East China Sea. Since we proved that the POM is dominated by phytoplankton, publishing $\delta$15N data of this study may create more interests among readers to conduct such investigation in the marginal seas of the western Pacific. Given this, we request anonymous reviewers and Associate Editor to allow this section for publication because we interpreted PN/ $\delta$15N data based on the published information from the southern East China Sea.

Specific comments:

Referee 3: 1) The title better be more specific in study region, such as southern East China Sea

Reply: The title of the manuscript has been modified as follows: Biogeochemical characteristics of suspended particulates at deep chlorophyll maximum layers in the southern East China Sea

Referee 3: 2) Abstract: OK

Reply: Thank you.

Referee 3: 3) Introduction: Better to emphasize the status of DCM in ECS and the potential role in POC inventory estimation and hypothesis

Reply: The text related to the POC inventory has been deleted in the revised version, as suggested by the Referee 2.

Referee 3: 4) Methods P5, the filtration volume was in the range of 0.5-2L, the author used half filter for POC or PN analysis, Did they have enough material for reliable analysis, especially for nitrogen?

Reply: We have mistakenly mentioned the filtration volume in the original manuscript. Our apologies! We filtered 4.1-19.1 L of water samples for the collection of SPM. The volume of filtration has been changed accordingly in the revised version.

In our study, the amount of measured C and N ranged from 68.24-322.18 $\mu$g and 14.46-64.69 $\mu$g, respectively. Precision for $\delta$13C and $\delta$15N decreases for samples containing less than 100 $\mu$gC and 20 $\mu$gN, respectively. Among thirty-six filters analyzed for the present study, only five (three) filters contain less than 100 $\mu$gC (20 $\mu$gN).

Referee 3: 5) Result and interpretations: The order of hydrographic characteristics can be adjusted as salinity, turbidity and chla

Reply: As suggested, we rearranged the order of hydrographic characteristics in the revised version.

6) Discussion: In general, the authors gave proper credit to related work and clearly indicate their own new contribution to the biogeochemical cycles in the study area.

Reply: Thank you.

Some minor suggestions:

Referee 3: a. P13-14, How to use C/P ratio to estimate the Yangtze-sourced nutrients for marine primary productivity, how does the lateral cross shelf transport contribute to the POC inventory?

Reply: The text related to the C/P ratio and POC inventory have been deleted in the revised version, as suggested by the Referee 2.

Referee 3: b. P15, L20, this paragraph is a bit speculative and need more straightforward data to support itself, the depth profile data could be bit helpful to elucidate.

Reply: Please refer to our reply to the general comment of Referee 3.

Referee 3: c. Comments on 5.5, why the Yangtze River will play an important role in DCM OM in south ECS, the paragraph of TGD can be moved which seems less related to this topic, also I am confused about how the author summarized in the abstract" SPM investigated here seems not to be influenced by the terrestrial organic matter supplied by the Yangtze River (Changjiang) in summer 2013, a finding that is contrary to a number of previous studies' conclusion.", which is not convinced in this part.

Reply: The transport pathway of Yangtze River debouched sediments to the Okinawa Trough is one of the unsettled issues in the oceanographic studies of the East China Sea. Recently, Chen et al. (2017) suggested that the Changjiang river plume flows southward when sediments are resuspended along the China coast by cyclonic storms. Most previous studies also had shown that cross-shelf transport of Yangtze-derived sediments to the Okinawa Trough. So, one would expect the influence of Yangtze-derived sediments in the southern ECS during summer.

The part of the sentence "a finding that is contrary to a number of previous studies' conclusion" in the abstract has been deleted in the revised version.

Referee 3: d. The quality of Figure 4 and 5 should be improved. Figure 8b, there is 35 samples summarized but all the other plotted based on 36 samples, why?

Reply: The quality of Figures 4 and 5 has been improved. There is no fluorescence/Chl a data for station DH6-1 (see Table S1 in the appendix) and therefore plots with Chl a data contain only 35 data points, including Fig. 8b.

Thank you very much.

―――――――――――――――――

[Figure]

**Figure 4.** Spatial distributions of suspended particulate matters (SPM, mg L$^{-1}$), particulate organic carbon (POC, μg L$^{-1}$) and nitrogen (PN, μg L$^{-1}$) around the deep chlorophyll maximum layer in the southern East China Sea during summer 2013.

**Fig. 1.** Revised Figure 4

**Figure 5.** Spatial distributions of stable isotopic values of particulate organic carbon and nitrogen ($\delta^{13}C_{POC}$ and $\delta^{15}N_{PN}$) around the deep chlorophyll maximum layers in the East China Sea during summer 2013.

**Fig. 2.** Revised Figure 5

---

## Author Response (AR1)

We are very thankful to the Associate Editor, PA Meyers and two anonymous reviewers for their constructive feedbacks and insightful comments on our manuscript entitled "Biogeochemical characteristics of suspended particulates at deep chlorophyll maximum layers in the East China Sea" (**MS. Ref. No.: bg-2017-290**).

**Referee 1 (Prof. P.A. Meyers)**

*Referee 1*: Liu and colleagues present the results of their study of the organic carbon and nitrogen contents in suspended particles collected around deep chlorophyll maximum layers in the East China Sea. They measured carbon and nitrogen concentrations and isotopic compositions of 36 samples collected from 7 cross-shelf transects and augmented these data with a suite of standard hydrographic measurements. These data allowed them to conclude that little land-derived organic matter contributes to the suspended particulate matter despite the proximity of the sampling locations to the mouth of the Yangtze River. Instead, the organic matter freshly produced by phytoplankton. The authors attribute variations that they found in the carbon and nitrogen isotopic values of the organic matter to local differences in productivity rates and differences in the nitrate isotopic signatures of the major water masses in the area. The study seems to have been designed well, and the authors seem to have interpreted their results properly, but problems with the presentation make this contribution hard to read and appreciate. The English badly needs refining, and some additional details should be addressed.

**Reply**: Thank you very much for your appreciation on the overall performance of the research work that presented in the manuscript. We took the utmost care to refine our English in the revised version.

*Referee 1*: For a start, the second paragraph of the Introduction seems to be missing something (lines 27-32).

**Reply**: It seems that there is no link between the last two sentences of the first paragraph and the first sentence of the second paragraph and therefore this part has been revised as follows (**P3, L3-17 in the marked-up manuscript**):

It is known that stable isotopes ($\delta^{13}$C, $\delta^{15}$N) and molar C/N ratios of POM in estuarine and marine environments are representative of these values in primary production-derived OM and in that they are largely synthesized by phytoplankton (Gearing et al., 1984). Since phytoplankton is the main primary producer of marine OM, the elemental and isotopic compositions of phytoplankton should therefore be considered while studying the dynamics of POM in the marine water column.

Chlorophyll *a* (Chl *a*) concentration in sea water is often used as an index of phytoplankton biomass and phytoplankton carbon (Cullen et al., 1982; Malone et al., 1983). The deep chlorophyll maximum (DCM) layer, which contributes significantly to the total biomass and primary production in the whole water column (Weston et al., 2005; Hanson et al., 2007;

Sullivan et al., 2010), is approximately equal to the subsurface biomass maximum layer (e.g., Sharples et al., 2001; Ryabov et al., 2010).

*Referee 1*: After that, Section 3.1 on sample collection should include a tabulation of the 36 samples that shows their water depths and some of their hydrographic properties. This tabulation could be an appendix, but the information that it would contain should be available to interested readers.

**Reply**: Table S1 is included in the appendix in the revised version. Table S1 includes station, coordinates, depth, hydrographic parameters (temperature, salinity and turbidity), chlorophyll fluorescence concentration and elemental (POC, PN) and isotopic ($\delta^{13}$C and $\delta^{15}$N) compositions of POM at water depths where the SPM sampled for the present investigation.

*Referee 1*: Then, the explanation for higher $\delta^{13}$C and C/N values in surface sediments that phytoplankton (page 14, lines 10-16) seems out of place. This contribution is about POM, not sediments.

**Reply**: We agree with the comment from Referees 1 and 2, and thus this part has been deleted in the revision.

*Referee 1*: To continue with details that need correction, neither Table 1 nor Table 2 contribute much to the paper as they exist. I suggest either expanding Table 1 as suggested above or deleting it and providing a detailed appendix. The figures are effective, but Figure 3 could be improved by inverting the salinity color code so that salinity (and hence density) increases downward and Figure 8 needs to have the spelling of Redfield corrected the left panel and in the legend.

**Reply**: To understand the range of parameters studied, Table 1 has been retained in the revised version. However, as suggested by the Referee, Table S1 with the hydrographic, elemental and isotopic data is also included in the appendix.

As suggested, Table 2 has been deleted in the revised revision. Figure 3 has been improved; however, we don't know how to invert the salinity colour code using Ocean Data View. The spelling of Redfield has been corrected in Figure 8.

Thank you very much.

**Anonymous Referee #2**

General overview

The manuscript of Liu et al. is focused on characteristics of suspended particulate organic matter (SPOM) in the deep chlorophyll maximum (DCM) of the Eastern China Sea (ECS) during summer 2013. It is based on bulk descriptors of SPOM (C/N and POC/Chl a ratios as well as $\delta^{13}C$ and $\delta^{15}N$). The mains findings are: 1) DCM SPOM mainly originates from in situ primary production, 2) terrestrial POM slightly or insignificantly contributes to DCM SPOM composition, and 3) the latter is contradictory to previous studies but illustrates the drastic decrease in the contribution of the terrestrial POM originating from the Yangtze River to the SPOM composition in the ECS. These findings are sounded and clearly illustrated by the present data set.

The manuscript is well organized and usually well illustrated.

It is of broad audience for scientist who are interested in organic matter cycling and land-to-sea export. It is within the scope of BG. However there are some issues in the present version of the manuscript that preclude the acceptance of the manuscript in its present version. These issues are:

- a lack of information in the methods
- many unneeded details and miscellaneous information that are not needed in the discussion, that rend the discussion too wordy and that dilute the main messages of the study. Authors should focus on what the data indicate, which is usually very clear.
- interpretations of $\delta^{15}N$ data set that are not correct or at least very partial. This data set cannot be published within this manuscript without deeply reconsidering its interpretation and without additional data set regarding N-nutrient (at least nitrate and ammonium) concentrations.

There are also some inconsistencies and language errors that have to be corrected.

Thus, I recommend major revision

**Reply**: Thank you very much for your appreciation on the overall performance of the manuscript and critics on the discussion part and interpretations of $\delta^{15}N$ data.

Detailed comments

*Referee 2*: Section 3: lack of information and details needed
Conversion of fluorescence into chlorophyll a concentration: Since Chl a is a key parameter of the study and is used to calculate POC/Chl a ratio (which values are compared to reference values), it should be explained how in situ fluorescence was converted into Chl a concentration.

**Reply** (**P6, L15-18**): Five SPM samples (DH1-2, DH2-1, DH3-1, DH7-1 and DH7-7; Fig. S1) from water depths ranging between 20 m and 50 m were randomly selected for the measurement of chlorophyll *a* (Chl *a*) concentration. Chlorophyll *a* was extracted using 90%

acetone and then determined spectrophotometrically according to Lorenzen (1967) and Aminot and Rey (2000).

(**P9, L7-13**) Linear correlation between the measured Chl *a* values and the fluorescence values obtained directly from the calibrated sensor attached with the CTD rosette is high with $R^2$ = 0.93 (Fig. S1 in the Supplementary material). This relationship was used to convert the fluorescence values to Chl *a* concentrations of all the remaining SPM using an equation: y = 0.708 x + 0.199, where y is Chl *a* and x is chlorophyll fluorescence concentration. The Chl *a* concentration varied from 0.28 to 3.08 µg $L^{-1}$. The highest value is observed in near coastal station DH5-1, whereas the lowest value is noted in station DH7-9 located off northeast Taiwan. The converted Chl *a* values were used to calculate POC/Chl *a* ratio (Table S1), which is discussed in section 5.2.2.

*Referee 2*: Section 3.1, line 9: indicate the range of depths of the samplings for SPM.

**Reply (P5, L27-30)**: The sentence has been revised as follows:

To investigate the biogeochemical characteristics of POM in the DCM layer of the southern East China Sea, suspended particles around the DCM water depths (10–130 m; Table 1) were collected from thirty-six stations along seven transects across the continental shelf by the Science Cruise during summer (June 22–July 21) 2013 (Fig. 1).

Depth of each station sampled is also listed in Table S1.

*Referee 2*: Section 3.1: indicate how the filters were rinsed right after the filtration

**Reply (P6, L5-7)**: In this study, filters were not rinsed/washed after the filtration. Therefore, the sentence has been revised as follows:

After filtration, filters were folded without rinsing and wrapped again in aluminium foil and then stored at –20 °C immediately in a freezer onboard before they were brought back to the laboratory for further analysis.

*Referee 2*: Section 3.2, line 28: detail how the filters were treated with HCl 1N

**Reply (P6, L20-22)**: Prior to the measurement of POC and PN contents and their stable isotope values ($\delta^{13}C_{POC}$ and $\delta^{15}N_{PN}$) in SPM samples, a half of each filter was placed in a culture dish and 3 ml of 1N HCl was then added into the dish by a dropper and allowed them to react for 16 h to remove inorganic carbon (mainly carbonate).

*Referee 2*: Section 3.2, line 30: indicate the diameter of the punches

**Reply (P6, L23-25)**: The diameter of filter has been included in the revised version as follows: Then a half of de-carbonated filter (i.e. a quarter of the original filter - 11 mm) was then

punched and placed in tin capsules for further analysis.

**Referee 2**: Section 3.2, lines 30-31: it looks like $\delta^{15}N$ and PN were analyzed on the decarbonated part of the filter. Why not on the un-decarbonated part of the filter? There is always chance to bias $\delta^{15}N$ and PN using decarbonated material for these measurements (e.g. Lorrain et al (2003) and other references). Also, it looks like there was a very small part of the filters that were analysed for C and N elemental and isotopic composition. What quantities of C and N were analysed?

**Reply**: We thank the referee to bring the reference Lorrain et al. (2003) to our kind attention.

(**P7, L8-18**) Lorrain et al. (2003) cautioned that the measurement of PN and $\delta^{15}N$ after freezing increases the uncertainty of $\delta^{15}N$ and in combination with concentrated HCl treatment, leads to a loss of PN and alteration of the $\delta^{15}N$ signature. Therefore, PN content and $\delta^{15}N$ values in the current study may have some bias due to de-carbonation. Nonetheless, similar methodological approach has been adopted by Wu et al. (2003) while investigating suspended particles along the *PN* transect in the East China Sea (Fig. 1) and by Hung et al. (1996) while studying the suspended particles in the entire East China Sea. For instance, the range of $\delta^{15}N$ values (~3.8–8.4 ‰) obtained in the present study is comparable to the range of $\delta^{15}N$ values (ca. 0.7–9.4 ‰) obtained by Wu et al. (2003) for the entire water column.

In the present study, the amount of measured C and N ranged from 68.24–322.18 µgC and 14.46–64.69 µgN, respectively (Table S1). Precision for $\delta^{13}C$ and $\delta^{15}N$ decreases for samples containing less than 100 µgC and 20 µgN, respectively. Among thirty-six filters analyzed for the present study, only five (three) filters contain less than 100 µgC (20 µgN).

Influence of CDW at DCM depth in the ECS

**Referee 2**: It cannot be stated that the influence of CDW in the study site is nil or insignificant. The low salinity measured at some of the sites (Fig. 3) clearly indicates the influence of CDW. It is mainly the case in surface water but also the case at some of the DCM depths where water was sampled for SPM (stations DH1-1, DH2-1, DH2-2, DH3-1 and CON02). This is also clear from Fig. 6b where five stations falls within the SMW square, SMW being clearly a water body composed of a mixing between KSSW and CDW.

It should better be written that the influence of CDW in the study site is low (see some of the 'minor points' below) or weak (as written in section 5.1, P8, line 34).

**Reply**: As suggested by the Referee, we have softened the tone of the influence of CDW shown in Fig. 3 and Fig. 6b in the revised text.

End of section 5.1 (P9, lines 1-9) and Fig. 7
**Referee 2**: Only the DCM depths (= the depths of interest for the present study) should be considered for delineate the polygons of Fig. 7. Was it the case? For similar reason, I think that

the sentence "Interestingly...study area (Fig. 7)" should be deleted or reworded without citing Fig. 7 (but rather Fig. 3?) since it is quite confusing. Another option may be to not cite depth limitation of the water masses influences but only describe Fig. 7.

**Reply**: We disagree with this suggestion mainly because the conditions of how different water masses (CDW, TWCW and KSSW) are influencing the DCM depths are shown in Fig. 6b. Further, in Fig. 7 we delineated areas influenced by three water masses both horizontally and vertically for the entire water column. Furthermore, the water masses were delineated based on the *T-S* combination and therefore, citing Fig. 3 along with Fig. 7 is fine, but deleting the mention of Fig.7 without water depths may mislead the meaning. Therefore, based on Referee's suggestions, we revised the first paragraph of section 5.1 as follows (**P11, L6-24**):

**5.1 Influence of different water masses in the southern ECS**

In order to identify the different water sources in the study area, temperature–salinity (*T–S*) diagrams were drawn for the entire water column (Fig. 6a) as well as for the SPM sampling depth around DCM layers (Fig. 6b). The *T–S* diagram for all the water depths shows a convergence at around 17 ºC, 34.6 (Fig. 6a), representing the upwelling of KSSW (Umezawa et al., 2014). There are two trends in the *T–S* diagram, indicating a mixing of three water masses: one is less saline and much colder water, mainly CDW, another is more saline and warmer, mainly Taiwan Warm Current Water (TWCW), and the third one is KSSW (Fig. 6a). The shelf water in the entire ECS in summer 2013 was mixed primarily by three water masses, CDW, KSSW, and TWCW (Fig. 6a). The low salinity observed at five coastal sites (DH1-1, DH2-1, DH2-2, DH3-1 and CON02) indicates the influence of CDW mostly in surface water, but also some of the DCM depths where water was sampled for SPM (Fig. 2). This is also evident from Fig. 6b where these five stations fall within the area of SMW, which is a water body composed of a mixing between CDW and KSSW. However, except these five coastal stations, most DCM depths where water was sampled for SPM seem to be weakly influenced by the CDW (Fig. 6b). Based on the *T–S* range of different water masses (Fig. 6), we further delineated the area influenced, along with water depths by three important water masses: CDW, TWCW and KSSW (Fig. 7). Interestingly, the influence of CDW was constrained only in the upper 0–10 m in five coastal stations during the sampling time, whereas TWCW influences around 0–30 m, covering three-fourths of the study area, and KSSW seems to be largely influenced the bottom water of the entire study area (Figs. 2, 6a and 7).

*Referee 2*: The last paragraph of section 5.1 is also quite confusing. Reword it as: "In summary, although the river runoff was huge, the influence of CDW plume in the southern part of the ECS was insignificant during summer 2013, mainly because most of the CDW plume was transported to northeastwardly of the Yangtze estuary to the Korean coast (Isobe et al., 2004; Bai et al., 2014; Gao et al., 2014). This contrasts with summer 2003 when the plume front moved southward (Bai et al., 2014). Meanwhile, the intrusion of TWCW and KSSW was strong in the continental shelf of the East China Sea during summer 2013."

**Reply (P11, L26-31)**: The last paragraph of section 5.1 has been reworded exactly, as

suggested above by the Referee.

Section 5.2.1
*Referee 2*: I fully agree the main conclusions of this section and most of data interpretations (especially the first and the last paragraphs).

However the second paragraph adds detailed discussion with literature comparison that is not needed (especially when dealing with zooplankton and Trichodesmium) for the present study. Authors should better goes directly to the conclusion (i.e. the last paragraph) without diluting the main conclusions with unneeded wording. Thus, the second paragraph should be deleted.

**Reply (From P12, L23 to P13, L2)**: As suggested by the Referee, the second paragraph of section 5.2.1 has been deleted in the revised version.

Section 5.2.2
Referee 2: As for the previous section, I fully agree the main conclusions and most of the data interpretation, but this section is too wordy and gives too many details (especially too many values from the literature). Authors should better focus on the main information and the main conclusions.

Thus, I suggest the following:

*Referee 2*: - P10, lines 26-34: one-two sentence(s) should be enough

**Reply (P13, L15-25)**: Lines 26-34 have been shortened/revised as follows:

Moreover, the POC/Chl *a* ratio of 34.1 g g$^{-1}$ derived from the slope of a regression line (y = 34.1 ($\pm$9.99) $\times$ +49.9 ($\pm$8.86) (Fig. 8b) is consistent with the reported POC/Chl *a* ratios in the ECS (36.1 g g$^{-1}$; Chang et al., 2003) and the North-western Pacific (48 g g$^{-1}$; Furuya, 1990). However, the POC/Chl *a* ratio obtained in this study is lower than that estimated (64 g g$^{-1}$) for the sinking particles in the ECS and the Kuroshio region, off northeast Taiwan Island (Hung et al., 2013). The range is well within the range (13–93 g g$^{-1}$) reported by Chang et al. (2003) in the ECS and estimated (18–94 g g$^{-1}$) from phytoplankton cell volumes by the same authors.

*Referee 2*: - P11, lines 2-4: keep this sentence but rephrase the last line as "filtered particles (Chang et al., 2003; Hung et al., 2013)"

**Reply (P13, L25-29)**: As suggested, the sentence has been rephrased as follows:

Although the Chl *a* concentration in our study was converted based on the linear relationship between measured Chl *a* and *in situ* fluorescence values (see Section 3.2 and Fig. S1 for more details), it is more or less similar to Chl *a* concentrations obtained in the above-mentioned studies, which were mostly extracted from filtered particles (Chang et al., 2003; Hung et al., 2013).

*Referee 2*: - P11, lines 8-10: do not report all these values

**Reply (From P13, L33 to P14, L2)**: These lines have been modified in the revised version as follows:

The POC/Chl *a* ratio of living phytoplankton was reported to be between 40 and 140 g g$^{-1}$ (Geider, 1987; Thompson et al. 1992; Montagnes et al. 1994; Head et al. 1996).

*Referee 2*: Regarding the high POC/Chl a ratio, did authors check if these high values were rather due to very low Chl a concentration or high POC concentration? If the former, these high values may be associated to high uncertainty on the Chl a estimation when values are low. If the latter (high POC concentration associated to Chl a concentration similar to surrounding stations), this may be effectively due to heterotrophic biomass.

**Reply**: When we use the converted Chl *a* concentration in the revised version, only one SPM shows high POC/Chl *a* ratio of >200 g g$^{-1}$ (CON02: 303 g g$^{-1}$). Although it contains neither high POC concentration (CON02: 92.6 µg L$^{-1}$) nor high Chl *a* content (CON02: 0.15 µg L$^{-1}$), the Chl *a* value seems to be relatively low, as shown in Fig. 8b.

Section 5.3: first three paragraphs
*Referee 2*: I fully agree the main conclusions and the data interpretations of the first three paragraph of this section.

I suggest authors to have a look at Lowe et al. (2014) and Miller et al (2013): these articles are of interest for the present section.

**Reply**: These two references are included in the revised version in appropriate places. Please refer to **P15, L11-22** in the marked-up manuscript for details.

Page 12, Lines 18-26: two other processes may influence phytoplankton δ$^{13}$C: temperature and degradation. This is discussed in Savoye et al. (2003) that authors cite in many occurences. Authors may have a look at biplots like δ$^{13}$C vs temperature and versus POC/Chl *a* and C/N (considering these ratios may also indicate phytoplankton decay). They also may check the normalization of δ$^{13}$C by temperature (as in Savoye et al., 2003) before plotting normalized δ$^{13}$C versus POC, since temperature usually have (indirect) influence on phytoplankton δ$^{13}$C.

**Reply (From P17, L24 to P18, L24)**: As directed, a section 5.4 on "Temperature effect on δ$^{13}$C$_{POC}$ data" has been included in the revised version as follows:

**5.4 Temperature effect on the δ$^{13}$C$_{POC}$ around the DCM layer**

Apart from primary production and the growth rate and species composition, temperature and biomass degradation may influence the carbon isotopic composition of phytoplankton (Savoye

et al., 2003). Temperature has an indirect effect on isotopic fractionation between phytoplankton carbon and dissolved $CO_2$, and therefore on phytoplankton $\delta^{13}C$ (e.g., Rau et al., 1992; Savoye et al., 2003). The C/N ratio, POC/Chl $a$ ratio and $\delta^{13}C_{POC}$ all indicated that the POM around the DCM layer is dominated by newly-produced phytoplankton OM (see Sections 5.1–5.3). Therefore, to understand the temperature effect on $\delta^{13}C$ of phytoplankton, we plotted our $\delta^{13}C_{POC}$ data against temperature into two groups by separating approximately at ~24°C (Fig. 11a). Data points of both groups show a decreasing $\delta^{13}C$ of phytoplankton biomass while increasing temperature around the water depths of DCM in the southern ECS (Fig. 11a). Such a relationship is in contrast to the positive relationship between these two variables observed for the surface ocean around the world (Sackett et al., 1965; Fontugne, 1983; Fontugne and Duplessy, 1981).

The negative relationship between $\delta^{13}C_{POC}$ and temperature is likely related to biological activity and carbonate dissolution equilibrium, both may control the concentration of dissolved inorganic carbon in the DCM layers, which are closer to euphotic depths (see Section 4.1). The weak correlation between $\delta^{13}C_{POC}$ and temperature supports a weak influence of temperature on $\delta^{13}C_{POC}$ around DCM layers in the study area (Fig. 11a). A decrease in fractionation of approximately –0.56‰ °C$^{-1}$ is estimated for POM collected at <24°C, whereas a decrease in fractionation of roughly –0.51 °C$^{-1}$ is estimated for POM collected at >24°C (Fig. 11a). In order to distinguish the influence of biological parameters from temperature on $\delta^{13}C_{POC}$, the $\delta^{13}C_{POC}$ data were corrected for the 'temperature effect' by normalizing the data using an equation: $\delta^{13}C_{POC} = f(T)$.

In the present study, since most $\delta^{13}C_{POC}$ values come from the DCM layer and the $\delta^{13}C_{POC}$ is negatively correlated with temperature (Fig. 11a), we applied our own temperature coefficients (–0.56‰ °C$^{-1}$ and –0.51‰ °C$^{-1}$) and $\delta^{13}C_{POC}$ was normalized at 24°C (i.e. the mean temperature at sampled water depths) using the formula (Savoye et al., 2003): $\delta^{13}C_{24°C} = \delta^{13}C_{POC} - s(T - 24)$, where $\delta^{13}C_{24°C}$ is the temperature-normalized $\delta^{13}C_{POC}$, T is the seawater temperature in °C from water depths where SPM sampled, and s is the slope of the linear regression $\delta^{13}C_{POC} = f(T)$ in ‰ °C$^{-1}$ obtained from Fig. 11a. There are significant correlations between $\delta^{13}C_{24°C}$ of biomass and POC concentration (circles: $R^2 = 0.71$; $p<0.0001$; n = 18 and triangles: $R^2 = 0.66$; $p<0.0001$; n = 18; Fig. 11b), indicating that primary production drives ~70% of the variation of phytoplankton $\delta^{13}C$ around DCM layers in the southern ECS. On the other hand, $\delta^{13}C_{24°C}$ correlated insignificantly with POC/Chl $a$ ratio and C/N ratio (Figs. 11c and 11d), implying that degradation has a minor effect on the carbon isotopic composition of POM in this study.

Section 5.3: last three paragraphs

**Referee 2**: I do not think that the last three paragraph of the section are needed. The objective of the two paragraphs before the last (from "The nutrient N/P ratio" to "this mechanism is most likely") is to decipher whether POM sampled in the DCM came from in situ production or from surface production (cf. the fourth paragraph of the section). In fact, these two paragraphs lay on very putative argumentation and do not allow (and are not convincing in) deciphering

between the two hypotheses. These hypotheses have already been discussed in sections 5.2.1 and 5.2.2 with sufficient argumentation for considering that POM mainly came from in situ production. To my point of view, these two paragraphs of section 5.3 are not needed in this section neither in the manuscript.

**Reply (From P16, L17 to P17, L13)**: As suggested, these two paragraphs have been deleted in the revised version.

*Referee 2*: The last paragraph of the section is a tentative of inventory of POC in the DCM layer. The estimation is very rough, is associated to large uncertainty, and the calculation is not convincing. It is also completely disconnected from the rest of the section, which is focused on $\delta^{13}C$ dynamics (see the title of the section). Again, this paragraph is not needed in this section neither in the manuscript. Thus, the last three paragraphs of the section should simply be deleted and the fourth paragraph of the section ("The range... DCM layers?") should be replaced with a brief conclusion of the first three paragraphs of the section.

**Reply (P17, L15-21)**: We agree that the POC inventory in the DCM layer has been done approximately and therefore can be deleted in the revised version.

*Referee 2*: Section 5.3 would better stand with the first three paragraphs and a conclusion without the unclear and unconvincing last three paragraphs.

**Reply**: As suggested, all three paragraphs have been deleted in the revised version.

Section 5.4
*Referee 2*: This section is the less clear and the less convincing of the manuscript. The main conclusion (POM $\delta^{15}N$ distribution is primarily governed by the nutrient status and $\delta^{15}N$ of nitrate) is mainly guess. One of the main issues of the section is the lack of nutrient data. This rend the data interpretation mainly guess-work. Another issue is that authors mainly take into account nitrate as a nutrient for phytoplankton. Ammonium appears only in the last paragraph. The other species of N-nutrient ($N_2$, dissolved organic nitrogen as urea) are not mentioned. However, it is reported that "Kuroshio Water and TWCW induced Trichodesmium" (P10, line 7). Thus, PN $\delta^{15}N$ values should also be discussed considering $N_2$-fixers (diazotrophs). At last, many sentences are not clear. This gives the impression that authors do not fully have in mind what processes drives PN and phytoplankton $\delta^{15}N$.

Thus, this section should be deeply reworked including deep data re-interpretation. To me, such section dedicated to PN/phytoplankton $\delta^{15}N$ cannot stand without data of nutrient concentration originating from the same cruise. If these data are not available, PN/phytoplankton $\delta^{15}N$ cannot be discussed. If it would be the case, $\delta^{15}N$ data should be removed from the manuscript.

**Reply**: Thank you for the suggestion! At present, we don't have depth profiles of nitrate or ammonia or nitrogen isotopic composition of nitrate to strengthen the PN/$\delta^{15}N$ data and related

interpretations. To our knowledge, all our interpretations of PN/$\delta^{15}$N are based on our understanding of nitrogen dynamics in the ECS and are consistent with the literature cited. This part is also appreciated by Referee 3, but also suggested to strengthen the interpretations using depth profiles of nutrients. However, there is no much information related to $\delta^{15}$N data, especially from the biota-dominated DCM layers, in the East China Sea or South China Sea. Since we proved that the POM is dominated by phytoplankton, publishing $\delta^{15}$N data of this study may fill the data gap and also create some awareness/interests among readers to conduct such investigations in the marginal seas of the western Pacific in detail. Given this, we request anonymous reviewers and Associate Editor to allow this section for publication because we interpreted PN/ $\delta^{15}$N data based on the published information from the East China Sea with some speculations, a practice that is normally encouraged in the scientific field when the availability of data is relatively scarce, such as $\delta^{15}$N of phytoplankton.

Section 5.5

**Referee 2**: This section gives an important conclusion: the influence of terrestrial POM (mainly originating from the Yangtze River) has drastically decreased in the ECS. This section is mainly based on literature data and conclusions. These inputs from the literature are of interest, but the section should also compare data from the present study with previous data. Thus, this section should start with the comparison of POC/Chl a and C/N ratios, and $\delta^{13}$C values between the present study and previous studies. Then, the decrease of terrestrial POC fluxes and deposition can be cited (literature data). Last lines of the section: there are again unneeded details in these lines. Avoid describing the degradation index but directly give the conclusions of Wu et al (2007b).

**Reply (From P20, L1 to P21, L4)**: As directed, section 5.5 has been revised as follows:

The range of POC/Chl *a* obtained in this study (33–200 g g$^{-1}$) is within the range (<200 g g$^{-1}$) reported for the phytoplankton-dominated POM in the coastal and shelf waters (e.g., Chang et al., 2003; Savoye et al., 2003; Hung et al., 2013; Liénart et al., 2016). We also obtained a narrow range of C/N ratio (4.1–6.3), but a wide range of $\delta^{13}C_{POC}$ (–25.8 to –18.2 ‰) compared to previous studies in the ECS (4.0–34.3, Liu et al., 1998; –24.0 to –19.8 ‰, Wu et al., 2003). These results indicated that POM around the water depths of DCM was largely derived from the synthesis of *in situ* phytoplankton and the influence of terrestrial OM supplied by the Yangtze River to the southern ECS is low. The missing of terrestrial OM signals seems to be related to reservoir and dam buildings along the river in recent years that has shifted the location of the Yangtze-derived POC deposition from the inner shelf of the ECS to terrestrial reservoirs (Li et al., 2015). The sediment delivered from the river to the estuary has reduced by 40 % since 2003 when the Three Gorges Dam (TGD) was completed (Yang et al., 2011 and references therein). Recently, Dai et al. (2014) reported that the particles discharged by the Yangtze has declined to 150 Mt yr$^{-1}$, less than ~70% of its sediment delivery to the ECS during 1950s. Although 87 % of the mean annual sediment of Yangtze River is discharged during the flood season from June to September (Wang et al., 2007; Zhu et al., 2011), approximately 60 out of 87% of the fine-grained sediments are temporarily deposited near the estuary and then later resuspended and transported southward along the inner shelf, off the mainland China

(Chen et al., 2017 and references therein). The Yangtze-transported POM moves up toward the northeast across the shelf along the so called the Changjiang transport pathway in summer season (e.g., Gao et al., 2014), which is largely affected by the combined effects of high river discharge, southwest summer monsoon and the intensified TWC (Beardsley et al., 1985; Ichikawa and Beardsley, 2002; Lee and Chao, 2003). The *T–S* diagrams (Figs. 6 and 7) of this study also illustrate this view.

Accompanying with the decreasing sediment input, dam building in the Yangtze River basin since 2003 has buried around 4.9±1.9 Mt yr$^{-1}$ biospheric POC, approximately 10% of the world riverine POC burial flux to the oceans (Li et al., 2015). The POC flux from the Yangtze to the ECS (range: 1.27–8.5 × 10$^{12}$ g C yr$^{-1}$; Wang et al., 1989; Qi et al., 2014) was significantly less than the estimated primary productivity (72.5 × 10$^{12}$ g C yr$^{-1}$; Gong et al., 2003), implying the predominance of marine-sourced organic matter in the ECS. Moreover, the substantial quantity of organic substances that transported by the Yangtze River may be completely modified before being ultimately deposited on the inner shelf of the ECS and being transported further offshore (Katoh et al., 2000; Lie et al., 2003; Chen et al., 2008; Isobe and Matsuno, 2008). Wu et al. (2007b), for instance, observed an advanced stage of POM degradation in the entire Yangtze River with an average degradation index of –1.1. Based on the investigation of lipid biomarkers in a sediment core collected from the ECS, Wang et al. (2016) suggested the dominant preservation of marine autochthonous organic matter (~90 %) in the ECS.

English language
*Referee 2*: The language is usually quite understandable, but there are many errors or mistakes.
Part of them is listed in the 'minor points' below. Nevertheless, the whole manuscript should be deeply checked for English language.

**Reply**: The whole manuscript has been carefully checked for grammatical errors.

Inconsistencies
*Referee 2*: There are some inconsistencies between values that are cited in the text and values reported in the tables (see 'minor points' below). Please, check the consistency between all the values reported in the text and tables.

**Reply**: Cross-checked and corrected.

Abstract, Introduction and Summary and conclusions
*Referee 2*: Sections 'Abstract' and 'Summary and conclusions' should partly be re-written taking into account the above detailed comments.

**Reply**: As suggested, these parts have been modified as follows (**Abstract: From P1 to P2, 17; Summary and conclusions: From P21, L8 to P22, L2**):

[revised manuscript text omitted]

*Referee 2*: The third objective that appear in section Introduction should be removed (see one of my comments dedicated to section 5.3 ă˘A˘T three last paragraphs ă˘A˘T above).

**Reply (P4, L15-16)**: The third objective in the Introduction part has been deleted in the revision.

Minor points

- P1, Line 20: what do you mean with 'straddling'?

**Reply**: It means locating or moving around DCM depth intervals.

- P2, line 9 and in the whole manuscript: replace 'endmember' with 'end-member'

**Reply**: Replaced.

- P3, line 8: remove 'which in turn, the elemental and isotopic compositions of marine productivity' since it is not correct

**Reply (P3, L23-24)**: Deleted.

- P5, line 1: depending what you want to say, add 'by' or 'to' between 'decreased' and '86%'

**Reply (P5, L20)**: As suggested, 'by' is added.

- P5, line 17: replace 'had' with 'have'

**Reply (P6, L3)**: Replaced.

- Last sentence of section 3.1: place this sentence in section 3.2 since it is not sample collection. Rename section 3.2 as 'Determination of SPM concentration and analysis of POC, PN, $\delta^{13}C$ and $\delta^{15}N$

**Reply (P6, L7-15)**: The last sentence of section 3.1 has been shifted to section 3.2. As suggested by the Referee, section 3.2 has also been renamed in the revised version.

- P5, line 30: replace 'with' with 'and placed in'

**Reply (P6, L24)**: Replaced.

- Section 3.2, P5-6, sentence "organic carbon and nitrogen ... entering the IRMS": remove the sentence since such level of detail is not needed

**Reply P6, L28-30)**: The sentence has been deleted in the revised version.

- Section 3.2: remove the three last sentences ("Conventional...Sigman et al., 2009" since the first one is unneeded detail and the two last ones do not stand in a section dedicated to methods.

**Reply (P7, L4-8)**: As suggested, the last three sentences in section 3.2 have been deleted in the revised version.

- P6, line 22: add 'usually' between 'profiles of Chl a' and 'show'.

**Reply (P8, L28)**: Added and modified.

- Section 4.1.3: since Fig. 3 illustrates at maximum the first 300m of the water column and since the sampling depth was within this depth interval, please do not describe deeper water, either the reading is quite disturbing. Thus, the temperature ranged between 30 and ca. 15 °C.

**Reply (P7, L27-33)**: We agree with the Referee's view and therefore the paragraph describing the range of temperature has been revised as follows:

Figure 2 illustrates the vertical distributions of temperature and salinity along seven transects across the ECS. In the entire study area, temperature in the 300-m water column varied from 15 °C to 30 °C and distinct water column stratification was evident from the temperature profiles (Fig. 2). The temperature decreases when depth increases and the highest temperature (~30 °C) seen mostly in the surface water and the lowest temperature (5 °C) was observed in stations DH7–8 and DH7–9 at water depths of 850 m and 800 m, respectively (not shown). Temperature at sampling depths of SPM ranged from 19.1 °C to 28.2 °C, showing a general decreasing trend from the inner to outer shelf in each transect (Fig. 2).

- P7, line 16: replace 'increasing' with 'increases'; reword "with the high temperature (>30 °C) spreads widely".

**Reply**: Replaced. Please see our reply to the previous comment.

- P7, line 29: replace 'insignificant' with 'low'.

**Reply (P8, L9)**: Replaced.

- P8, line 4: '4.4' or '4.5' as reported in table 1?

**Reply (P33)**: The correct value is 4.4 and it is corrected in Table 1.

- P8, line 5: '17.7' or '17.8'?

**Reply (P33)**: The correct value is 17.7 and Table 1 is corrected accordingly.

- P8, lines 7-8: please also indicate where the highest POC and PN concentration were located.

**Reply (P10, L19-20)**: The following sentence has been included in the revised version:

The highest concentrations of POC (263 µg L$^{-1}$) and PN (52.8 µg L$^{-1}$) are associated with station DH5-1 (Fig. 4).

- P8, line 17: '8.0' or '7.8' as reported in Table 1?

**Reply (P33)**: The correct value is 8.0 and Table 1 is revised accordingly.

- P8, lines 17-18: please also indicate where the highest $\delta^{13}C$ values were located.

**Reply (P10, L30-31)**: The following sentence has been included in the revised version:

Consistent to the POC concentration, the highest $\delta^{13}C_{POC}$ value (−18.2 ‰) is also associated with station DH5-1.

- P8, line 21: Fig. 10 is cited before Fig. 5. Check the numbering of the figures.

**Reply**: Fig. 10 is cited after Fig.5, which was cited just above in the text. We have cross-checked all figure numbers in the revised version.

- P8, line 33: SMW is a water body that is composed of a mixture between two other water masses (CDW and KSSW; Fig. 6). So, do not consider SMW as a water mass and remove it from this list.

**Reply (P11, L14)**: Removed.

- P10, line 25: remove the word 'moderate' since this information is not useful here.

**Reply (P13, L13)**: Deleted.

- P10, line 29: '48' or '52' as reported in Table 2?

**Reply (P13, L18)**: According to Table 2, the value 48 is for the northwestern Pacific and the number 52 is for the western Pacific. As suggested by the referees, Table 2 has been deleted in the revised version.

- P11, line 28: replace 'less' with 'low'; delete "and unrecognized content of terrestrial POM".

**Reply (P14, L21-22)**: Replaced and deleted.

- P12, 2: replace 'to be' with 'would be'.

**Reply (P14, L28)**: Replaced.

- P12, line 8: replace 'more positive' with 'less negative'.

**Reply (P14, L34)**: Replaced.

- P12, line 34: delete 'As for species,'.

**Reply (P15, L30)**: Deleted.

- P13, line 7: delete 'that'.

**Reply (P16, L3)**: Deleted.

- P13, line 8, 9 and 10: replace 'larger' with 'higher'

**Reply (P16, L4)**: Replaced.

- P13, lines 9 and 10: replace 'size species' with 'phytoplankton'

**Reply (P16, L5)**: Replaced.

- P15, line 33: replace 'significantly less' with 'low'.

**Reply (P20, L8)**: Replaced.

- P16, line 12: replace 'proved' with 'illustrate'.

**Reply (P20, L23)**: Replaced.

- Table 1: add POC/Chl a values in this table; indicate in the caption what means 'SD'.

**Reply (P33)**: POC/Chl a values have been included in Table 1 with SD abbreviation.

- Figure 1: indicate KSSW on the figure; be consistent with Fig. 6. Indicate on this figure the location of the stations that appear on Fig. 2 and 3 but were not sampled for SPM in the DCM.

**Reply**: Fig. 1 shows the simplified current pattern in the ECS, and the center of the upwelling region. As one of the water masses, it is appropriate to show KSSW in Figures 6 and 7 along with other water masses.

**Reply**: Stations where SPM were not sampled at the deep chlorophyll maximum layer are marked in Fig. 1 in the revised version (red circles in Fig. 1).

Figure 6: the two colours are not distinguishable. Choose other colours. Remove 'from' in the second line of the caption. Add 'were' after 'matters' in the third line of the caption.

**Reply**: The two colours in Figure 6 are changed. Other corrections are included, as suggested.

Figure 7: replace 'black' with 'grey' in the second line of the caption.

**Reply**: Replaced.

Figure 8: first line of the caption: it is POC vs. PN and POC vs. Chl a.

**Reply**: Corrected.

Additional references

Lorrain A., N. Savoye, L. Chauvaud, Y-M. Paulet and N. Naulet, 2003. Decarbonation and preservation method for the analysis of organic C and N contents and stable isotope ratios of low-carbonated suspended particulate materiel. Analytica Chimica Acta, 491, 125-133.
Lowe, A. T., A. W. E. Galloway, J. S. Yeung, M. N. Dethier, and D. O. Duggins. 2014. Broad sampling and diverse biomarkers allow characterization of nearshore particulate organic matter. Oikos. 123: 1341–1354, doi:10.1111/oik.01392
Miller, R. J., H. M. Page, and M. A. Brzezinski. 2013. _13C and _15N of particulate organic matter in the Santa Barbara Channel: drivers and implications for trophic inference. Mar. Ecol. Prog. Ser. 474:53-66, doi:10.3354/meps10098

**Reply**: These additional references are included in the revised version.

Thank you very much.

**Anonymous Referee #3**

General Comment: This manuscript characterized the bulk and isotopic composition of organic matter collected in DCM layer of the south East China Sea in summer time. The study is well designed and neatly presented. It observed the marine derived material is the dominant organic matter in DCM layer. The influence of the Yangtze River is very limited. Additionally, the nitrogen isotopes elucidated the potential role of N2 fixing in middle shelf where TWCW is dominated and remineralized nutrients plays an important role. However, in this point, the depth profile will be very helpful to strengthen the discussion but it is lack in the manuscript. Furthermore, the influence of lateral transport (cross shelf) better to be considered when the authors estimate the nutrients contributions from different sources and POC inventory.

**Reply**: Thank you for your positive opinions on the overall work presented here. At present, we don't have depth profiles of nitrate or ammonia or nitrogen isotopic composition of nitrate to strengthen the PN/$\delta^{15}$N data and related interpretations. However, we reiterate that there is no much information related to $\delta^{15}$N data, especially from the biota-dominated DCM layer in the southern East China Sea. Since we proved that the POM is dominated by phytoplankton, publishing $\delta^{15}$N data of this study may fill the data gap and also create more interests among readers to conduct such investigations in the marginal seas of the western Pacific. Given this, we request anonymous reviewers and Associate Editor to allow this section for publication because we interpreted PN/ $\delta^{15}$N data based on the published information from the southern East China Sea.

Specific comments:

*Referee 3*: 1) The title better be more specific in study region, such as southern East China Sea

**Reply**: The title of the manuscript has been modified as follows:

Biogeochemical characteristics of suspended particulates at deep chlorophyll maximum layers in the southern East China Sea

*Referee 3*: 2) Abstract: OK

**Reply**: Thank you.

*Referee 3*: 3) Introduction: Better to emphasize the status of DCM in ECS and the potential role in POC inventory estimation and hypothesis

**Reply**: The text related to the POC inventory has been deleted in the revised version, as suggested by the Referee 2.

*Referee 3*: 4) Methods P5, the filtration volume was in the range of 0.5-2L, the author used

half filter for POC or PN analysis, Did they have enough material for reliable analysis, especially for nitrogen?

**Reply**: We have mistakenly mentioned the filtration volume in the original manuscript. Our apologies! We filtered 4.1-19.1 L of water samples for the collection of SPM. The volume of filtration has been changed accordingly in the revised version as follows **(P5, L34 to P6, L3)**:

The volume of each water sample was measured by graduated cylinder before filtration. Suspended particles were obtained by filtering 4.1–19.1 L seawater collected at the fluorescence maximum layer through 0.7 µm/47 mm Whatman Glass Fiber Filters (GF/F), which were wrapped in aluminium foil.

In our study, the amount of measured C and N ranged from 68.24-322.18 µg and 14.46-64.69 µg, respectively. Precision for $\delta^{13}C$ and $\delta^{15}N$ decreases for samples containing less than 100 µgC and 20 µgN, respectively. Among thirty-six filters analyzed for the present study, only five (three) filters contain less than 100 µgC (20 µgN).

*Referee 3*: 5) Result and interpretations: The order of hydrographic characteristics can be adjusted as salinity, turbidity and chla

**Reply (From P7, L25 to P9, L5)**: As suggested, we rearranged the order of hydrographic characteristics in the revised version.

6) Discussion: In general, the authors gave proper credit to related work and clearly indicate their own new contribution to the biogeochemical cycles in the study area.

**Reply**: Thank you.

Some minor suggestions:

*Referee 3*: a. P13-14, How to use C/P ratio to estimate the Yangtze-sourced nutrients for marine primary productivity, how does the lateral cross shelf transport contribute to the POC inventory?

**Reply (From P16 to P17, L21)**: The text related to the C/P ratio and POC inventory have been deleted in the revised version, as suggested by the Referee 2.

*Referee 3*: b. P15, L20, this paragraph is a bit speculative and need more straightforward data to support itself, the depth profile data could be bit helpful to elucidate.

**Reply**: Please refer to our reply to the general comment of Referee 3.

*Referee 3*: c. Comments on 5.5, why the Yangtze River will play an important role in DCM OM in south ECS, the paragraph of TGD can be moved which seems less related to this topic, also

I am confused about how the author summarized in the abstract" SPM investigated here seems not to be influenced by the terrestrial organic matter supplied by the Yangtze River (Changjiang) in summer 2013, a finding that is contrary to a number of previous studies' conclusion.", which is not convinced in this part.

**Reply**: The transport pathway of Yangtze River debouched sediments to the Okinawa Trough is one of the unsettled issues in the oceanographic studies of the East China Sea. Recently, Chen et al. (2017) suggested that the Changjiang river plume flows southward when sediments are resuspended along the China coast by cyclonic storms. Most previous studies also had shown that cross-shelf transport of Yangtze-derived sediments to the Okinawa Trough. So, one would expect the influence of Yangtze-derived sediments in the southern ECS during summer.

The part of the sentence "a finding that is contrary to a number of previous studies' conclusion" in the abstract has been deleted in the revised version (**P2, L10-11 in the marked-up manuscript**).

*Referee 3*: d. The quality of Figure 4 and 5 should be improved. Figure 8b, there is 35 samples summarized but all the other plotted based on 36 samples, why?

**Reply**: The quality of Figures 4 and 5 has been improved. There is no fluorescence/Chl *a* data for station DH6-1 (see Table S1 in the appendix) and therefore plots with Chl *a* data contain only 35 data points, including Fig. 8b.

Thank you very much.

Additional References

Aminot, A. and Rey, F.: Standard procedure for the determination of chlorophyll *a* by spectroscopic methods, ICES Techniques in Marine Environmental Sciences, Copenhagen, Denmark, 8–11, 2000.

Cullen, J. J., Reid, F. M. H., and Stewart, E.: Phytoplankton in the surface and chlorophyll maximum off southern California in August, 1978, J. Plank. Res., 4, 665–694, 1982.

Fontugne, M. R.: Les isotopes stables du carbone organique dans l'océan: application à la paléoclimatologie, PhD thesis, Université de Paris XI, 1983.

Fontugne, M. R. and Duplessy, J. -C.: Organic carbon isotopic fractionation by marine plankton in the temperature range –1 to 31°C, Oceanol Acta, 4, 85–90, 1981.

Lorrain A., Savoye, N., Chauvaud, L., Paulet Y. -M., and Naulet, N.: Decarbonation and preservation method for the analysis of organic C and N contents and stable isotope ratios of low-carbonated suspended particulate material. Anal. Chim. Acta, 491, 125–133, 2003.

Lorenzen, C. J.: Determination of chlorophyll and pheo-pigments: Spectrophotometric equations, Limnol. Oceangr., 12, 343–346, 1967.

Lowe, A. T., Galloway, A. W. E., Yeung, J. S., Dethier, M. N., and Duggins, D. O.: Broad sampling and diverse biomarkers allow characterization of nearshore particulate organic matter, Oikos, 123, 1341–1354, 2014.

Malone, T. C., Falkowski, P. G., Hopkins, T. S., Rowe, G. T., and Whitledge, T. E.: Mesoscale response of diatom populations to a wind event in the plume of the Hudson River, Deep-Sea Res., 30, 149–170, 1983.

Miller, R. J., Page, H. M., and Brzezinski, M. A.: $\delta^{13}$C and $\delta^{15}$N of particulate organic matter in the Santa Barbara Channel: drivers and implications for trophic inference, Mar. Ecol. Prog. Ser., 474, 53–66, 2013.

Rau, G. H., Takahashi, T., Des Marais, D. J., Repeta, D. J., and Martin, H.: The relationship between $\delta^{13}$C of organic matter and [$CO_2(aq)$] in ocean surface water: data from a JGOFS site in the northeast Atlantic Ocean and a model, Geochim. Cosmochim. Acta, 56, 1413–1419, 1992.

Sackett, W. M., Eckelmann, W. R., Bender, M. L., and Bé, A. W. H.: Temperature dependence of carbon isotope composition in marine plankton and sediments, Science, 148, 235–237, 1965.

Wong, W. W. and Sackett, W. M.: Fractionation of stable carbon isotopes by marine phytoplankton, Geochim. Cosmochim. Acta 42, 1809–1815, 1978.

[revised manuscript text omitted]

---

## Referee Report (RR1)

**MS Ref: bg-2017-290**

**General overview:**

Liu et al. present isotopic compositions and concentrations of particulate organic carbon and particulate nitrogen (POC and PON, respectively) in samples collected from the deep chlorophyll maximum (DCM) in the south East China Sea in summer 2013. They combine these data sets with temperature, salinity, turbidity and calibrated chlorophyll fluorescence data to determine the sources of particulate organic matter in the DCM and what factors govern isotopic dynamics of POC and PON in the DCM. The authors attribute variation in $\delta^{13}C_{POC}$ to be governed by changes in primary productivity and community composition and variation in $\delta^{15}N_{PN}$ to be governed by changes in uptake of dissolved inorganic nitrogen (i.e. $NH_4^+$ or $NO_3^-$) and source (with a link to water masses). The handling and interpretation of isotopic data is good, however with a lack of certain auxiliary data sets (e.g. dissolved inorganic nutrients, community composition), and full methodological detail in the determination of others (chlorophyll), it is hard to fully critique their interpretation of the data and their conclusions. Following revision, this study could contribute to our understanding of organic matter cycling and production within shelf seas.

My primary concerns with the manuscript are as following:

- A lack of information in the methods relating to chlorophyll (sample treatment, analysis and calibration of the fluorescence sensor).
- Interpretation of $\delta^{13}C_{POC}$ and $\delta^{15}N_{PN}$ is sometimes highly speculative and in some cases data to support certain arguments are not provided (e.g. dissolved inorganic nutrient data).
- Grammar and sentence structure need improvement in some sections.

**Specific points:**

**Introduction**

P2, L8: Suggest the following grammatical changes to paragraph one of the introduction. Note I also suggest inserting a reference to support the statement of typical isotopic values for end members.

"Stable isotopes of organic carbon and nitrogen ($\delta$13C, $\delta$15N) and molar carbon to nitrogen (C/N) ratios are natural tracers frequently used to identify the source and fate of terrestrial organic matter (OM) in estuarine and marine environments (Meyers, 1994; Hedges et al., 1997; Goñi et al., 2014; Selvaraj et al., 2015). This approach is based on $\delta$13C, $\delta$15N and C/N ratios being significantly different between different end-members (e.g. terrestrial and marine), and the assumption that only conservative physical mixing of bulk properties occur in these marginal settings (Thornton and McManus, 1994; Hedges et al., 1986). Quantifying the relative contribution of end-members using mass balance models thus requires known and constant elemental and isotopic values of end-members and major sources of OM in the study region (e.g., Goñi et al., 2003). Therefore, application of mixing models for the discrimination of OM sources discrimination requires clearly identified representative values for local OM sources. However, often end-member values of $\delta$13C, $\delta$15N and molar C/N ratios are represented by 'typical' numbers, such as ca. –20 ‰ and –27 ‰ for $\delta$13C of marine phytoplankton and terrestrial plants [INSERT SUPPORTING REFERENCE], respectively, but without measuring discrete end-member values in real, local or regional OM source materials. For example, a number of earlier studies failed to measure isotopic values of marine phytoplankton despite using end-member mixing models to distinguish marine versus terrestrial OM in surface sediments (e.g., Kao et al., 2003; Wu et al., 2013), or distinguish marine phytoplankton values from

bulk surface particulate organic matter (POM) values (e.g., Zhang et al., 2007), or allochthonous POM (e.g., Hale et al., 2012). As POM in estuarine and marine systems is mostly derived from primary production, the stable isotope values ($\delta$13C, $\delta$15N) and molar C/N ratios of POM are largely representative of phytoplankton biomass (Gearing et al., 1984). Therefore, since phytoplankton are the main primary producers of marine OM, the elemental and isotopic compositions of phytoplankton should be considered while studying the dynamics of POM in the marine water column."

P2, L30 – L31: Please delete "and phytoplankton carbon". The C: Chl ratio of phytoplankton can vary with species, depth and nutrient status and so conversion of Chl a concentration to C concentration can involve significant errors.

P2, L30 – P3, L10: Subsurface or deep chlorophyll maxima (SCM and DCM, respectively) form within the thermocline in shelf sea systems. To my knowledge the formation of SCM in shelf sea systems is largely linked to turbulence, diapycnal nutrient fluxes and light acclimation. I recommend the following papers for further detail and references therein:

Sharples et al. (2001) Phytoplankton distribution and survival in the thermocline, L&O, DOI: 10.4319/lo.2001.46.3.0486

Moore et al. (2006) Phytoplankton photoacclimation and photoadaptation in response to environmental gradients in a shelf sea, L&O DOI: 10.4319/lo.2006.51.2.0936

Hickman et al. (2012) Primary production and nitrate uptake within the seasonal thermocline of a stratified shelf sea, MEPS, DOI: 10.3354/meps09836

Williams et al. (2013) Wind-driven nutrient pulses to the subsurface chlorophyll maximum in seasonally stratified shelf seas, GRL, DOI: 10.1002/2013GL058171

P3, L13: replace "carbon" with "C", note: be consistent with abbreviations, once an abbreviation is defined e.g. carbon (C), or East China Sea (ECS) be sure to use the abbreviation from then on.

P3, L14: Suggest the following grammatical changes: "Nutrient-rich freshwater inputs in turn stimulate water column productivity in coastal water compared to the open ocean. Annual primary production over the entire shelf of the East China Sea is high relative to other marginal seas and was estimated to be …."

P3, L18: Replace "nutrients" with "nutrient"

P3, L19: Insert "composition" after "phytoplankton species"

P3, L20: Replace "constrained in" with "determined for"

P3, L22: Suggest the following grammatical changes: "Nonetheless, studies on elemental ratios and stable isotopic compositions of POM in DCM layers in the East Chin continental shelf sea, especially transfers between the Yangtze and Okinawa Trough are poorly studied (Chen et al., 2017)."

P3, L24: Suggest the following grammatical changes: "A recent study in the north East China Sea investigated elemental and isotopic compositions of POM in the surface, DCM and bottom layer on both seasonal and inter-annual timescales (Gao et al., 2014), however there was minimal attention given to biogeochemical processes associated with the DCM."

P3, L28: Replace "around" with "at", delete "layer", insert "South" before "East China Sea".

P3, L29: Replace "comprehend" with "determine".

P4, L1: Please insert a reference to support this statement.

P4, L2: Suggest the following grammatical changes: "The ECS shelf is wide (>500 km), but relatively shallow (<130 m) with an average water depth of 60 m (INSERT REFERENCE)".

P4, L3: Suggest the following restructuring: "With a catchment area of more than $1.94 \times 10^6$ km$^2$ (Lui et al., 2007) resulting in an annual freshwater discharge of 900 km$^3$ yr$^{-1}$, the fifth largest in the World, and a sediment discharge of 470 Mt yr-1, the fourth largest in the World (Milliman and Farnsworth, 2011), the Yangtze River is the main source of freshwater and sediment in the ECS."

P4, L12: Replace "middle" with "central". Is "north" intentionally repeated?

P4, L13: Suggest the following: "The Changjiang Diluted Water (CDW) is a mixture of Yangtze River freshwater and ECS shelf water and is characterised by…".

P4, L16: Replace "it has been believed" with "it is thought that", perhaps replace "source" with "component".

P4, L16: Suggesting the following: "In winter, the CDW flows southwards along the coastline of a mainland China as a narrow jet (Chen, 2008; Han et al., 2013) and in summer spreads to the northeast (Isobe et al., 2004). These changes are driven by the East Asian monsoon, which constitutes a strong northeast monsoon in winter and weaker southwest monsoon in summer."

P4, L19: Insert "The" before "Taiwan Warm Current".

P4, L20: Delete "the" before "intruding".

P4, L21: Suggest: "In addition, Kuroshio Subsurface Water (KSSW) is upwelled in the northeast near Taiwan Island due to an abrupt change in seafloor topography at the ECS outer shelf…"

P4, L24: Perhaps "oxygen-under saturated". Delete "but".

P4, L25: Replace "East China Sea" with "ECS".

P4, L27: Suggest: "Furthermore, Kuroshio water accounts for up to 90 % of shelf waters in the ECS…"

P4, L30: Suggest: "Primary production in the ECS is nutrient-limited in summer and light-limited in winter (Chen et al., 2001; Chen and Chen 2003), with production being higher in summer. In 2008, annual primary production rates showed distinct spatial variation, with rates in the north-western ECS (155 g C m$^{-2}$ y$^{-1}$) being higher than those of the south-eastern ECS (144 g C m$^{-2}$ y$^{-1}$) and the overall average for the ECS (145 g C m$^{-2}$ y$^{-1}$) (Gong et al., 2003). However, primary production rates have decreased by 86 % between 2008 and 2003, due to installation of a number of reservoirs within the Yangtze River drainage basin (Gong et al., 2006).

**Material and methods**

P5, L4: I think it is usually "Materials and methods"

P5, L4: "Water samples were collected at 36 stations along seven transects in the ECS during the *Science 3* cruise in summer (June 22 – July 21) 2013 (Fig. 1). Water samples were collected from X? DCM depths (10 -130 m; Table 1) at each station using ?L Niskin bottles mounted on a sampling rosette. A Seabird Conductivity-Temperature-Depth (CTD, SBE911+) sensor fitted with a calibrated? Seapoint chlorophyll fluorometer was mounted on the rosette to record the physical properties of the water column and the depth of the DCM, respectively."

P5, L12 - 20: More detail is needed with respect to sample collection. How many depths did you sample during each cast? What auxiliary samples were collected, e.g. Chl a, dissolved inorganic nutrients etc.? What volume were the Niskin bottles? How were samples collected from the CTD – did you use any tubing and if so what kind? How were the PVC bottles cleaned prior to receiving samples? What volume were the PVC bottles? Approximately how long were samples stored prior to filtering? What vacuum pressure/positive pressure conditions did you filter under? What was the range of volumes that you filtered for each sample, e.g. was it of the order of 1L, 2L, 10L? If you did not rinse the filters prior to storage do you think that the potential contribution of salts to the SPM weights was negligible for the volumes you were filtering? Details like these will give the reader more confidence in your results.

P5, L24: If the filters were freeze dried, what is the rationale behind then drying them again at 50 °C for 48h?

P5, L25: "counterpart" is not the right word here, it implies two separate filters were weighed and their difference was considered to be the SPM weight.

P5, L28: If you randomly selected samples for Chl a analysis did you store all filters in the dark? Did the pigments survive the freeze-drying plus 48h drying at 50 °C? How did you prepare your standards? Did you take standards through the full drying/extraction process to check recoveries? How long was the acetone extraction? Further detail is needed here to convince the reader that chlorophyll concentration data are reliable.

P5, L32: Please clarify if it was the half of the filter for POC analysis that was de-carbonated and whether the half for PN analysis was also subjected to decarbonation.

P6, L1: What diameter was the punch, you could say "transferred to tin capsules", which further analysis are you referring to here?

P6, L5: Suggest: "A range of working standards with compositional similarities to the samples were selected (bovine liver, glutamic acid, enriched alanine and nylon 6) and were calibrated against NIST Standard Reference Materials…"

P6, L8: Replace "is" with "was".

P6, L14 – 23: You have raised this issue that your results may show some bias due to de-carbonation of the PN filters and you have provided evidence that your results are in line with those of Wu et al. (2003), who also de-carbonated their samples from this study region. It would perhaps be more useful to be able to quote similar values for this region obtained without de-carbonation of PN filters to suggest that the bias resulting from the freezing and de-carbonation process did not significantly alter your results.

Overall, this paragraph could be improved by restructuring and better linking (or not linking) between sentences.

**Results and interpretations**

P6, L32: Suggest restructuring: "Water temperature in the upper 300-m varied from 15 to 30 °C, with distinct thermal stratification of the water column across the entire study area (Fig. 2)."

P6, L33 – P7, L1: Suggest deleting this sentence as Fig. 2 does not show data from 850 m or 800 m.

P7, L2 – 3: The statement "showing a general decreasing trend from the inner to outer shelf in each transect" with respect to temperature in Fig. 2 does not appear to hold true for the top 4 of the 7 transects.

P7, L5-13: The first and last sentences of this paragraph are repetitive, but quote different average salinity values. In addition, I do not feel that the sentence describing the "middle salinity" adds any useful information to the description of salinity. Therefore I suggest the following reorganisation or something similar: "The salinity distribution at depths of SPM sampling showed an increasing trend from the inner to outer shelf (Fig. 2), varying from 32.7 to 34.7 with an average salinity of $34.0 \pm$ S.D. Low salinity water (<30) was observed in the upper 10-m at four of the coastal stations where water temperatures were <24°C (Fig. 2), suggesting that there was limited influence of the CDW plume in the study region. The highest salinities were observed at depth and off shelf (Fig. 2)."

P7, L19: Replace "limited along the coast" with "restricted to coastal stations".

P7, L27-28: Suggest restructuring to "The highest Chl fluorescence concentration (18.0 µg $L^{-1}$) was observed in surface waters at station DH3-1. All other values were less than 8.0 µg $L^{-1}$ (Fig. 3)."

P7, L28: "were" not "are"

P7, L29: "showed" not "show"

P7, L32: Replace "straddling around" with "across"

P7, L33: Delete the comma after depth

P8, L7: Delete "productivity".

P8, L10: How was the fluorescence sensor on the CTD calibrated? Please provide full details in the methods section, including sample collection and treatment prior to extraction. If the sensor was calibrated then I am not sure I understand the need to re-calibrate the sensor output data with the Chl a concentrations determined here. The slope is less than 1 suggesting that pigments may have deteriorated using the approach outlined on P5, L 24-29 relative to the samples that were collected and extracted for calibration of the sensor. Providing further methodological details on the sample treatment and extraction process during Chl a extraction is needed, for example, were standards and internal reference materials treated in the same way as samples?

P8, L20: "showed" not "shows"

P8, L21: "concentration" should be plural

P8, L24: "were" not "are"

P8, L25: Suggest restructuring this sentence as follows: "POC and PN concentrations were highest near the coast on the inner shelf (>90 µg L$^{-1}$ and >21 µg L$^{-1}$, respectively), and decreased gradually with distance offshore (Fig. 4)."

P8, L27: Delete "nearby off" and insert "of" between "northeast" and "Taiwan"

P8, L28: Insert "by" between "varied" and "more".

P8, L29: Replace "of the entire ECS" with "throughout sampling". Please specify whether $5.6 \pm 0.5$ is the mean $\pm$ S.D. or mean $\pm$ 95% confidence interval.

P8, L34: "Consistent with the POC distribution"

P9, L1-L3: Suggest the following restructuring: "The lowest $\delta^{13}C_{POC}$ values were observed northeast of Taiwan Island in the Okinawa Trough, whereas $\delta^{15}N_{PN}$ values in this region were higher than those of the surrounding area (Fig. 5)."

P9, L4: "was" not "is"

**Discussion**

P9, L17: This sentence is repetitive, suggest deleting as the detail is already covered in the sentence before.

P9, L21: Define SMW (if not already defined)

P9, L22: Insert "at" between "except" and "these"

P9, L24: Suggest "…further delineated the area and water depths influenced by…"

P9, L 25: Suggest "Interestingly, the influence of CDW was constrained to the upper 10 m in five coastal stations, whereas TWCW influenced the upper 30 m and covered three quarters of the study region, with KSSW largely influencing the bottom water across the entire study region (Fig. 2, 6a and 7)."

P9, L30: Delete "to"

P10, Molar C/N Ratio: Do Wu et al (2003) and Liu et al (1998) see any cross shelf trends in C/N ratio? You quote considerably more variable C/N ratios from their studies than you have found in the DCM layers. Is this due differences in sampling depth, water mass, community composition, nutrient status, light status, age/residence time, degree of degradation, season, year etc.?

While I agree that there is likely a dominance of marine-POM in your DCM samples, I think a narrow range of C/N ratios alone does not provide enough evidence to confirm a lack of terrestrial signals transported mainly by the Yangtze River. Perhaps link theses C/N ratios to salinity data presented in Fig. 2, which shows a greater influence of higher S waters and Fig. 7 which indicates a strong influence of KSSW across the shelf and minimal influence of the less saline CDW.

P11, POC/Chl a: I think it would perhaps be useful here to also note that both vertical gradients in community composition and photoacclimation can influence C/N and POC/Chla within subsurface chlorophyll maxima.

Moore et al. (2006) Phytoplankton photoacclimation and photoadaptation in response to environmental gradients in a shelf sea, L&O DOI: 10.4319/lo.2006.51.2.0936

Latasa et al. (2017) Distribution of phytoplankton groups within the deep chlorophyll maximum. L&O, DOI: 10.1002/lno.10452.

P12, L 10-14: Is the increasing trend of $\delta^{13}$C evident in SPM and surface sediments from this study or from the literature? What C does it refer to? This is a little unclear.

P12, L34: This needs a supporting reference e.g. Burkhardt et al. (1999).

P14, L30 – P15, L6: How would local regeneration of N and regenerated production influence these isotopic values, in addition to the $\delta^{15}$NO$_3$ of source waters?

P15, L17 – L33: I found this paragraph hard to follow and overly speculative.

"… may not be resulted from the high degree of nitrate utilisation, but the incorporation of inorganic nitrogen in the POM." Nitrate is inorganic nitrogen, do you mean that there is a higher proportion of particulate inorganic nitrogen relative to particulate organic nitrogen and that this could be driving the isotopic signal here?

"The low Chl fluorescence might be limited by the low temperature in this high nutrient low chlorophyll region." The depths where you sampled were quite warm and of similar temperature to one another, I think a temperature effect on Chl here is unlikely. These appear to be the deepest samples at ~100m. Is it possible that this signature of low Chl and low POM concentration is because you were sampling below the euphotic layer? This could possibly contribute to the isotopic signal at this station, as below the euphotic layer remineralisation and degradation of POM would exceed production. The $\delta^{13}$C$_{DIC}$ at depth may have been quite different to that in the euphotic layer. If dissolved inorganic nutrient data and PAR data are available and support your argument that it is a temperature effect then this should be included.

No NH$_4^+$ data are available, and no $\delta^{15}$N$_{NH4+}$ values have been previously published for this region, therefore I find the discussion linking the high $\delta^{15}$N$_{PN}$ to ammonium assimilation quite speculative.

P16, L5: insert "respectively" after "Wu et al"

P16, L7: Suggest restructuring to this or similar: "Our results indicate that POM at the DCM was largely produced in situ and derived from phytoplankton biomass, with little terrestrial influence. The lack of terrestrial OM signals…"

P16, L10: "has been reduced"

P16, L12: "…reported that the particulate load discharged by…"

P16, L22: "Accompanying the decreasing…"

P17, L7: "…despite the study are being the best…"

P17, L9: As I understand it you did not measure primary production rates, you are inferring primary productivity from POC concentration. Therefore it may be more accurate here to say biomass rather than primary productivity. Again, to my knowledge you did not collect data on community

composition and therefore I am concerned that this statement is overly speculative for your concluding remarks.

P17, L11: Again, linking changes in $\delta^{15}N_{PN}$ when no $NH_4^+$ or $NO_3^-$ data or uptake data are available is perhaps overly speculative.

P17, L20: Please expand on the link between the DCM and the inner shelf mud-belt that accumulated during the Holocene.

---

## Author Response (AR2)

We are grateful to the Associate Editor, PA Meyers and an anonymous reviewer for their constructive feedbacks and insightful comments on our manuscript entitled "Biogeochemical characteristics of suspended particulates at deep chlorophyll maximum layers in the southern East China Sea" (**MS. Ref. No.: bg-2017-290-R1**).

**Referee 1 (Prof. P. A. Meyers)**
Liu and colleagues have nicely revised their interesting paper that describes the results of their study of the organic carbon and nitrogen contents in suspended particles collected around deep chlorophyll maximum layers in the East China Sea. The English has been much improved, and the whole paper is now easier to read and appreciate. I suggest a few minor additional refinements that might be considered by the authors before the contribution is ready to be published in Biogeosciences:

**Reply**: Thank you.

*Referee 1*: Title – change to read "'….suspended particulate matter in deep……"

**Reply (Page 1, Lines 1-2)**: Changed.

*Referee 1*: Page 1, line 23 – replace "particulates" with "particles" (particulate is an adjective, not a noun)

**Reply (Page 1, Line 19)**: Replaced.

*Referee 1*: Page 2, line 2 – change to read "additional radiocarbon and biomarker data are needed to re-evaluate"

**Reply (Page 2, Lines 4-6)**: This sentence has been changed as follows:

Nonetheless, additional radiocarbon and biomarker data are needed to re-evaluate whether or not the POM around the DCM water depths is influenced by terrestrial OM in the river-dominated East China Sea.

*Referee 1*: Page 8, line 27 – change to read "near northeast Taiwan"

**Reply (Page 9, Line 19)**: Changed.

*Referee 1*: Page 9, line 27 – change to read "influenced by the bottom"

**Reply (Page 10, Lines 18-22)**: By combining suggestions of Referees 1 and 4, this sentence has been changed as follows:

Interestingly, the influence of CDW was constrained to the upper 10 m in five coastal stations, whereas TWCW influenced the upper 30 m and covered three quarters of the study region,

with KSSW largely influencing the bottom water across the entire study region (Figs. 2, 6a and 7).

*Referee 1*: Page 9, line 30 – change to read "transported northeastward of the"

**Reply (Page 10, Line 26)**: Changed.

*Referee 1*: Page 10, line 23 – change to read "to more intense biological"

**Reply (Page 11, Line 19)**: Changed.

*Referee 1*: Page 10, line 29 – change to read "ratio to more than that of"

**Reply (Page 11, Line 25)**: Corrected.

*Referee 1*: Page 14, line 2 – change to read "biomass with increasing"

**Reply (Page 14, Line 32)**: Changed.

*Referee 1*: Page 15, line 18 – change to read "may not result from the high degree of nitrate utilization, but instead from the"

**Reply (Page 16, Lines 15-17)**: This sentence has been changed as follows:

There is another possibility that high $\delta^{15}N_{PN}$ (DH7-8: 6.7 ‰, DH7-9: 7.8 ‰) in the DCM layer, off northeast Taiwan (Fig. 5), may not result from the high degree of nitrate utilization, but instead from the incorporation of inorganic nitrogen (mainly $NH_4^+$) in the POM.

*Referee 1*: Page 17, line 10 – change "accounted" to "accounting"

**Reply (Page 18, Line 12)**: Corrected.

*Referee 1*: Page 17, line 13 – replace "but" with "which"

**Reply (Page 18, Line 13)**: Replaced.

*Referee 1*: Page 17, line 15 – change "low" to "lower"

**Reply (Page 18, Line 17)**: Changed.

Thank you very much.

**Anonymous Referee #4**

General overview:

Liu et al. present isotopic compositions and concentrations of particulate organic carbon and particulate nitrogen (POC and PON, respectively) in samples collected from the deep chlorophyll maximum (DCM) in the south East China Sea in summer 2013. They combine these data sets with temperature, salinity, turbidity and calibrated chlorophyll fluorescence data to determine the sources of particulate organic matter in the DCM and what factors govern isotopic dynamics of POC and PON in the DCM. The authors attribute variation in $\delta^{13}C_{POC}$ to be governed by changes in primary productivity and community composition and variation in $\delta^{15}N_{PN}$ to be governed by changes in uptake of dissolved inorganic nitrogen (i.e. $NH_4^+$ or $NO_3^-$) and source (with a link to water masses). The handling and interpretation of isotopic data is good, however with a lack of certain auxiliary data sets (e.g. dissolved inorganic nutrients, community composition), and full methodological detail in the determination of others (chlorophyll), it is hard to fully critique their interpretation of the data and their conclusions. Following revision, this study could contribute to our understanding of organic matter cycling and production within shelf seas.

My primary concerns with the manuscript are as following:

- A lack of information in the methods relating to chlorophyll (sample treatment, analysis and calibration of the fluorescence sensor).
- Interpretation of $\delta^{13}C_{POC}$ and $\delta^{15}N_{PN}$ is sometimes highly speculative and in some cases data to support certain arguments are not provided (e.g. dissolved inorganic nutrient data).
- Grammar and sentence structure need improvement in some sections.

Specific points:

*Referee 4*: Introduction

P2, L8: Suggest the following grammatical changes to paragraph one of the introduction. Note I also suggest inserting a reference to support the statement of typical isotopic values for end members.

"Stable isotopes of organic carbon and nitrogen ($\delta^{13}C$, $\delta^{15}N$) and molar carbon to nitrogen (C/N) ratios are natural tracers frequently used to identify the source and fate of terrestrial organic matter (OM) in estuarine and marine environments (Meyers, 1994; Hedges et al., 1997; Goñi et al., 2014; Selvaraj et al., 2015). This approach is based on $\delta^{13}C$, $\delta^{15}N$ and C/N ratios being significantly different between different end-members (e.g. terrestrial and marine), and the assumption that only conservative physical mixing of bulk properties occur in these marginal settings (Thornton and McManus, 1994; Hedges et al., 1986). Quantifying the relative

contribution of end-members using mass balance models thus requires known and constant elemental and isotopic values of end-members and major sources of OM in the study region (e.g., Goñi et al., 2003). Therefore, application of mixing models for the discrimination of OM sources discrimination requires clearly identified representative values for local OM sources. However, often end-member values of $\delta^{13}C$, $\delta^{15}N$ and molar C/N ratios are represented by 'typical' numbers, such as ca. –20 ‰ and –27 ‰ for $\delta^{13}C$ of marine phytoplankton and terrestrial plants [INSERT SUPPORTING REFERENCE], respectively, but without measuring discrete end-member values in real, local or regional OM source materials. For example, a number of earlier studies failed to measure isotopic values of marine phytoplankton despite using end-member mixing models to distinguish marine versus terrestrial OM in surface sediments (e.g., Kao et al., 2003; Wu et al., 2013), or distinguish marine phytoplankton values from bulk surface particulate organic matter (POM) values (e.g., Zhang et al., 2007), or allochthonous POM (e.g., Hale et al., 2012). As POM in estuarine and marine systems is mostly derived from primary production, the stable isotope values ($\delta^{13}C$, $\delta^{15}N$) and molar C/N ratios of POM are largely representative of phytoplankton biomass (Gearing et al., 1984). Therefore, since phytoplankton are the main primary producers of marine OM, the elemental and isotopic compositions of phytoplankton should be considered while studying the dynamics of POM in the marine water column."

**Reply (Page 2, Lines 10-35)**: This paragraph has been changed appropriately as follows:.

Stable isotopes of organic carbon and nitrogen ($\delta^{13}C$, $\delta^{15}N$) and molar carbon to nitrogen (C/N) ratios are natural tracers frequently used to identify the source and fate of terrestrial organic matter (OM) in the estuarine and marine environments (Meyers, 1994; Hedges et al., 1997; Goñi et al., 2014; Selvaraj et al., 2015). This approach is based on the significant difference in $\delta^{13}C$, $\delta^{15}N$ and C/N ratios between different end-members (e.g., terrestrial and marine), and the assumption that only a physical mixing of OM from compositionally distinct end-members occurs in these marginal settings (Thornton and McManus, 1994; Hedges et al., 1986). Quantifying the relative contributions of end-members using mass balance models thus requires known and constant elemental and isotopic values of end-members and major sources of OM in the study region (e.g., Goñi et al., 2003). Therefore, application of mixing models for the discrimination of OM sources requires clearly identified representative values for local OM sources. However, in most cases, end-member values of $\delta^{13}C$, $\delta^{15}N$ and molar C/N ratios are represented by 'typical' numbers, such as ca. –20 ‰ and –27 ‰ for $\delta^{13}C$ of marine phytoplankton and terrestrial plants (Kandasamy and Nagender Nath 2016 and references therein), respectively, but without measuring discrete end-member values in real, local or regional OM source materials. For example, a number of earlier studies failed to measure isotopic values of marine phytoplankton despite using end-member mixing models to distinguish marine versus terrestrial OM in surface sediments (e.g., Kao et al., 2003; Wu et al., 2013), or these numbers simply represented by values of particulate organic matter (POM) in surface waters in the studied system (e.g., Zhang et al., 2007) or elsewhere from other ocean basins (e.g., Hale et al., 2012). It is known that stable isotopes ($\delta^{13}C$, $\delta^{15}N$) and molar C/N ratios of POM in estuarine and marine areas are representative of primary production-derived OM when POM are mostly derived from phytoplankton biomass (Gearing et al., 1984). Since

phytoplankton are the main primary producer of marine OM, the elemental and isotopic compositions of phytoplankton should therefore be considered while studying the dynamics of POM in the marine water column.

*Referee 4*: P2, L30 – L31: Please delete "and phytoplankton carbon". The C: Chl ratio of phytoplankton can vary with species, depth and nutrient status and so conversion of Chl a concentration to C concentration can involve significant errors.

**Reply (Page 3, Lines 1-2)**: Deleted.

*Referee 4*: P2, L30 – P3, L10: Subsurface or deep chlorophyll maxima (SCM and DCM, respectively) form within the thermocline in shelf sea systems. To my knowledge the formation of SCM in shelf sea systems is largely linked to turbulence, diapycnal nutrient fluxes and light acclimation. I recommend the following papers for further detail and references therein:

Sharples et al. (2001) Phytoplankton distribution and survival in the thermocline, L&O, DOI: 10.4319/lo.2001.46.3.0486
Moore et al. (2006) Phytoplankton photoacclimation and photoadaptation in response to environmental gradients in a shelf sea, L&O DOI: 10.4319/lo.2006.51.2.0936
Hickman et al. (2012) Primary production and nitrate uptake within the seasonal thermocline of a stratified shelf sea, MEPS, DOI: 10.3354/meps09836
Williams et al. (2013) Wind-driven nutrient pulses to the subsurface chlorophyll maximum in seasonally stratified shelf seas, GRL, DOI: 10.1002/2013GL058171

**Reply (Page 3, Lines 6-12)**: Thanks for providing these important references on the DCM formation in the shelf seas. Based on some of these references, we revised the text as follows:

The formation of maximum chlorophyll concentration at the DCM layer has been explained by several mechanisms: the differential zooplankton grazing with depths (Riley et al., 1949; Lorenzen, 1967), adaption of phytoplankton to light intensities or to increased concentration of nutrients (Nielsen and Hansen, 1959; Gieskes et al., 1978; Hickman et al., 2012), chlorophyll accumulation by sinking detritus of phytoplankton (Gieskes et al., 1978; Karlson et al., 1996), decomposition of chlorophyll by light (Nielsen and Hansen, 1959), and wind-driven nitrate supply and nitrate uptake in seasonally-stratified shelf seas (Hickman et al., 2012; Williams et al., 2013).

*Referee 4*: P3, L13: replace "carbon" with "C", note: be consistent with abbreviations, once an abbreviation is defined e.g. carbon (C), or East China Sea (ECS) be sure to use the abbreviation from then on.

**Reply (Page 3, Line 22)**: Replaced.

*Referee 4*: P3, L14: Suggest the following grammatical changes: "Nutrient-rich freshwater inputs in turn stimulate water column productivity in coastal water compared to the open ocean.

Annual primary production over the entire shelf of the East China Sea is high relative to other marginal seas and was estimated to be ...."

**Reply (Page 3, Lines 22-25)**: Corrected, as suggested.

*Referee 4*: P3, L18: Replace "nutrients" with "nutrient"

**Reply (Page 3, Line 27)**: Replaced.

*Referee 4*: P3, L19: Insert "composition" after "phytoplankton species"

**Reply (Page 3, Line 27)**: Inserted.

*Referee 4*: P3, L20: Replace "constrained in" with "determined for"

**Reply (Page 3, Line 29)**: Replaced.

*Referee 4*: P3, L22: Suggest the following grammatical changes: "Nonetheless, studies on elemental ratios and stable isotopic compositions of POM in DCM layers in the East Chin continental shelf sea, especially transfers between the Yangtze and Okinawa Trough are poorly studied (Chen et al., 2017)."

**Reply**: We disagree with the Referee's view here because there are three pathways of material transfer between the Yangtze and Okinawa Trough. Therefore, we prefer to keep our original text here to specify "along the indirect pathway of the Yangtze-derived terrestrial material to the Okinawa Trough". Thank you.

*Referee 4*: P3, L24: Suggest the following grammatical changes: "A recent study in the north East China Sea investigated elemental and isotopic compositions of POM in the surface, DCM and bottom layer on both seasonal and inter-annual timescales (Gao et al., 2014), however there was minimal attention given to biogeochemical processes associated with the DCM."

**Reply (Page 4, Line 4 and Page 5, Lines 1-4)**: As suggested, the sentence has been modified as follows:
A recent study in the northern East China Sea investigated elemental and isotopic compositions of POM in the surface, DCM and bottom layers on both seasonal and inter-annual timescales (Gao et al., 2014); however, there was minimal attention given to biogeochemical processes associated with the DCM.

*Referee 4*: P3, L28: Replace "around" with "at", delete "layer", insert "South" before "East China Sea".

**Reply**: Some samples investigated in this study fall contiguous to the DCM layer and therefore we prefer to keep our original text here. Thank you.

*Referee 4*: P3, L29: Replace "comprehend" with "determine".

**Reply**: We think here "comprehend" is better than "determine", since OM sources' identification in this study was based on multi-data analyses.

*Referee 4*: P4, L1: Please insert a reference to support this statement.

**Reply (Page 4, Line 11)**: Inserted.

*Referee 4*: P4, L2: Suggest the following grammatical changes: "The ECS shelf is wide (>500 km), but relatively shallow (<130 m) with an average water depth of 60 m (INSERT REFERENCE)".

**Reply (Page 4, Lines 12-13)**: This sentence has been changed as follows:

The ECS shelf is wide (>500 km), but relatively shallow (<130 m) with an average water depth of 60 m (Gong et al., 2003; Liu et al., 2006).

*Referee 4*: P4, L3: Suggest the following restructuring: "With a catchment area of more than 1.94 x 10$^6$ km$^2$ (Lui et al., 2007) resulting in an annual freshwater discharge of 900 km$^3$ yr$^{-1}$, the fifth largest in the World, and a sediment discharge of 470 Mt yr$^{-1}$, the fourth largest in the World (Milliman and Farnsworth, 2011), the Yangtze River is the main source of freshwater and sediment in the ECS."

**Reply**: As given by the referee, this sentence is too long! In general, the scientific literature prefers shorter and meaningful sentences. We therefore keep the original text here without any change.

*Referee 4*: P4, L12: Replace "middle" with "central". Is "north" intentionally repeated?

**Reply (Page 4, Lines 22-23)**: Corrected.

*Referee 4*: P4, L13: Suggest the following: "The Changjiang Diluted Water (CDW) is a mixture of Yangtze River freshwater and ECS shelf water and is characterised by…".

**Reply (Page 4, Lines 23-25)**: Corrected.

*Referee 4*: P4, L16: Replace "it has been believed" with "it is thought that", perhaps replace "source" with "component".

**Reply (Page 4, Lines 26-27)**: Replaced.

*Referee 4*: P4, L16: Suggesting the following: "In winter, the CDW flows southwards along the coastline of a mainland China as a narrow jet (Chen, 2008; Han et al., 2013) and in summer

spreads to the northeast (Isobe et al., 2004). These changes are driven by the East Asian monsoon, which constitutes a strong northeast monsoon in winter and weaker southwest monsoon in summer."

**Reply**: Since we are unhappy about the referee's mere shifting of sentences in the name of grammar and restructuring, we prefer to keep the original text as such here.

*Referee 4*: P4, L19: Insert "The" before "Taiwan Warm Current".

**Reply (Page 4, Line 30)**: Inserted.

*Referee 4*: P4, L20: Delete "the" before "intruding".

**Reply (Page 4, Line 32)**: Deleted.

*Referee 4*: P4, L21: Suggest: "In addition, Kuroshio Subsurface Water (KSSW) is upwelled in the northeast near Taiwan Island due to an abrupt change in seafloor topography at the ECS outer shelf…"

**Reply (Page 4, Lines 33-35)**: This sentence has been changed appropriately as follows:

In addition, Kuroshio Subsurface Water (KSSW) is upwelled in the northeast off Taiwan Island due to an abrupt change in seafloor topography at the ECS outer shelf

*Referee 4*: P4, L24: Perhaps "oxygen-under saturated". Delete "but".

**Reply**: Sorry! To the best our knowledge, oxygen-unsaturated is more scientific and widely used while dealing biogeochemical cycles/processes. Please refer to the definition of oxygen saturation – *a ratio of the concentration of dissolved oxygen ($O_2$) in the water to the maximum amount of oxygen that will dissolve in that water*. Deleted (**Page 5, Line 1**).

*Referee 4*: P4, L25: Replace "East China Sea" with "ECS".

**Reply (Page 5, Line 2)**: Replaced.

*Referee 4*: P4, L27: Suggest: "Furthermore, Kuroshio water accounts for up to 90 % of shelf waters in the ECS…"

**Reply (Page 5, Lines 5)**: Corrected.

*Referee 4*: P4, L30: Suggest: "Primary production in the ECS is nutrient-limited in summer and light-limited in winter (Chen et al., 2001; Chen and Chen 2003), with production being higher in summer. In 2008, annual primary production rates showed distinct spatial variation, with rates in the north-western ECS (155 g C $m^{-2}$ $y^{-1}$) being higher than those of the south-eastern ECS

(144 g C m$^{-2}$ y$^{-1}$) and the overall average for the ECS (145 g C m$^{-2}$ y$^{-1}$) (Gong et al., 2003). However, primary production rates have decreased by 86 % between 2008 and 2003, due to installation of a number of reservoirs within the Yangtze River drainage basin (Gong et al., 2006).

**Reply**: Since we are unhappy about the referee's version, we prefer to keep the original text as such here because that is more simple and understandable than the referee's version.

*Referee 4*: Material and methods

P5, L4: I think it is usually "Materials and methods"

**Reply (Page 5, Line 14)**: Corrected.

*Referee 4*: P5, L4: "Water samples were collected at 36 stations along seven transects in the ECS during the Science 3 cruise in summer (June 22 – July 21) 2013 (Fig. 1). Water samples were collected from X? DCM depths (10 -130 m; Table 1) at each station using ?L Niskin bottles mounted on a sampling rosette. A Seabird Conductivity-Temperature-Depth (CTD, SBE911+) sensor fitted with a calibrated? Seapoint chlorophyll fluorometer was mounted on the rosette to record the physical properties of the water column and the depth of the DCM, respectively."

**Reply**: These are standard ways of collecting and analyzing samples and measuring onboard parameters and researchers throughout the world are following these standardized procedures. Details given in the original version are clear and concise enough to the readers.

*Referee 4*: P5, L24: If the filters were freeze dried, what is the rationale behind then drying them again at 50 °C for 48h?

**Reply**: This is the standard way to eliminate the influence of temperature and humidity on the determination of SPM weight.

*Referee 4*: P5, L25: "counterpart" is not the right word here, it implies two separate filters were weighed and their difference was considered to be the SPM weight.

**Reply (Page 6, Line 2)**: "its counterpart" was changed as "the same filter".

*Referee 4*: P5, L28: If you randomly selected samples for Chl a analysis did you store all filters in the dark? Did the pigments survive the freeze-drying plus 48h drying at 50 °C? How did you prepare your standards? Did you take standards through the full drying/extraction process to check recoveries? How long was the acetone extraction? Further detail is needed here to convince the reader that chlorophyll concentration data are reliable.

**Reply**: All filters were wrapped in aluminum foil, which protected filters from light and stored at

–20 °C freezer onboard immediately after filtration. We used the monochromatic method with acidification to determine Chl a concentration by UV-Vis spectrophotometer and samples were measured against 90% acetone as blank.

This paragraph has been changed as follows (**Page 6, Lines 1-12**):

In the laboratory, filters with suspended particles were freeze-dried and then dried in an oven at 50 ºC for 48 h. The weight difference between the dried filter and the same filter before the filtration was used to calculate the weight of SPM. Five SPM samples (DH1-2, DH2-1, DH3-1, DH7-1 and DH7-7; Fig. S1) from water depths ranging between 20 m and 50 m were randomly selected for the measurement of chlorophyll a (Chl a) concentration. Chlorophyll a was extracted using 90% acetone and then determined spectrophotometrically according to Lorenzen (1967) and Aminot and Rey (2000). Briefly, the absorbance of sample extraction was measured at 665 nm and 750 nm against a 90% acetone blank before ($E665_o$, $E750_o$) and after ($E665_a$, $E750_a$) acidification with 1% HCl by the UV-Vis spectrophotometer (UV 1800, Shimadzu). Chl a concentration ($\mu g\ L^{-1}$) was calculated as: Chl $a = 11.4 \times 2.43 \times ((E665_o − E750_o) − (E665_a − E750_a)) \times V_e /L \times V_f$, where $V_e$ and $V_f$ were the volumes of sample extraction and sea water filtered (mL), respectively, and L was the cuvette light-path (cm) (Aminot and Rey, 2000).

*Referee 4*: P5, L32: Please clarify if it was the half of the filter for POC analysis that was de-carbonated and whether the half for PN analysis was also subjected to decarbonation.

**Reply**: Please refer to our reply to Referee 2 comments in the previous round of review.

*Referee 4*: P6, L1: What diameter was the punch, you could say "transferred to tin capsules", which further analysis are you referring to here?

**Reply**: Please refer to our reply to Referee 2 comments in the previous round of review.

*Referee 4*: P6, L5: Suggest: "A range of working standards with compositional similarities to the samples were selected (bovine liver, glutamic acid, enriched alanine and nylon 6) and were calibrated against NIST Standard Reference Materials…"

**Reply**: The original text is better than the suggested by the referee. For instance, "A range was selected, not were"! We prefer to keep the original sentence here.

*Referee 4*: P6, L8: Replace "is" with "was".

**Reply (Page 6, Line 25)**: Replaced.

*Referee 4*: P6, L14 – 23: You have raised this issue that your results may show some bias due to de-carbonation of the PN filters and you have provided evidence that your results are in line with those of Wu et al. (2003), who also de-carbonated their samples from this study region. It would perhaps be more useful to be able to quote similar values for this region obtained

without de-carbonation of PN filters to suggest that the bias resulting from the freezing and de-carbonation process did not significantly alter your results.

Overall, this paragraph could be improved by restructuring and better linking (or not linking) between sentences.

**Reply**: Thanks for your suggestion. However, this paragraph is included based on the suggestion of Referee 2 in the previous round of review. Thank you.

*Referee 4*: Results and interpretations

P6, L32: Suggest restructuring: "Water temperature in the upper 300-m varied from 15 to 30 °C, with distinct thermal stratification of the water column across the entire study area (Fig. 2)."

**Reply (Page 7, Lines 15-17)**: Corrected.

*Referee 4*: P6, L33 – P7, L1: Suggest deleting this sentence as Fig. 2 does not show data from 850 m or 800 m.

**Reply (Page 7, Line 23)**: We cite Table S1 where one can see temperatures of 850 m and 800 m.

*Referee 4*: P7, L2 – 3: The statement "showing a general decreasing trend from the inner to outer shelf in each transect" with respect to temperature in Fig. 2 does not appear to hold true for the top 4 of the 7 transects.

**Reply**: This is mainly because stations, DH1-1, DH2-1, DH3-1, CON02 and DH2-2 in the top 4 of 7 transects, that were influenced by SMW (see section 5.1 for more details).

*Referee 4*: P7, L5-13: The first and last sentences of this paragraph are repetitive, but quote different average salinity values. In addition, I do not feel that the sentence describing the "middle salinity" adds any useful information to the description of salinity. Therefore I suggest the following reorganisation or something similar: "The salinity distribution at depths of SPM sampling showed an increasing trend from the inner to outer shelf (Fig. 2), varying from 32.7 to 34.7 with an average salinity of 34.0 ± S.D. Low salinity water (<30) was observed in the upper 10-m at four of the coastal stations where water temperatures were <24°C (Fig. 2), suggesting that there was limited influence of the CDW plume in the study region. The highest salinities were observed at depth and off shelf (Fig. 2)."

**Reply**: The first sentence referred to salinity profile in the whole water column, while the last sentence referred to salinity at the sampling depths. Please check carefully. The description on "middle salinity" is necessary as it is related to the TWCW and SMW, and the same is discussed in section 5.1 with more details. In fact, the entire paragraph was restructured based on the correction of Referee 2 in the previous round of review. Referee 4 should refer to those corrections before providing more or less similar, but a distorting way of restructuring in the version already revised.

*Referee 4*: P7, L19: Replace "limited along the coast" with "restricted to coastal stations".

**Reply (Page 8, Line 8)**: We feel that "stations were limited along the coast" is more appropriate than "stations were restricted to the coastal stations.

*Referee 4*: P7, L27-28: Suggest restructuring to "The highest Chl fluorescence concentration (18.0 µg L-1) was observed in surface waters at station DH3-1. All other values were less than 8.0 µg L-1 (Fig. 3)."

**Reply (Page 8, Lines 12-14)**: Restructured.

*Referee 4*: P7, L29: "showed" not "show"

**Reply (Page 8, Line 14)**: Corrected. However, when we describe our own results with figure and table citations, it is better to use the present tense than the past tense.

*Referee 4*: P7, L32: Replace "straddling around" with "across"

**Reply (Page 8, Line 18)**: Replaced.

*Referee 4*: P7, L33: Delete the comma after depth

**Reply (Page 9, Line 19)**: Deleted.

*Referee 4*: P8, L7: Delete "productivity".

**Reply (Page 9, Line 26)**: Deleted.

*Referee 4*: P8, L20: "showed" not "shows"

**Reply (Page 9, Line 7)**: Corrected.

*Referee 4*: P8, L21: "concentration" should be plural

**Reply (Page 9, Line 8)**: Corrected.

*Referee 4*: P8, L24: "were" not "are"

**Reply (Page 9, Line 14)**: Corrected.

*Referee 4*: P8, L25: Suggest restructuring this sentence as follows: "POC and PN concentrations were highest near the coast on the inner shelf (>90 µg L-1 and >21 µg L-1, respectively), and decreased gradually with distance offshore (Fig. 4)."

**Reply (Page 9, Lines 12-15)**: Corrected.

*Referee 4*: P8, L27: Delete "nearby off" and insert "of" between "northeast" and "Taiwan"
**Reply (Page 9, Line 16)**: Corrected.

*Referee 4*: P8, L28: Insert "by" between "varied" and "more".

**Reply (Page 9, Line 17)**: Inserted.

*Referee 4*: P8, L29: Replace "of the entire ECS" with "throughout sampling". Please specify whether 5.6 ± 0.5 is the mean ± S.D. or mean ± 95% confidence interval.

**Reply (Page 9, Line 18)**: Replaced. 5.6 ± 0.5 is the mean ± S.D. (see Table 1 for clarification).

*Referee 4*: P8, L34: "Consistent with the POC"

**Reply (Page 9, Line 23)**: Corrected.

*Referee 4*: P9, L1-L3: Suggest the following restructuring: "The lowest $\delta^{13}C_{POC}$ values were observed northeast of Taiwan Island in the Okinawa Trough, whereas $\delta^{15}N_{PN}$ values in this region were higher than those of the surrounding area (Fig. 5)."

**Reply (Page 9, Lines 25-28)**: Restructured as follows:

The lowest $\delta^{13}C_{POC}$ values (–25.8 ‰ and –25.2 ‰) were observed northeast of Taiwan Island in the Okinawa Trough, whereas $\delta^{15}N_{PN}$ values (6.73 ‰ and 7.78 ‰) in this region were higher than those of the surrounding area (Fig. 5).

*Referee 4*: P9, L4: "was" not "is"

**Reply (Page 9, Line 28)**: Corrected.

*Referee 4*: Discussion
P9, L17: This sentence is repetitive, suggest deleting as the detail is already covered in the sentence before.

**Reply (Page 10, Lines 6-7)**: Deleted.

*Referee 4*: P9, L21: Define SMW (if not already defined)

**Reply (Page 10, Line 10)**: Defined.

*Referee 4*: P9, L22: Insert "at" between "except" and "these"

**Reply (Page 10, Line 11)**: Inserted.

*Referee 4*: P9, L24: Suggest "…further delineated the area and water depths influenced by…"

**Reply (Page 10, Lines 13-14)**: Corrected.

*Referee 4*: P9, L 25: Suggest "Interestingly, the influence of CDW was constrained to the upper 10 m in five coastal stations, whereas TWCW influenced the upper 30 m and covered three quarters of the study region, with KSSW largely influencing the bottom water across the entire study region (Fig. 2, 6a and 7)."

**Reply (Page 10, Lines 14-18)**: Corrected.

*Referee 4*: P9, L30: Delete "to"

**Reply (Page 10, Line 22)**: Deleted.

*Referee 4*: P11, POC/Chl a: I think it would perhaps be useful here to also note that both vertical gradients in community composition and photoacclimation can influence C/N and POC/Chla within subsurface chlorophyll maxima.
Moore et al. (2006) Phytoplankton photoacclimation and photoadaptation in response to environmental gradients in a shelf sea, L&O DOI: 10.4319/lo.2006.51.2.0936
Latasa et al. (2017) Distribution of phytoplankton groups within the deep chlorophyll maximum. L&O, DOI: 10.1002/lno.10452.

**Reply**: Thanks for your suggestions. We agree with the referee that light or photoacclimination also influences the phytoplankton groups/community distribution in the water column, and may further contribute to the variations of POC/Chl a ratio as well as carbon and nitrogen isotopes in DCM. More complex physical forcings are possible, but these are beyond the scope of the present study.

*Referee 4*: P12, L 10-14: Is the increasing trend of $\delta^{13}C$ evident in SPM and surface sediments from this study or from the literature? What C does it refer to? This is a little unclear.

**Reply**: The inner to outer shelf increasing trend of $\delta^{13}C$ in SPM and surface sediments was evident from the literature cited in **Page 13, Lines 4-5**. Carbon isotopic composition of particulate organic carbon - $\delta^{13}C_{POC}$.

*Referee 4*: P12, L34: This needs a supporting reference e.g. Burkhardt et al. (1999).

**Reply (Page 13, Line 25)**: The following two references are included.

Falkowski, P. G.: Species variability in the fractionation of $^{13}C$ and $^{12}C$ by marine phytoplankton, J. Plank. Res., 13, 21–28, 1991.

Hinga, K. R., Arthur, M. A., Pilson, M. E. Q., and Whitaker, D.: Carbon isotope fractionation by marine phytoplankton in culture: The effects of $CO_2$ concentration, pH, temperature, and species, Global Biogeochem. Cy., 8, 91–102, 1994.

*Referee 4*: P14, L30 – P15, L6: How would local regeneration of N and regenerated production influence these isotopic values, in addition to the $\delta^{15}NO_3$ of source waters?

**Reply**: $NO_3^-$ is considered as the dominant source of nitrogen in this region. In case if the regenerated N from OM was the nitrogen source and it is deficit relative to other nutrients (Si, P), then $\delta^{15}N_{PN}$ would show isotopic values similar to regenerated N.

*Referee 4*: P15, L17 – L33: I found this paragraph hard to follow and overly speculative.
"… may not be resulted from the high degree of nitrate utilisation, but the incorporation of inorganic nitrogen in the POM." Nitrate is inorganic nitrogen, do you mean that there is a higher proportion of particulate inorganic nitrogen relative to particulate organic nitrogen and that this could be driving the isotopic signal here?
"The low Chl fluorescence might be limited by the low temperature in this high nutrient low chlorophyll region." The depths where you sampled were quite warm and of similar temperature to one another, I think a temperature effect on Chl here is unlikely. These appear to be the deepest samples at ~100m. Is it possible that this signature of low Chl and low POM concentration is because you were sampling below the euphotic layer? This could possibly contribute to the isotopic signal at this station, as below the euphotic layer remineralisation and degradation of POM would exceed production. The $\delta^{13}C_{DIC}$ at depth may have been quite different to that in the euphotic layer. If dissolved inorganic nutrient data and PAR data are available and support your argument that it is a temperature effect then this should be included.
No $NH_4^+$ data are available, and no $\delta^{15}N_{NH4+}$ values have been previously published for this region, therefore I find the discussion linking the high $\delta^{15}N_{PN}$ to ammonium assimilation quite speculative.

**Reply**: We agree with the referee that some of our statements are speculative, but we explained why our speculation is valid in our reply to Referee 2 comments in the previous round of review. It is really unfortunate that Referee 4 here provided almost similar comments (~90%) what Referee 2 have raised in the previous round of review.

*Referee 4*: P16, L5: insert "respectively" after "Wu et al"

**Reply (Page 16, Line 34)**: We used semicolon (not and!) in between these values, and therefore inserting "respectively" is not necessary here!

*Referee 4*: P16, L7: Suggest restructuring to this or similar: "Our results indicate that POM at the DCM was largely produced in situ and derived from phytoplankton biomass, with little terrestrial influence. The lack of terrestrial OM signals…"

**Reply (Page 16, Line 35 and Page 17, Lines 1-2)**: Corrected as suggested.

*Referee 4*: P16, L10: "has been reduced"

**Reply (Page 17, Line 5)**: Corrected.

*Referee 4*: P16, L12: "…reported that the particulate load discharged by…"

**Reply (Page 17, Line 7)**: Corrected.

*Referee 4*: P16, L22: "Accompanying the decreasing…"

**Reply (Page 17, Line 18)**: Corrected.

*Referee 4*: P17, L7: "…despite the study are being the best…"

**Reply (Page 18, Line 3)**: Corrected.

*Referee 4*: P17, L9: As I understand it you did not measure primary production rates, you are inferring primary productivity from POC concentration. Therefore it may be more accurate here to say biomass rather than primary productivity. Again, to my knowledge you did not collect data on community composition and therefore I am concerned that this statement is overly speculative for your concluding remarks.

**Reply**: In our study, primary productivity was inferred from the good positive relationship between $\delta^{13}C$ and POC, instead from POC alone (see Page 12, Lines 24–L26 in the original version). In terms of phytoplankton biomass, Chl *a* may be an ideal parameters than POC.

Sufficient information on the community composition of ECS were referred in Page 13, Lines 1-15 of the original text, and that information is consistent with our $\delta^{13}C_{POC}$ results and concluding remarks of our study.

*Referee 4*: P17, L11: Again, linking changes in $\delta^{15}N_{PN}$ when no $NH_4^+$ or $NO_3^-$ data or uptake data are available is perhaps overly speculative.

**Reply**: We agree with the referee's view and we therefore mentioned that the possibility of such mechanism "needs to be substantiated by the nutrient data in future studies" (**Page 18, Line 13**).

*Referee 4*: P17, L20: Please expand on the link between the DCM and the inner shelf mud-belt that accumulated during the Holocene.

**Reply:** In this sentence, we stress that $\delta^{13}C$ values of DCM of this study can provide an ideal marine organic carbon end-member to evaluate the carbon burial along the mud-belt during

[revised manuscript text omitted]